# Deep-sea benthic communities and oxygen fluxes in the Arctic Fram Strait controlled by sea-ice cover and water depth

Ralf Hoffmann[1], Ulrike Braeckman[2,3], Christiane Hasemann[1], Frank Wenzhöfer[1, 3]

[1]Alfred Wegener Institute, Helmholtz Centre for Polar- and Marine Research, Am Handelshafen 12, 27570 Bremerhaven, Germany
[2]Ghent University, Marine Biology Research Group, Krijgslaan 281 S8, 9000 Gent, Belgium
[3]Max Planck Institute for Marine Microbiology, Celsiusstraße 1, 28359 Bremen, Germany

*Correspondence to*: Ralf Hoffmann (ralf.hoffmann@awi.de)

**Abstract**

Arctic Ocean surface sea-ice conditions are linked with the deep sea benthic oxygen fluxes via a cascade of inter-dependencies across ecosystem components such as primary production, food supply, activity of the benthic community, and their functions. Additionally, each ecosystem component is influenced by abiotic factors such as light availability, temperature, water depth, and grain size structure. In this study, we investigated the coupling between surface sea-ice conditions and deep-sea benthic remineralisation processes through a cascade of inter-dependencies in Fram Strait. We measured sea-ice concentrations, a variety of different sediment characteristics, benthic community parameters, and oxygen fluxes at 12 stations of the LTER HAUSGARTEN observatory, Fram Strait, at water depths of 275–2500 m. Our investigations reveal that the Fram Strait is bisected into two long-lasting and stable regions: (I) a permanently and highly sea-ice covered area and (II) a seasonally and low sea-ice covered area. Within the Fram Strait ecosystem, sea-ice concentration and water depth are two independent abiotic factors, controlling the deep-sea benthos. Sea-ice concentration correlated with the available food and water depth with the oxygen flux. In addition, both abiotic factors sea-ice concentration and water depth correlate with the macrofauna biomass. However, at water depths >1500 m the influence of the surface sea-ice cover is minimal with water depth becoming the more dominant. Benthic remineralisation across the Fram Strait on average is ~ 1 mmol C m$^{-2}$d$^{-1}$. Our data indicate that the portion of newly produced carbon that is remineralised by the benthos is 5 % in the seasonally low sea-ice covered eastern part of Fram Strait but can be 14 % in the permanently high sea-ice covered western part of Fram Strait. Here, by comparing a permanently sea-ice covered area with a seasonally sea-ice covered area, we discuss a potential scenario for the deep-sea benthic ecosystem in the future Arctic Ocean, in which an increased surface primary production may lead to increasing benthic remineralisation at water depths <1500 m.

## Copyright statement

# 1 Introduction

Benthic deep-sea remineralisation depends on primary production and is as such closely linked with primary production patterns, a process known as pelagic–benthic coupling (Graf, 1989). The relationship, however, includes many and partly inter-dependent factors. Benthic deep-sea remineralisation is positively correlated with surface primary production (Graf et al., 1995; Wenzhöfer and Glud, 2002; Smith et al., 2016), which is in turn controlled by light availability and nutrient supply (Kirk, 2011; Cherkasheva et al., 2014; Fernández-Méndez et al., 2015). Only the annual new production, however, leaves the euphotic zone (Platt et al., 1989) to supply the benthos with organic carbon. Benthic remineralisation is negatively correlated to water depth (Jahnke et al, 1990; Jahnke, 1996; Wenzhöfer and Glud, 2002), a consequence of pelagic remineralisation (Rullkötter, 2006; Belcher et al., 2016) which reduces the benthic food supply. After organic carbon reaches the seafloor, it may be ingested and remineralised by the benthic community. Benthic community parameters, e.g. biomass, density, structure, and functionings of various fauna size classes, are controlled by food supply (and thus by primary production) and water depth (Piepenburg et al., 1997; Flach et al., 2002; Smith et al., 2008) but also by sediment properties (Wheatcroft, 1992; Vanreusel et al., 1995). Benthic remineralisation rates also depend on benthic community biomass (Glud et al., 1994), and can, in turn, be enhanced if the benthic community intensifies oxygenation of the seafloor (Glud, 2008). Therefore, the ecosystem processes of primary production, pelagic remineralisation, and benthic remineralisation, as well as the ecosystem components of benthic community biomass, density, and structure are controlled by abiotic and biotic factors, creating a cascade of inter-dependencies from the ocean's surface zone of primary production to and within the deep-sea benthos.

In the Arctic Ocean, pelagic–benthic coupling is assumed to be stronger relative to temperate and tropical waters (Ambrose and Renaud, 1995; Graf et al., 1995; Grebmeier and Barry, 2007). A pan-arctic benthic remineralisation model showed a better fit when water depth and benthic chlorophyll data (representing food supply from primary production) were taken into account when compared to a model using only water depth as the controlling factor (Bourgeois et al., 2017). This indicates that surface primary production patterns and water depth are both relevant factors controlling benthic remineralisation in the Arctic Ocean. The occurrence of sea ice in the Arctic Ocean, however, ultimately reduces the light availability and thereby suppresses primary production (Arrigo et al., 2008; Bourgeois et al., 2017). As a consequence, climate change induced alterations in the sea-ice cover will likely influence biogeochemical cycles in the Chukchi and Beaufort Sea (Harada, 2015). Boetius and Damm (1998) also found convincing correlations between sea-ice cover, benthic chlorophyll and benthic carbon remineralisation in the Laptev Sea. However, the principal factor controlling microbial activity in their study was most likely the supply of labile organic matter such as chloroplastic pigment equivalents (CPE; Thiel, 1978), proteins and dissolved free amino acids. Therefore, the strength of the relationship between sea-ice cover (controlling primary production) and benthic remineralisation, even if assumed as direct and strong, needs to be considered in greater detail (Renaud et al., 2008).

We were interested in the question, of whether it is possible to link contrasting sea-ice conditions between the eastern and western Arctic Fram Strait (Soltwedel et al., 2005; Soltwedel et al., 2015; Spielhagen et al., 2015) with deep-sea benthic oxygen consumption through a cascade of inter-dependencies. Benthic oxygen fluxes thereby represent benthic remineralisation rates of carbon (Thamdrup and Canfield, 2000; Wenzhöfer and Glud, 2002; Smith et al., 2013). Our study provides sea-ice concentrations, sediment properties, biogenic sediment compounds, benthic community parameters, and benthic oxygen fluxes from 12 stations across the Arctic Fram Strait at water depths from 275 m to 2500 m. We hypothesise that the contrasting sea-ice conditions in the eastern and western Fram Strait lead to differences between parameters representing the cascade of inter-dependencies and result in contrasting benthic oxygen fluxes. Furthermore, our results allow us to estimate the portion of newly produced carbon that is remineralised by the benthic ecosystem. Furthermore, by comparing a permanently sea-ice covered area with a seasonally sea-ice covered area (western and eastern Fram Strait, respectively), we discuss a potential scenario for how this deep-sea benthic ecosystem may differ in the future Arctic Ocean.

## 2 Material and Methods

### 2.1 Study area and field sampling

The Fram Strait is located in the northern Greenland Sea and forms a large passage (ca. 500 km wide) between northeast Greenland and the Svalbard archipelago (Fig. 1). It provides the only exchange route of intermediate and deep water masses between the Arctic and the Atlantic Ocean (Soltwedel et al., 2005; Forest et al., 2010). Two main currents influence the upper 300 m of Fram Strait waters (Manley, 1995): the East Greenland Current (EGC) and the West Spitsbergen Current (WSC). The EGC is located in the western Fram Strait and transports cold, less saline and nutrient poor (1 °C, ≤34) Arctic waters southward (Manley, 1995; Mauritzen et al., 2011; Graeve and Ludwichowski, 2017a, b). In contrast, the WSC, located in the eastern Fram Strait, transports warmer, nutrient-rich Atlantic waters of higher salinity (>3 °C, >34) northward (Manley, 1995; Mauritzen et al., 2011; Graeve and Ludwichowski, 2017a, b). Approximately 22 % of the WSC is recirculated as the Return Atlantic Current (RAC). The remaining current bifurcates into the Svalbard Branch (SB; 33 %) and the Yermak Branch (YB; 45 %) following the Svalbard islands or flowing along the north-west flanks of the Yermak Plateau, respectively (Schauer, 2004). A high sea-ice cover is reported for the western Fram Strait and a low sea-ice cover for the eastern Fram Strait (Soltwedel et al., 2005; Soltwedel et al., 2015; Spielhagen et al., 2015). The sea-ice cover is relatively stable within the Fram Strait, even in summer (Comiso et al., 2008; Soltwedel et al., 2015). However, the average age of monitored sea ice is becoming younger, at a rate of 0.6 years per decade (2001–2012, Krumpen et al., 2015), correlating with the progressive reduction in observed sea-ice thickness (Renner et al., 2014; Krumpen et al., 2015). The onset of the spring bloom usually starts in Mai (Cherkasheva et al., 2014).

Two sampling campaigns were conducted at the long-term ecology research observatory HAUSGARTEN (Soltwedel et al., 2005) in the Fram Strait with RV *Polarstern*, expeditions "PS85" (6th June – 3rd July 2014) and "PS93.2" (22nd July – 15th August 2015). Samples were taken at five stations at the East Greenland continental slope (EG area) and at

seven stations at the West Spitsbergen continental slope (WS area) at water depths of 275–2500 m (Fig. 1, Table 1). Thereby the stations in the EG area (namely EG I, EG II, EG III, EG IV and EG V) and in the HG area (namely SV I, HG I, SV IV, HG II, HG III, HG IV, and N5) form a bathymetric transect with a similar bottom slope of ~11°. The station EG IV includes two sites which are located <2 km from each other (Table 1) and the stations HG I, HG II, HG III, and HG IV were sampled during both sampling years, 2014 and 2015.

Sediment sampling was performed using a multiple corer (MUC) with eight tubes and autonomous benthic lander systems (Reimers, 1987; Glud et al., 1994) equipped with three benthic chambers and a sediment profiler with oxygen sensors (Donis et al., 2016). A detailed list of the number of used samples per station for the determination of different parameters is given in Supplement Table S1.

## 2.2 Sea ice data

Daily sea ice concentrations for each of the analysed stations were obtained from the Center for Satellite Exploitation and Research (CERSAT) at the Institut Français de Recherche pour l'Exploitation de la Mer (IFREMER), France (Ezraty et al., 2007) and were previously published (Krumpen, 2017), except for station EG V. Sea-ice concentration was calculated based on the ARTIST Sea Ice (ASI) algorithm developed at the University of Bremen, Germany (Spreen et al., 2008). The data used in this study covered the period 1$^{st}$ September 2001 – 31$^{st}$ August 2015 (long-term data) with a 12.5 x 12.5 km² spatial resolution around each station. Satellite mismeasurements, which were <0.5% of the long-term data, resulted in an algorithm output value of "128" and were omitted from the dataset. Three additional and shorter time periods were chosen for analysis of sea ice concentrations: the period a year before sampling, the period since May 1$^{st}$ until date of sampling, and a period of one month before sampling. The period a year before sampling was determined as 1$^{st}$ July 2013 – 30$^{th}$ June 2014 for stations sampled in 2014 and 1$^{st}$ August 2014 – 31$^{st}$ July 2015 for stations sampled in 2015. From each dataset (long-term and short-term) the sea-ice cover and the percentage of days with sea-ice cover were extracted.

## 2.3 Sediment compounds and properties

Various biogenic sediment compounds including grain size, water content, chlorophyll a (Chl *a*) and phaeopigment concentrations (Phaeo), portion of total organic carbon (TOC), phospholipids concentrations, protein concentrations, portion of organic matter, and the bacterial enzymatic turnover rate (FDA) as bacterial activity proxy were determined from the sediments sampled by the MUC and chambers of the autonomous benthic lander system. Generally, three pseudo-replicates from each MUC (sampled from different sediment cores, inner MUC tube diameter = 9.5 cm) were taken. Sediment samples of the 0–5 cm layer were taken by means of syringes with cut-off tips (1.17 / 3.14 cm² cross-sectional area). Samples for FDA, Chl *a*, and Phaeo were immediately analysed on board. All other samples were shock frozen at -80°C and stored at -20°C until they were analysed at the home laboratory. Sediment samples, taken by the benthic chambers of the autonomous lander system, were treated similarly.

The grain size partitions were determined with a Malvern Mastersizer 2000G, hydro version 5.40. The Mastersizer utilizes a laser diffraction method with a measuring range of 0.02–2000 μm. The water content of the sediment was determined by the difference in weight before and after drying at 105°C. The bioavailability of phytodetritus at the seafloor was assessed by analysing sediment bound Chl *a* and Phaeopigments. Chloroplastic pigments were extracted in 90 %

acetone and measured with a TURNER fluorometer (Shuman and Lorenzen, 1975). The bulk of pigments (Chl *a* plus Phaeo) are termed chloroplastic pigment equivalents (CPE) after Thiel (1978). Additionally, the ratio of Chl *a* to CPE (% Chl *a*), which serves as a quality indicator of the labile organic matter (Cathalot et al., 2015), and the ratio of Chl *a* to Phaeo were calculated. As the Arctic benthos receives mainly one large food input per year, the spring bloom (Wassmann, 2011), and Chl *a* is degraded to Phaeo over time (Kirk, 2011), the benthic Chl *a*/Phaeo ratio will therefore be at its highest when the

spring bloom arrives on the seafloor, and decrease thereafter with ongoing degradation. Therefore, the Chl *a*/Phaeo ratio can act as a proxy for time since spring bloom arrival. The percentage of the TOC was measured by combustion using an ELTRA CS2000 with infrared cells. To indicate the quantity of cell wall material, phospholipids were measured following Findlay et al. (1989) with modifications after Boetius and Lochte (1994). Particulate proteins, defined as γ-globulin equivalents (Greiser and Faubel, 1988), were measured to differentiate between living organisms and detrital organic matter

in the sediments. Hereafter, particulate proteins will be referred to only as proteins. The organic matter volume was determined as the ash free dry weight after combustion (2 h, 500°C). Bacterial enzymatic turnover rates were calculated using the fluorogenic substrate fluorescein-di-acetate (FDA) as an indicator of the potential hydrolytic activity of bacteria (Köster et al., 1991).

## 2.4 Benthic community parameters

For the bacterial density determination, sediment subsamples were taken with modified syringes (1.17 cm² cross-sectional area) from MUC recovered sediment cores after oxygen flux measurements were performed and from benthic chambers. The first centimetre of each sample, generally holding the highest bacterial density (Quéric et al., 2004), was stored in a 2 % filtered formalin solution at 4 °C. The acridine orange direct count (AODC) method (Hobbie et al., 1977) was used to stain bacteria in the subsamples and subsequently bacteria were counted with a microscope (Axioskop 50, Zeiss) under UV–light

(CQ-HXP-120, LEj, Germany).

For the determination of the meiofauna density and identification of meiofauna taxa, sediment subsamples were taken with modified syringes (3.14 cm² cross-sectional area) from MUC recovered sediment cores after oxygen flux measurements were performed and from benthic chambers. The first centimetre of each sample, usually supporting the highest meiofauna density (Górska et al., 2014), was stored in borax buffered 4 % formaldehyde solution at 4 °C. The

samples were sieved over a 1000 μm and 32 μm mesh. Both fractions were centrifuged three times in a colloidal silica solution (Ludox TM-50) with a density of 1.18 g cm$^{-3}$ and stained with Rose Bengal (Heip et al., 1985). Afterwards, the taxa were identified and counted at order level. Foraminifera were not considered, as the extraction efficiency of Ludox for different groups of foraminifera is insufficient for a quantitative assessment of the group. Therefore, only metazoan

meiofauna was considered herein, and henceforth the term meiofauna will be used to refer only to metazoan meiofauna organisms.

After taking subsamples for bacteria and meiofauna density determination, the remaining sediment from MUC recovered sediment cores and from the benthic chambers was used for macrofauna taxonomical identification, and density and biomass determination. For macrofauna analyses, the 0–5 cm layer from MUC sediment cores and the entire remaining sediment from the benthic chambers was used, sieved over a 500 μm mesh and stored in borax buffered 4 % formaldehyde and stained with Rose Bengal (Heip et al., 1985). Afterward, macrofauna taxa were identified to the highest taxonomic level (at least class level), counted and weighed (blotted wet weight).

From the macrofauna density ($Ai$) and biomass ($Bi$), together with a mobility score ($Mi$) and sediment reworking score ($Ri$) of each taxon, the community bioturbation potential ($BPc$) was calculated following Queirós et al. (2013, Eq. (1)):

$$BP_c = \sum_{i=1}^{n} \sqrt{B_i / A_i} \times A_i \times M_i \times R_i \tag{1}$$

in which $i$ represents the specific taxon in the sample. This index describes the bioturbation potential of the benthic macrofauna community.

## 2.5 Oxygen and bromide fluxes

Immediately after the retrieval of sediment cores by the MUC, a quantity of the overlying water was removed and stored separately. At least 10 cm of overlying water was maintained in the cores. The sediment of each core was carefully pushed upwards without disturbing the surface sediment layer until the sediment–water interface (SWI) was at a distance of approximately 10 cm from the upper edge of the core. A magnetic stirrer was added to the overlying water to assure a well-mixed overlaying water body. In this position, the sediment cores were stored in a water bath at in situ temperature (-0.75°C) until the start of the oxygen flux measurements.

For the determination of the ex situ diffusive oxygen uptake (DOU) at least two oxygen microprofiles per sediment core were measured simultaneously within 2 h after sampling with a vertical resolution of 100 μm. The profiling was performed by oxygen optical microsensors (OXR50, Pyroscience, Aachen, Germany) with a tip size of 50 μm diameter, a response time of <2 s and an accuracy of ±0.02 %, calibrated with a two-point calibration using air saturated and anoxic waters (by adding sodium dithionite). The overlying water in the MUC cores was magnetically stirred and the water surface was gently streamed with a soft air stream during the profiling. The maximum penetration depth of the sensors during ex situ profiling was 42 mm. For in situ DOU determination autonomous landers were used (Reimers, 1987; Glud et al., 1994; Glud, 2008). The profiling unit was equipped with electrochemical oxygen microsensors (custom made after Revsbech (1989)) and calibrated with a two-point calibration. As the first calibration point, the bottom water oxygen concentration (water sample were taken by Niskin bottle), estimated by Winkler titration (Winkler, 1888), was used. As the second calibration point, the sensor signal in the anoxic zone of the sediment (when reached) or the sensor signal in an anoxic solution of sodium dithionite recorded on board was used. The measurements started three hours after the deployment of the autonomous lander,

allowing resuspended sediment to settle beforehand. Profiling was performed with a depth resolution of 100 μm. The maximum penetration depth of the sensors during in situ profiling was 180 mm. Running average smoothed oxygen profiles from ex situ and in situ approaches were used to calculate the DOU rates across the SWI using Fick's first law (Eq. (2)):

$$DOU = -D_s \times \left[\frac{\delta O_2}{\delta z}\right]_{z=0},$$ (2)

in which $D_s$ is the molecular diffusion coefficient of oxygen in sediments at in situ temperature and salinity, and $\left[\frac{\delta O_2}{\delta z}\right]_{z=0}$ is the oxygen gradient at the SWI was calculated by linear regression from the first alteration in the oxygen concentration profile across a maximum depth of 1 mm. $D_s$ was calculated following Schulz (2006) as $D/\theta^2$, with $D$ as the molecular diffusion coefficient of oxygen in water after Li and Gregory (1974), and $\theta^2$ as $1-ln(\varphi^2)$ (Boudreau, 1997). The sediment porosity $\varphi$ was calculated following the equation of Burdige (2006, Eq. (3)):

$$\varphi = \frac{m_w/\rho_w}{m_w/\rho_w+(m_d-(S \times m_w))/\rho_s},$$ (3)

In this equation, $m_w$ is the mass of evaporated water, $\rho_w$ is the density of the evaporated water, $m_d$ is the mass of dried sediment plus salt, $S$ is the salinity of the overlying water and $\rho_s$ is the density of deep-sea sediment (2.66 g cm$^{-3}$, after Burdige, (2006)). To calculate $m_w$, $\rho_w$, and $m_d$, the weight loss of wet sediment samples was measured by weighing wet samples, drying them overnight at 70 °C, weigh them again, drying the sample for 1 h at 70 °C and weighing a second time. This procedure was repeated until the weights of the two dried samples differ not more than 0.05 %. Over all samples, 4.5 ± 1.9 % of the sediment mass was attributed to salt. Non-local mixing was observed in some microprofiles and therefore the reported DOUs for those cases are underestimations. However, within only eight of the 81 ex situ obtained oxygen microprofiles and one out of the 34 in situ obtained oxygen microprofiles showed signs of non-local mixing.

For ex situ total oxygen uptake (TOU) measurements, sediment cores were used after oxygen microprofiling (see upper paragraph in this section). The sediment cores were closed airtight with no air bubbles in the overlying water. The distance between the SWI and the edge of the lid was measured for volume calculations of the overlying water. An optical oxygen microsensor (Pyroscience, Aachen, Germany) with a tip size diameter of 50 μm was mounted in the lid, allowing a continuous measurement of the oxygen concentration in the overlying water. The sediment cores were incubated in darkness for >40 h and the overlying water kept homogenised by rotating magnets throughout. For in situ TOU measurements, benthic chambers (K/MT 110, KUM, Kiel, Germany) with an inner dimension of 20x20 cm were used. These chambers were pushed into the sediment and thereby enclosed a sediment volume of approximately 8 L and an overlying water volume of approximately 2–3 L. The oxygen concentration was measured in the overlying water continuously with an Aanderaa optode (4330, Aanderaa Instruments, Norway, two-point calibrated as described in the upper section) over an incubation period of 20–48 h. During the measurement, the overlying water was kept homogenised by a stirring cross at the inner top of the chamber. TOU from both ex situ sediment cores and in situ benthic chamber incubations were calculated using Eq. (4):

$$TOU = \frac{\delta O_2 \times V}{\delta t \times A},$$ (4)

in which $\delta O2$, $\delta t$, $V$ and $A$ represent the difference in oxygen concentration, the difference in time, the volume of the overlying water and the enclosed surface area, respectively.

Both, the diffusive and total oxygen fluxes were converted to carbon equivalents (C-DOU and C-TOU) by applying the Redfield ratio (C:O = 106:138; Redfield (1934)) in order to compare them to the carbon fixed by primary production. Modifications, as suggested by Takahashi et al. (1985) and Anderson and Sarmiento (1994), would result only in minor changes of <10 % in the benthic carbon flux.

To assess the exchange of solutes across the SWI, which results from molecular diffusion, physical advection and faunal ventilation activities, sodium bromide (NaBr) was added to the removed overlying water of the sediment cores to create a NaBr–solution of similar density as seawater (1028 g $L^{-1}$). The NaBr–solution was added to the sediment cores before the TOU incubation started. Three subsamples of water were taken during the incubation at three different times ($t_0$, $t_1$, $t_2$) and stored at 4 °C. Removed water volume of the subsampling at $t_1$ was replaced with the NaBr–seawater solution. The bromide concentrations were measured using ion chromatography. The dilution of the $t_2$–sample, due to the sampling procedure, was corrected by the known bromide concentration in the removed and the added water. The bromide exchange is represented by the bromide flux, calculated using the Eq. (5):

$$Bromide\ flux = \left( \frac{\delta Bromide\ concentration \times V}{\delta t \times A} \right), \tag{5}$$

in which $\delta Bromide\ concentration$, $\delta t$, $V$ and $A$ represent the difference in bromide concentration, the difference in time, the volume of the overlying water and the enclosed surface area, respectively.

## 2.6 Data analyses

The analysed data were obtained during two consecutive years (Table 1). To test whether there is a significant offset between sampling years, a principal component analysis (PCA) was performed on standardised (x to zero mean and unit variance) abiotic parameters (year, water depth, sea ice cover, percentage of days with sea ice cover, the portion of grain size >63 μm, median grain size) and all sediment compounds and property parameters from the 0-1cm sediment horizon, as this was the most complete dataset. Additionally, a non–parametric Wilcoxon signed rank sum test was performed on station specific mean values of both years on water content, TOC, organic matter, Chl $a$, Phaeo, protein, phospholipids, FDA, DOU and TOU following Cathalot et al. (2015). Both tests were performed only on data of stations that were sampled in both 2014 and 2015.

To reveal significant differences in measured parameters between the EG and the WS area, Students t-tests were performed. If the t-test assumption of Gaussian distribution of the data (tested with a Shapiro–Wilk test) was not met, a non–parametric Wilcoxon signed rank sum test was performed. In case of heteroscedasticity (tested with a Levene's test) a Welch two-sample t-test was carried out. The values from station SV I were excluded from the tests, due to its exceptionally shallow water depth.

To identify the most important parameters influencing the benthic Fram Strait ecosystem, a second PCA was performed in the scaling II mode on standardised (x to zero mean and unit variance) ex situ mean values of abiotic parameters (water depth, short-term sea-ice cover (year before sampling), the portion of grain size >63 µm, water content), biogenic compound parameters (Chl *a*, TOC, organic matter), oxygen fluxes (DOU, TOU), the benthic community (bacterial density, macrofauna biomass), and the *BPc*. All other parameters were excluded from the PCA as they correlated strongly (correlation >0.74, Pearson correlation, Supplement Table S2) with one of the outlined PCA parameters. This procedure results in a more resilient outcome of the PCA. Because of the shallow water depth, the values from station SV I were also excluded from the PCA. For further insights and descriptions of the usage and interpretation of a PCA, the reader is referred to Buttigieg and Ramette (2014).

Water depth and sea ice have a profound impact on benthic oxygen fluxes (Wenzhöfer and Glud, 2002; Harada, 2015). To investigate the influence of water depth and sea ice in our data, the stations were merged into two sea-ice cover categories. First, a "high sea-ice concentration" area (HSC), which include stations with a short-term (a year before sampling) mean sea-ice concentrations of ≥ 30 %. Second, a "low sea-ice concentration" area (LSC), which include stations with a short-term (a year before sampling) mean sea-ice concentrations of <30 %. Regression analysis was used to test the water depth dependence of sediment compounds and property parameters, the benthic community parameters, the oxygen fluxes, and parameters of the macrofauna mediated environmental functions within the HSC and LSC categories. If the residuals over the slope did not follow the Gaussian distribution (tested with a Shapiro–Wilk test), values were transformed, either by square root or logarithmic transformation. Individual values that failed due to technical failure or mismeasurements were removed before statistical analyses. For all above mentioned statistical treatments, R Statistical Software (version 3.4.0) was used.

Analyses of the multivariate meio- and macrofauna community structure were based on square root transformed density and biomass data of sediment core replicates. Non–metric multidimensional scaling (MDS, (Kruskal, 1964)) and hierarchical cluster analysis with group average clustering were used to present the multivariate similarities between samples based on Bray–Curtis similarity. Significant multivariate differences between pre-defined group structures within the meio- and macrofaunal data were tested by the ANOSIM procedure (ANalysis Of SIMilarity) based on Clarke's R statistic (Clarke and Warwick, 1994) with 9999 permutations. The SIMPER (SIMilarity PERcentage) routine was applied to determine the contribution of certain meio- and macrofauna taxa towards the discrimination between sea-ice cover categories and water depth categories. Differences (p < 0.05) between HSC, LSC and water depth regarding macrofauna density and macrofauna biomass were examined using a two-way crossed PERMANOVA (PERMANOVA+ for PRIMER; Anderson, 2005; Anderson et al., 2007) analysis with "site" (levels "HSC" and "LSC") or "water depth" (levels:1000, 1500, 2000, 2500 m) as fixed factors. The significance level was set at 0.05. Significant main PERMANOVA tests were followed by pairwise PERMANOVA tests. Permutational P-values (PPERM) were interpreted when the number of unique permutations was >100; alternatively, Monte Carlo P-values (PMC) were considered. Bray–Curtis similarity was used to construct resemblance matrices. Data were standardised and fourth–root transformed (to down weigh the importance of the most

dominant taxa) prior to the construction of resemblance matrices. The station SV I and the in situ stations HG I Lander and HG IV Lander were excluded from these tests, due to shallowness (SV I) and differences in sampling devices (benthic chambers instead of MUC). All analyses of multivariate community structure were performed using the routines implemented in PRIMER vers. 6.1.15 (Clarke and Gorley, 2006; Anderson et al., 2007). Results are expressed as means ±
standard deviation.

## 3 Results

### 3.1 Short- and long-term sea ice concentration comparison between the EG and WS area

Short-term and long-term data of the mean sea-ice concentrations and the percentage of sea-ice covered days were in a similar range (Table 2). Both parameters decreased from west to east with a sharp drop in concentrations between N5 and
10 HG IV in both the short and the long-term datasets (Table 2). Therefore, the categorisation into a high sea-ice covered area (HSC) and a low sea-ice covered area (LSC) was introduced. The HSC included all East Greenland stations (EG I–V) and the most northern West Spitzbergen station N5, while the LSC included the remaining West Spitzbergen stations (HG I–IV, SV I, and SV IV).

As expected, the east Greenland stations showed the highest sea ice concentration due to the influence of the East
Greenland current. The short-term sea-ice concentration in the EG area one year before sampling was highest at EG I with 82 ± 20 % (n = 364) and lowest at EG V with 56 ± 34 % (n = 364). In the WS area, sea-ice concentration was highest at N5 with 40 ± 31 % (n = 365) and lowest at SV IV with 0.1 ± 2 % (n = 365). The percentage of days, which showed sea-ice cover, during the short-term period in the EG area was highest at EG I, EG II and EG III (each with 100 %) and lowest at EG V (93 %). In the WS area the percentage of days, which showed sea-ice cover, during the short-term period was highest at
N5 (82 %) and lowest at SV IV (>0.1%, Table 2). This pattern also occurred in the other short-term datasets and in the long-term dataset. The latter indicated that the sea-ice cover in terms of sea-ice concentration was stable across the Fram Strait throughout the last 15 years (Fig. 2, Supplement Table S3).

### 3.2 Sediment properties and benthic biogenic compounds in the EG and WS area

Sediment properties and biogenic compound values at the deeper stations (>1500 m) in the EG and WS area were in the
25 same range. In contrast, shallow stations (≤ 1500 m) of the WS area showed higher values compared to shallow stations of the EG area (Table 3). This indicated a higher variability in the WS area for most of the determined parameters (Fig. 3).

The median grain size in the EG area ranged between 13 ± 1 μm (n = 15) at EG I and 74 ± 30 μm (n = 15) at EG V and in the WS area between 10 ± 3 μm (n = 15) at N5 and 24 ± 5 μm (n = 30) at HG IV. The portion of sediment grain size >63 μm in the EG area ranged between 4 ± 2 % (n = 15) at EG I and 52 ± 7 % (n = 15) at EG V and in the WS area between
30 11 ± 6 % (n = 30) at HG I and 25 ± 5 % (n = 30) at HG IV. The water content in the EG area ranged between 42 ± 6 % (n = 15) at EG V and 51 ± 7 % (n = 15) at EG I and in the WS area it ranged between 51 ± 14 % (n = 15) at SV I and 66 ± 5 % (n

= 30) at HG I. The porosity in the EG area ranged between 0.69 ± 0.06 % (n = 15) at EG V and 0.77 ± 0.06 (n = 15) at EG I and in the WS area it ranged between 0.77 ± 0.06 % (n = 30) at HG I and 0.88 ± 0.04 % (n = 30) at HG II. Results of all stations are listed in Table 3. Median grain size, water content and porosity differed significantly between the WS and EG area, while the portion of sediment grain size >63 µm was similar (Supplement Table S4).

5        The sediment bound Chl *a* concentration ranged between 0.4 ± 0.3 µg ml$^{-1}$ sediment$^{-1}$ (n = 15) at EG III and 12.7 ± 3.1 µg ml$^{-1}$ sediment$^{-1}$ (n = 15) at SV I (Table 3) and differed significantly between the EG and WS area (Figure 3, Supplement Table S4). A similar pattern was found for sediment bound Phaeo concentrations and CPE concentration with over 4 –times higher median values in the WS area compared to the EG area (Figure 3). The Chl *a*/CPE and Chl *a*/Phaeo ratios did not differ between the EG and WS area (Supplement Table S4), which indicates that the benthic community in both areas fed on a similar food quality and received the spring bloom food supply at the same time, respectively. Sediment bound TOC ranged between 0.44 ± 0.04 % (n = 15) at EG II and 1.58 ± 0.27 % (n = 15) at SV I and differed between the EG and WS area, similar to organic matter, which ranged between 3.45 ± 0.6 % (n = 15) at EG II and 12.0 ± 4.2 % (n = 30) at HG III (Table 3, Figure 3, Supplement Table S4). Proteins, lipids, and FDA also differed between the EG and WS area with 5.6 times, 2.3 times, and 1.8 times higher median values in the WS area, respectively (Figure 3, Supplement Table S4).

### 3.3 Benthic communities and community functions in the EG and WS area

Overall, 17 meiofauna taxa and 18 macrofauna taxa were identified (Supplement Tables S5, S6, S7). The meiofauna density was dominated by nematodes (86 %), the only taxon present at each station. Crustaceans were the second most dominant group with 4.5 % nauplii and 3.5 % Copepoda. The macrofauna density was dominated by polychaetes (40 %), followed by Copepoda (26 %), and Nematoda (12 %). Polychaetes (57 %) also dominated the macrofauna biomass, followed by Bivalvia (16 %) and Porifera (14 %). The mean values of the benthic faunal community parameters meiofauna density, macrofauna density, and macrofauna biomass were 1.5, 4.6, and 2.5 times higher in the WS area than in the EG area respectively (Supplement Table S2) and differed significantly from each other (Supplement Table S4). In contrast, the bacterial density was similar between the EG and WS areas but showed a greater variability within the WS area (Supplement Table S4, Fig. 3).

        The solute exchange across the SWI, represented by the bromide flux, did not differ between the EG and WS area (Supplement Table S4). The observed lack of difference may have a methodological explanation. Bromide flux incubations were performed on 40 sediment cores but measurements from 13 sediment cores were omitted (seven from EG area, six from WS area), as either the calculations revealed a positive flux or the residuals were not homogenously distributed across the decreasing slope of the bromide concentration over time or slopes were not significantly different from zero. The community bioturbation potential, represented by the *BPc*, was also similar between the EG and WS area (Supplement Table S4) but the median *BPc* at the WS area was 2.9 times higher than in the EG area (Fig. 3). This indicates that the benthic macrofauna community in the WS area is potentially able to rework the sediment to a larger degree than the benthic macrofauna community in the EG area.

## 3.4 Benthic remineralisation

All oxygen microprofiles showed decreasing oxygen concentrations across the SWI (Supplement Fig. S1) and steepness of oxygen gradients varied among microprofiles and across various stations. Further, all sediment core incubations resulted in decreasing oxygen concentrations in the overlying water, with varying gradients measured among sediment cores and across various stations. The mean DOU in the EG area ranged between $0.4 \pm 0.1$ mmol $O_2$ $m^{-2}d^{-1}$ (n = 10) at EG V and $1.0 \pm 0.1$ mmol $O_2$ $m^{-2}d^{-1}$ (n = 10) at EG II. In the WS area, DOUs at stations within the same water depth range as the EG stations ranged between $0.5 \pm 0.2$ mmol $O_2$ $m^{-2}d^{-1}$ (n = 8) at HG IV and $2.1 \pm 0.6$ mmol $O_2$ $m^{-2}d^{-1}$ (n = 8) at SV IV. At the shallow station SV I the DOU reached $3.0 \pm 1.7$ mmol $O_2$ $m^{-2}d^{-1}$ (n = 6, Table 3). The mean TOU in the EG area ranged between $0.9 \pm 0.3$ mmol $O_2$ $m^{-2}d^{-1}$ (n = 2) at EG I and $1.6$ mmol $O_2$ $m^{-2}d^{-1}$ (n = 1) at EG II. Similar mean TOU values were measured in the WS area, at stations within the same water depth range as the EG stations. TOU values ranged between $0.5 \pm 0.2$ mmol $O_2$ $m^{-2}d^{-1}$ (n = 5) at HG IV Lander and $1.9 \pm 0.6$ mmol $O_2$ $m^{-2}d^{-1}$ (n = 5) at HG I. At the shallow SV I station TOU reached $5.1 \pm 0.3$ mmol $O_2$ $m^{-2}d^{-1}$ (n = 3, Table 3). DOU differed significantly between the WS and EG areas, whereas TOU was similar between areas (Fig. 3, Supplement Table S4). The mean DOU/TOU ratio, which describes the fraction of the total community mediated oxygen flux covered by the microbial-mediated oxygen flux (Glud, 2008,) across the entire Fram Strait was $0.8 \pm 0.3$, with $0.6 \pm 0.2$ in the EG area and $0.9 \pm 0.3$ in the WS area, indicating that the total oxygen uptake is mainly microbial-mediated. In the EG area, DOU values showed no correlation with water depth, while in the WS area the correlation of DOU with water depth was significant (Fig. 4) and showed greater variability (Fig 3). In contrast, TOU values in the EG and in the WS areas showed no correlation with water depth (Supplement Fig. S3), but again, the variability of TOU values was higher in the WS area (Fig. 3). C-DOU and C-TOU followed the same trends as DOU and TOU, respectively, and are listed in Table 3.

## 3.5 Relationships of the benthic remineralisation with the benthic community and environmental parameters

The PCA, which included only abiotic parameters (year, water depth, sea ice cover, the percentage of days with sea ice cover, portion of grain size >63 μm, and median grain size) and biogenic compounds within the first sediment centimetre (Chl $a$, Phaeo, CPE, TOC, organic matter, lipids, and proteins), revealed differences between the sampling years 2014 and 2015 (Supplement Fig. S2). The difference occurred only in the second dimension, which explained 15.4 % of the variability and was primarily influenced by the parameters Phaeo and CPE (Supplement Table S8). The non-parametric Wilcoxon signed rank sum test of the station specific mean values revealed no differences (p > 0.05) for any of the parameters between the sampling years. This follows Henson et al. (2016), in which they showed that it can take at least 15 years of continuous data to prove temporal trends in ocean biogeochemistry; and even longer in high latitudinal areas. Therefore, it is more likely that statistically revealed differences between sampling years reflect spatial variability rather than time-related differences. Consequently, the data from stations sampled in 2014 and 2015 were merged and thus this study focuses solely on spatial patterns.

The PCA on station specific, ex situ obtained mean values (Fig. 5) revealed that water depth was positively correlated with median grain size and negatively correlated with DOU, TOU, bacterial density, and *BPc*. Sea-ice concentration was negatively correlated with porosity, Chl *a*, TOC, organic matter, and solute exchange. Similarly, macrofauna biomass was negatively correlated with water depth, sea-ice concentration, and median grain size. The stations of the WS and EG area both followed the water depth gradient with shallower stations exhibiting higher oxygen fluxes. However, stations in the EG area were strongly influenced by the sea-ice cover, contained less organic matter and Chl *a*, and macrofauna biomass, when compared with WS stations. The two dimensions of the plot explained 72 % of the total variability of the data (Fig. 5). The eigenvalues indicated that 'Chl a', 'TOC', and 'Macrofauna biomass' (-0.89, -0.88, -0.83, respectively) were responsible for the gradient along the x-axis and 'Bacterial density', 'water depth', organic matter' and 'sea-ice concentration' (0.59, -0.57, -0.54, respectively) for the gradient along the y-axis.

Across the HSC area, DOU and TOU were not linearly dependent on water depth (Fig. 4, Supplement Fig. S3, Supplement Table S9). The same was found for water content, FDA, meiofauna and macrofauna densities, macrofauna biomass, and solute exchange across the SWI. Otherwise, the fraction of sand in the sediment (% of grain size >63 μm), Phaeo, CPE, the Chl *a*–Phaeo ratio, the Chl *a*–CPE ratio, and lipids were positively linearly dependent on water depth across the HSC area and the BPc was negatively linearly dependent on water depth. Across the LSC area, the DOU was negatively linearly dependent on water depth, as well as sediment water content, Chl *a*, Phaeo, CPE, FDA, bacteria density and bioturbation potential. Contrastingly, TOU, Chl *a*–Phaeo ratio, protein, meio- and macrofauna densities, macrofauna biomass, and the solute exchange were not water depth dependent in the LSC area. Within both sea-ice categories HSC and LSC, no linear water depth dependencies were found for median grain size, TOC, and organic matter as the residuals over the slopes did not follow a Gaussian distribution. This also applied for Chl *a*, protein, and bacteria density across the HSC area and for the portion of grain size >63 μm, the Chl *a*–CPE ratio, and lipids across the LSC area (Supplement Table S9).

The ANOSIM (Global R = 0.122, p = 0.063) and SIMPER (33 % dissimilarity) routine revealed essentially no differences between the HSC and LSC area regarding the meiofauna community based on density (Table 4). Regarding macrofauna communities based on density (Global R = 0.257, p = 0.007) and biomass (Global R = 0.238, p = 0.003), the ANOSIM revealed significant but weak differences between the HSC and LSC area. SIMPER routine results indicated dissimilarities of 56 % for the macrofauna density and 76 % for the macrofauna biomass between the HSC and LSC areas. The taxa which contributed most to the average similarity within and to the average dissimilarity between the HSC and LSC area are given in Supplement Table S10. The ANOSIM results for water depth groups showed that bathymetry could at least partly explain the dissimilarity in meiofauna communities based on density (Global R = 0.219; p = 0.01), even if the difference was weak. The SIMPER analysis, however, showed that the observed differences in meiofauna density regarding water depth were mainly due to the marked difference between the shallowest station (SV I at 275 m) and all other stations deeper than 1000 m (dissimilarity >50 %, Supplement Table S11). ANOSIM results for macrofauna communities based on density (Global R = 0.2, p = 0.008) and biomass (Global R = 0.346, p = 0.0001) revealed significant but also weak differences between water depth categories with >50 % dissimilarity between all water depth categories for macrofauna

density (except between 1000 m and 1500 m) and macrofauna biomass (SIMPER, Supplement Table S11). Further, the two-way crossed PERMANOVA revealed that the sea-ice coverage (LSC and HSC) explains a significant (p = 0.008) portion of the macrofauna density variability. The results of the pairwise test showed that only the neighbouring water depth classes 1000 m and 1500 m showed no significant differences (p = 0.45) whereas all other pairwise comparisons showed significant differences between water depths (Supplement Table S13). For macrofauna biomass, the two-way crossed PERMANOVA revealed that the interaction of sea-ice cover and water depth explains a significant (p = 0.034) portion of the macrofauna biomass variability. The results of the pairwise test showed that only the water depth classes 1000 m and 2500 m showed significant differences (p = 0.0187), while all other pairwise comparisons showed no significant differences between water depths (Supplement Table S13).

## 4 Discussion

### 4.1 Linking contrasting sea-ice conditions with benthic oxygen fluxes

The main aim of this study was to link sea-ice conditions within the Arctic Fram Strait with the deep-sea benthic oxygen fluxes over a cascade of inter-dependencies. Our results documented two contrasting sea-ice concentration regimes in the Fram Strait with a high sea-ice concentration in the western Fram Strait and a low sea-ice concentration in the eastern Fram Strait (Table 2, Fig. 2). This is similar to sea-ice concentration snapshot observations by Schewe and Soltwedel (2003) and satellite observations of Krumpen et al. (2015). The observed pattern can be explained by the two major current systems present in the Fram Strait (Schauer, 2004), the EGC transporting cold, nutrient-poor water and sea ice from the central Arctic Ocean southwards into the EG area and the WSC transporting warmer, nutrient richer and sea-ice free water from the Atlantic Ocean northwards into the WS area (Manley, 1995; Mauritzen et al., 2011; Graeve and Ludwichowski, 2017a, b). If there were a strong link between sea-ice conditions and deep-sea benthic oxygen fluxes, we would expect contrasting primary production, benthic food supply, benthic community parameters and benthic oxygen fluxes between the EG and the WS area.

The results of Pabi et al. (2008) showed that in the Fram Strait the annual primary production pattern has historically followed the general sea-ice concentration pattern and that the annual primary production has been up to 10 times greater in the WS area than in the EG area. Thus, sea-ice concentration reflects and guides the general primary production pattern in the Fram Strait. As the sampling in the current study was performed in mid/end of June 2014 and July/August 2015, it is very likely that the spring bloom, which usually starts in May (Cherkasheva et al., 2014), had finished. This is indicated by lower nutrient concentrations at water depth ≤50 m compared to the nutrient concentrations between >50–300 m water depths (Graeve and Ludwichowski, 2017a, b). The N:P ratio in the upper 50 m during the expeditions was six and seven in the EG and WG area, respectively (Graeve and Ludwichowski, 2017a, b), indicating that primary production was nitrate limited, as reported from the permanently sea-ice covered central Arctic Ocean (Tremblay et al., 2012; Fernández-Méndez et al., 2015). Furthermore, the timing of our sampling suggests that the increased carbon supply by the spring bloom had

already reached the seafloor and enhanced the benthic remineralisation (Graf, 1989) in both areas. The pattern of contrasts between the EG and WS areas continued in the benthic food supply, which has also been observed by Boetius and Damm (1998) for areas with contrasting sea-ice cover at the continental margin of the Laptev Sea.

Continuing the cascade of inter-dependencies, benthic community parameters should follow the same pattern as the sea ice at the surface and the benthic food supply parameters. Indeed, there were differences between the EG and WS areas regarding meiofauna density and macrofauna density but not in the macrofauna biomass. The macrofauna community structure also differed between areas with high and low sea-ice cover, when sea ice and water depth was taken into account. The performed PERMANOVA confirmed the influence of water depth on the macrofauna community and indicated that water depth is a considerable factor, in addition to that of sea-ice cover. Consequently, in the low sea-ice covered WS area macrofauna is mainly influenced by the abiotic factor water depth (Soltwedel et al., 2015), whereas in the highly sea-ice covered EG area the abiotic factor sea-ice cover co-acts or even replaces water depth as the most influencing abiotic factor.

Benthic remineralisation across the Fram Strait, represented by oxygen consumption, was not correlated with sea-ice concentrations or benthic food supply, only with water depth (Fig. 5). This is in contrast to our expectations and to the findings of Boetius and Damm (1998). However, a PCA only shows correlations, not in themselves sufficient to prove causal relationships, and a PCA does not test for the significance of these relationships. Therefore, we tested the significance of the correlation of water depth with DOU within the sea-ice concentration categories HSC and LSC, which revealed a slightly different pattern. The regression of the DOU on water depth is only significant in the LSC category, but not in the HSC (Fig. 4). Therefore, the bacterial benthic remineralisation, which makes up ~ 80 % of the TOU, depends on water depth in low sea-ice covered areas, but not in the highly sea-ice covered EG area. Bacterial density, however, did not show differences between the HSC and LSC categories and therefore was not the biotic link which connected the food input pattern (Supplement Fig. S3) with the remineralization pattern (Figure 4). Benthic bacterial biomasses and benthic bacterial community structures, factors which may explain the differences in the benthic mineralization patterns of high and low sea-ice covered areas, have been to date only investigated in the eastern Fram Strait (Jacob et al., 2013) but not in the western Fram Strait. A test, if this remineralization pattern is also true for the macro- and meiofauna remineralisation, represented by the fauna mediated oxygen uptake (= TOU minus DOU), was determined to be not reliable in the current study given a low reproducibility of TOU values.

The results of the PCA were used to display an ecosystem snapshot. However, the included factors likely respond on different time scales. For example, benthic faunal biomass, density, and structure will respond to food-related parameters in a more seasonally to decadal fashion, while benthic remineralisation responds on short time scales such as days to weeks (Graf, 1989; Renaud et al., 2008). To acknowledge this, we decided to use the short-term dataset 'year before sampling' in the PCA. Additionally, the origin of the primary production responsible for the benthic food supply is difficult to assess and can be located >3000 km from Fram Strait (Lalande et al., 2016). In turn, the complexity of advective and vertical pelagic food input influencing processes in the Fram Strait is not considered in the ecosystem snapshot. Furthermore, by comparing

only two sites (HSC/LSC) a statistical investigation of the actual relationship between ice cover and the response variables is not possible.

To summarise, sea-ice cover in the Fram Strait is a proxy for light availability and nutrient supply and therefore is also indicative of primary production in Fram Strait. In addition, water depth increase represents a progressive degradation state of settling organic material towards the sea floor (Belcher et al., 2016). Both processes are responsible for determining the food supply to the benthos. Therefore, the independent factors of 'sea-ice cover' and 'water depth' were the most important abiotic factors in the Fram Strait as they were the primary factors in controlling the benthic food supply. This fits earlier observations, that labile organic matter is the most important factor determining Arctic deep-sea benthic communities (Grebmeier et al., 1988; Boetius and Damm; 1998; Klages et al., 2004). Regarding benthic remineralisation, the Fram Strait is bisectional: water depth independent in the highly sea-ice covered western Fram Strait and water depth dependent in the low sea-ice covered eastern Fram Strait. However, the impact of sea-ice on the benthic remineralisation cannot be distinguished from the impact of water depth in water depth >1500 m.

## 4.2 Primary production and benthic remineralisation in the Fram Strait

The reported oxygen fluxes within the HSC and LSC categories are comparable to earlier findings within the Fram Strait (Sauter et al., 2001; Cathalot et al., 2015) and the continental margin of the Laptev Sea (Boetius and Damm, 1998), but are slightly lower than modelled results for the pan-Arctic region (Bourgeois et al., 2017; Fig. 6). In general, the total benthic carbon remineralisation across the entire Fram Strait is on average ~1 mmol C $m^{-2}d^{-1}$.

The new primary production, the part of total production which can fuel the benthos (Platt et al., 1989), is estimated at 55 g C $m^{-2}yr^{-1}$ (Sakshaug, 2004, and references therein) in the West Spitsbergen area. This is equal to 38 mmol C $m^{-2}d^{-1}$, assuming a production period of 120 days (Gradinger, 2009). Codispoti et al. (2013) reported net community production from nutrient depletion for the WS area of 27–32 g C $m^{-2}$. These values reflect the annual new production and thus can be converted to 19–22 mmol C $m^{-2}d^{-1}$ (under the same assumption of 120 days of production). This indicates that on average approximately 2.6–5.2 % of the new primary production in the WS area may be remineralised by the benthos. Lalande et al. (2016) reported from sediment trap studies that 2.7 g C $m^{-2}yr^{-1}$ (= 1.9 mmol C $m^{-2}d^{-1}$ under the same assumption of 120 days of production, particle trap study at HG IV) and therefore 5–14 % of the primary production reaches the seafloor. Taking these export fluxes into account, this indicates that only 40 % of the organic material reaching the seafloor is remineralised by the benthos in the West Spitzbergen area in the eastern Fram Strait.

The net primary production in the mainly sea-ice covered western Fram Strait is approximately 8 g C $m^{-2}yr^{-1}$ (Codispoti et al., 2013), which represents 5.6 mmol C $m^{-2}d^{-1}$ (under the same assumption of 120 days of production). This is comparable to the similarly sea-ice covered central Arctic Ocean (Codispoti et al., 2013; Fernández-Méndez et al. 2015), Thus, 18 % of the new primary production in the EG area could be remineralised by the benthos. Annual POC flux values of 1–2.7 g C $m^{-2}yr^{-1}$ (= 0.7–1.9 mmol C $m^{-2}d^{-1}$, under the same assumption of 120 days of production) were reported for the ice-

covered regions at the Greenland shelf at 80 °N (Bauerfeind et al., 1997) and 1.6 g C m$^{-2}$yr$^{-1}$ (= 1.1 mmol C m$^{-2}$d$^{-1}$, under the same assumption of 120 days of production) at the Greenland shelf at 74 °N (Bauerfeind et al., 2005). These values indicate that 13–34 % of the primary production reaches the seafloor, which is comparable to Arctic shallow shelf regions (Grebmeier et al., 1988; Renaud et al., 2007). It further suggests that 50 % to >100 % of the organic material that reaches the

seafloor is remineralised by the benthic organisms at the East Greenland continental margin and that this area must, therefore, be supplied by organic carbon from other areas.

It should be noted that the numbers presented here must be interpreted with caution, as the reliable calculation of the primary production across the entire Fram Strait still remains problematic. Satellite-based chlorophyll measurements are only available in ice-free areas for periods with no clouds or fog (Cherkasheva et al., 2014). Additionally, satellites only

measure chlorophyll a in the upper water column. Therefore, to calculate the total primary production, additional information on the mixed water depth, photosynthetically active radiation, water temperature, salinity, nutrient availability, chlorophyll a to carbon ratio, growth rates of the different occurring algae (Sakshaug, 2004) and further parameters needs to be measured during the bloom period, which can be exclusively obtained by ship-based expeditions. The approach of Codespoti (2013) is preferable, under which primary production and benthic remineralisation are compared. However, this approach requires a

good spatial resolution of nutrient profiles in the water column. Furthermore, measurements of the benthic oxygen flux, crucial to evaluate the pelagic-benthic-coupling, remain only snapshots of remineralisation activity and also have some uncertainties (TOU: ~5 %, DOU: ~ 1 %) due to methodological error propagation. The question, if the Arctic deep-sea benthic oxygen fluxes follow seasonal changes, has only been sparsely partially addressed to date (Bourgeois et al., 2017). However, a pulsed supply of food and thus pulsed temporal response of the benthic community has been observed in other

deep-sea communities (e.g. Witte et al., 2008; Smith et al., 2016). A full annual cycle of benthic remineralisation in the Arctic is still missing and as such, a more reliable discussion of the pelagic-benthic-coupling and the carbon cycle remains problematic.

### 4.3 A future deep-sea benthic Arctic Ocean scenario

Our results indicate that a development from a permanently sea-ice covered to a seasonally sea-ice covered Arctic Ocean

will change the bentho–pelagic relationship from a sea-ice dependent towards a water depth dependent environment (Fig. 4). This shift will likely occur in parallel to the predicted compositional shift in the spring phytoplankton bloom from diatom dominated to coccolithophorid (Bauerfeind et al., 2009) or *Phaeocystis* sp. and nanoflagellates dominated bloom (Soltwedel et al., 2015). Altering the algal composition of the upper waters will affect zooplankton communities (Caron and Hutchins, 2013) and organic particle fluxes (Wohlers et al., 2009). An additional predicted effect of a progressive sea-ice cover

reduction is an increasing annual matter flux towards the seafloor (Wassmann, 2011; this study). Results regarding the vertical flux of labile material are, however, contradictory (Hop et al., 2006; van Oevelen et al., 2011; Boetius et al., 2013). Consequently, the change in sea-ice cover in the Arctic Ocean may alter the quality and quantity of the organic matter flux to the seafloor, where it may influence benthic deep-sea communities, in both biomass and species mix (Jones et al., 2014;

Harada, 2015).The comparable DOU of the EG and HG site at water depth >1500 m (Fig. 4) indicates, however, that the remineralisation by the deep-sea benthos will possibly remain stable in the Arctic Ocean.

Our scenario is only suitable if sea-ice disappears and nutrient supply increases, which will result in enhanced primary production. The development of future Arctic Ocean primary production patterns and changes is still under debate (Wassmann, 2011; Arrigo et al., 2012; Nicolaus et al., 2012; Boetius et al., 2013). However, it is likely that the described scenario will occur in the Chukchi Sea and the Beaufort Sea, owing to the predicted strengthening of the nutrient rich Pacific inflow (Harada, 2015). Furthermore, owing to an increased Atlantification, an increased nutrient supply is also likely for the continental margin at the Barents Sea (Neukermans et al., 2018). In addition, nutrient inflow by glacial and permafrost soil melt is also predicted to increase (Vonk et al., 2015). However, this riverine load may only enhance primary production on an in the vicinity of the shelf areas, and therefore have a negligible influence on the deep sea. An enhanced primary production in the western Fram Strait is unlikely even if light availability increases, as the required associate nutrient supply increase is not predicted for this region (Mauritzen et al., 2011). Additionally, the sea ice in the Fram Strait is already thinning (Krumpen et al., 2015). This may be led to more light in the upper water column and an already higher primary production in the EG area, which consequently may have resulted in a higher food supply to the deep-sea benthos in this area and thereby biases our former-Arctic-Ocean perspective. However, fast sinking algae patches as reported by Boetius et al (2013) in the central Arctic, which would lead to increased benthic remineralisation, were not observed during a video transect at EG IV in 2014 (pers. Comm. J. Taylor). A further limitation of our scenario might be that in contrast to the HG stations, there are no long-term data available on the benthic environment at the EG stations. Thus, an assessment of ongoing changes in the EG area of comparable type to those made at the HG stations (Soltwedel et al., 2015), and therefore conclusions on the natural variability of benthic conditions in that area, remains difficult at the moment. Nevertheless, the general sea-ice concentration pattern in Fram Strait has been stable over the last 14 years (Fig. 2). This indicates that at least the primary production periods and therefore periods of the low food supply were also stable at the EG stations throughout the last 14 years. In addition, the scenario is only valid for areas changing from permanent to very low sea-ice cover as our data does not allow conclusions to be made on the likely impact of a scenario where a high sea-ice cover is replaced with an intermediate (20–60 %) sea-ice cover.

Despite these uncertainties, observations are currently still the best method for creating and evaluating scenarios of future ecosystem developments, as consistent time series data from the entire Arctic Ocean, required to model reliable future predictions with confidence, are to date not available (Wassmann et al., 2011). Thus, our comparative study provides new insights into the relationship between sea–ice cover at the surface and benthic oxygen fluxes in the Fram Strait via surface primary production, benthic food supply, benthic community and their functions. We hypothesise that if surface primary and secondary production will increase due to the retreating sea–ice cover, that the deep-sea benthos of the Arctic Ocean may shift from a sea-ice dependent state towards a water depth dependent state in response to the changing environment. There may be a slightly increased food supply and an altered macrofauna community, but remineralisation rates at water depths greater than 1500 m seem unlikely to be affected by these changes as the process is primarily food limited.

**Figures**

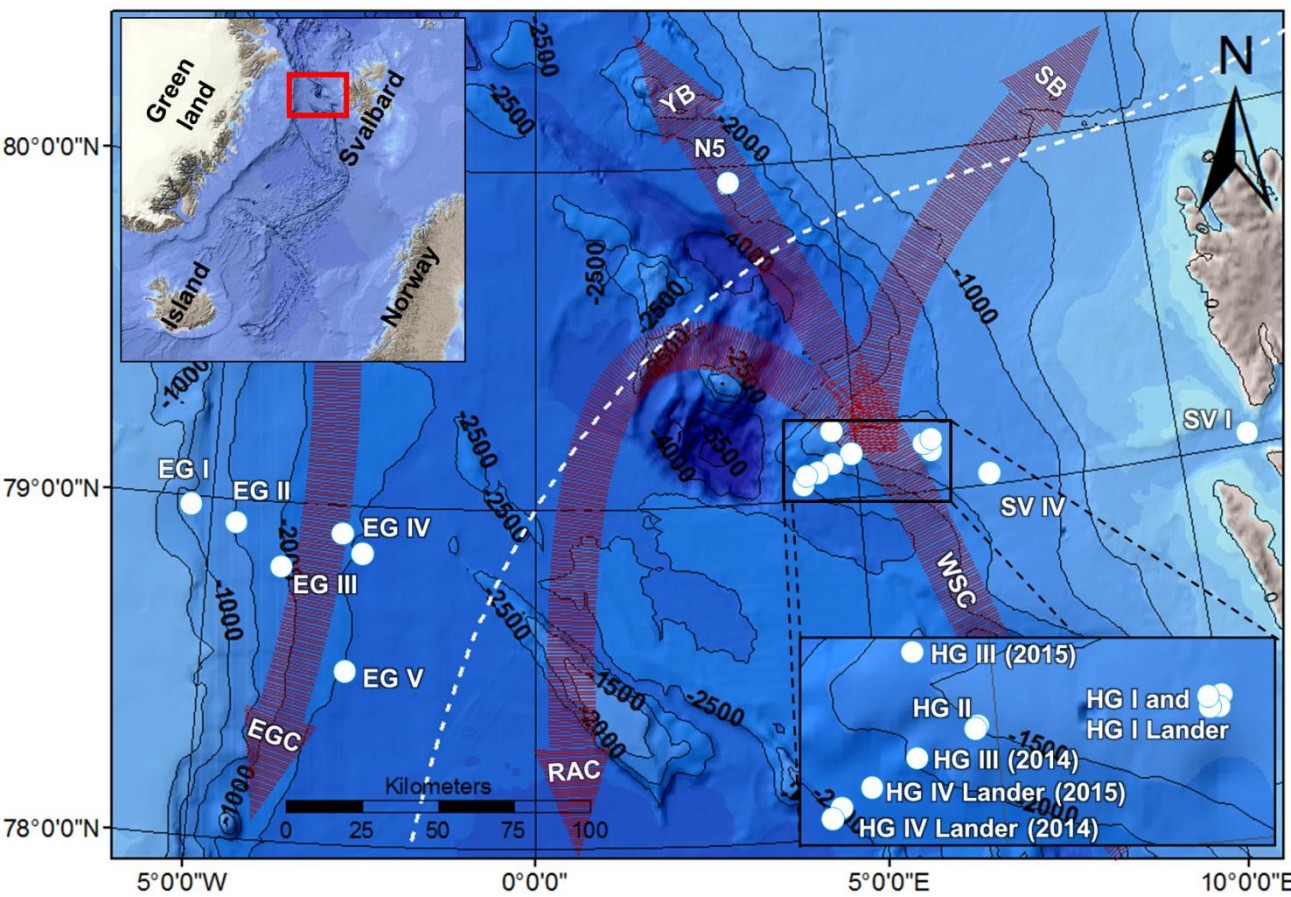

**Figure 1.** Locations of the sampled stations in the Arctic Fram Strait. White dashed line = mean summer sea-ice extent in September (1981-2010, (http://nsidc.org)). Red arrows = general current system. EGC = East Greenland Current, WSC = West Spitsbergen Current, SB = Svalbard branch, YB = Yermak branch, RAC = Return Atlantic current. White dots = stations with station names. More station-specific details are given in Table 1.

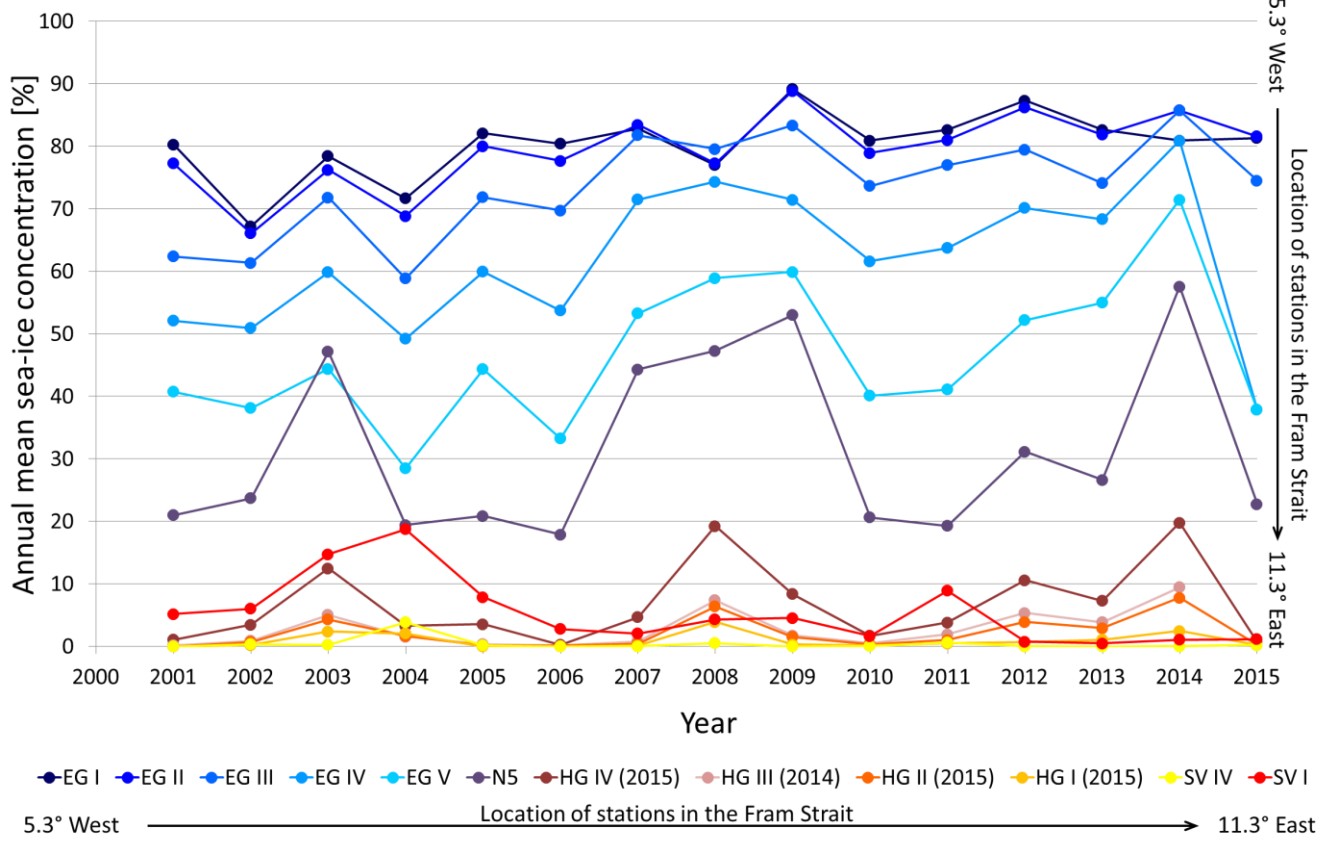

**Figure 2.** Annual mean sea-ice concentrations from 2001 to 2015 of a subset of sampled stations. The sampling year at the HG stations (central HAUSGARTEN stations, Soltwedel et al., 2005) is given, as HG stations where sampled in 2014 and 2015 and therefore, the given sampling year refers to the exact position from which the sea ice data were obtained.

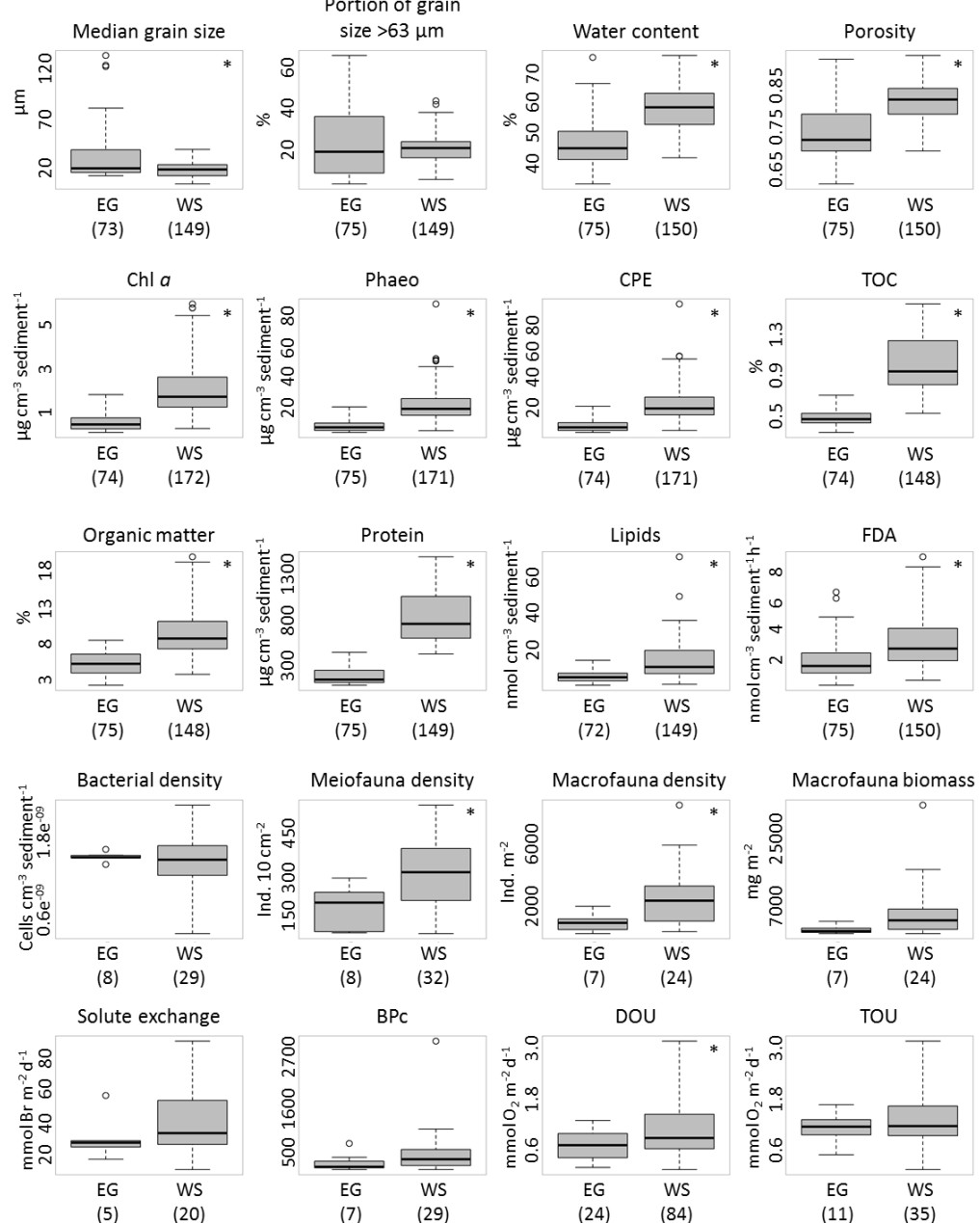

**Figure 3.** Boxplots of sediment properties, biogenic compound values (Chl *a* = chlorophyll a, Phaeo = phaeophytin, CPE = chloroplastic pigment equivalents, TOC = total organic carbon, FDA = bacterial enzymatic turnover rates calculated using the fluorogenic substrate fluorescein-di-acetate), benthic community data and function (BPc = bioturbation potential), and oxygen fluxes (DOU = diffusive oxygen uptake, TOU = total oxygen uptake), of the East Greenland (EG) and West Spitsbergen (WS) area. For a detailed description of which stations were included at which site, see section 2.1. The number of observations is given in brackets below the area. Parameters showing significant differences between areas are marked with an asterisk. For comparability, the WS site does not contain values from SV I station.

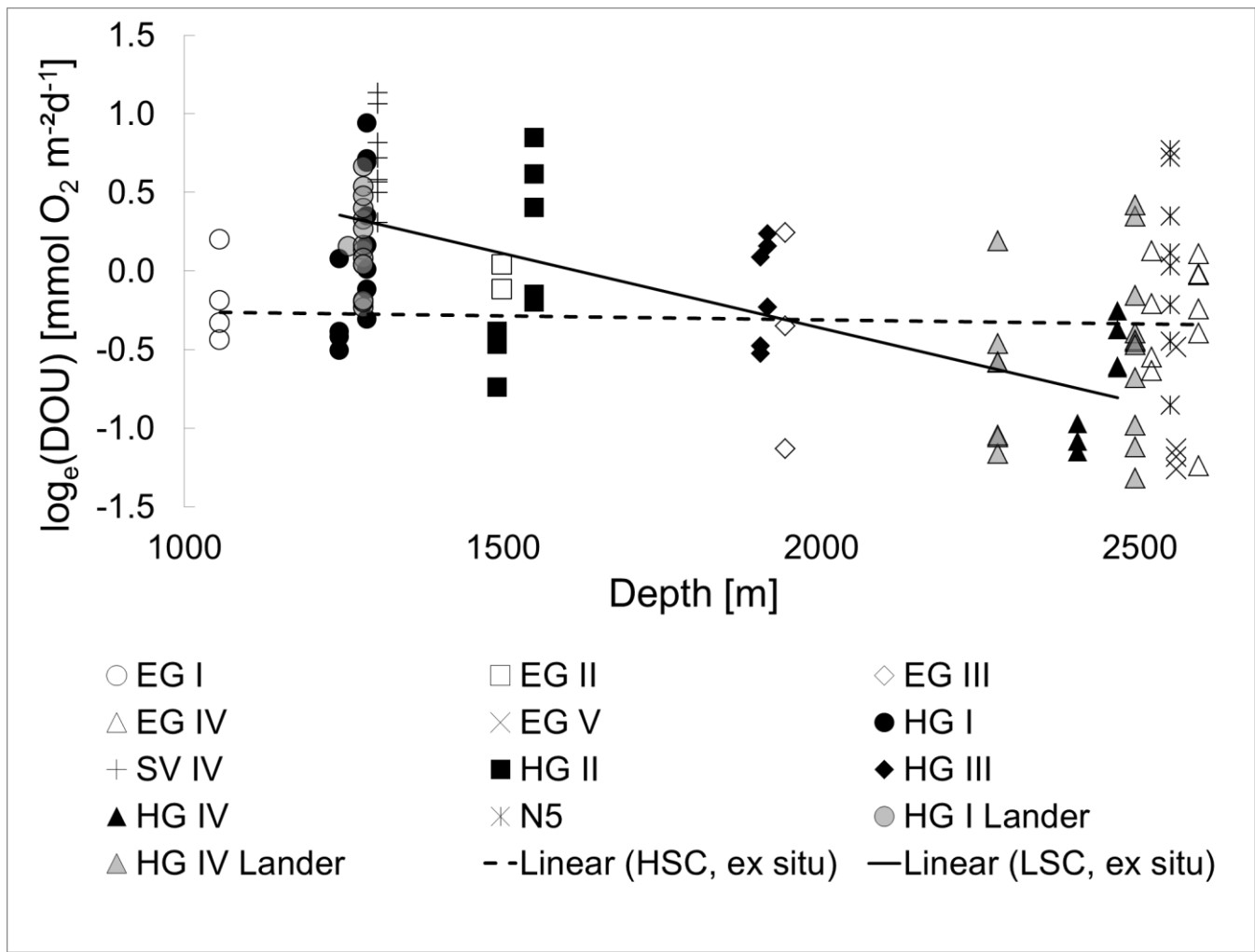

**Figure 4.** Station specific diffusive oxygen uptake data (DOU, log-transformed) at investigated stations and as linear function of water depth (from ex situ values) in the area of high sea-ice concentration (HSC) and low sea-ice concentration (LSC) in Fram Strait. The full line indicates a significant decrease of DOU with water depth in the LSC area, while the dashed line indicates that the slope did not differ significantly from zero in the HSC area.

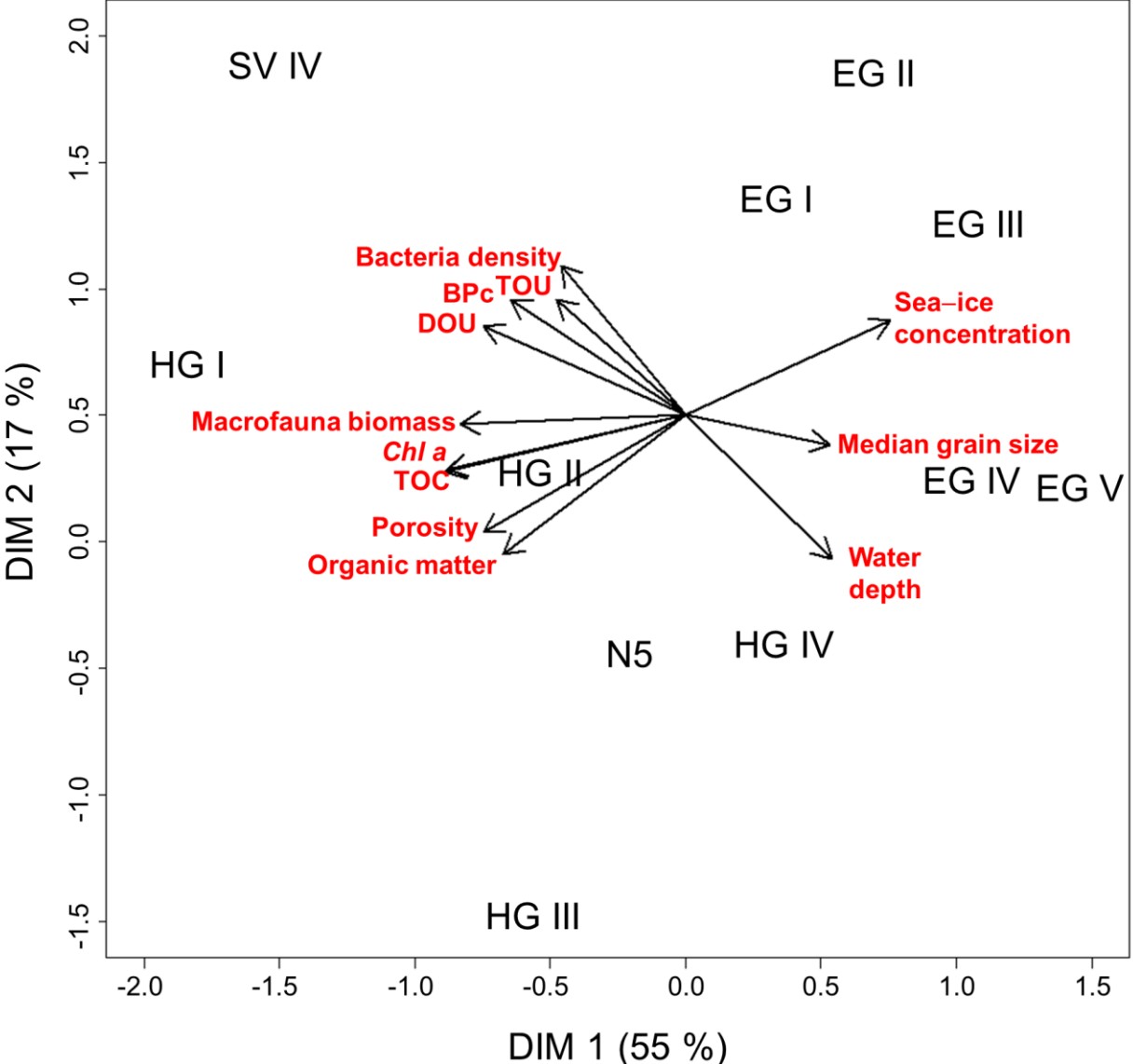

**Figure 5.** Visualisation of PCA results on standardised ex situ mean values of abiotic parameters (water depth, sea-ice concentration, median grain size, porosity), biogenic compound parameters (chlorophyll a (Chl *a*), total organic carbon (TOC), organic matter), benthic community parameters (bacterial density, macrofauna biomass), bioturbation potential (BPc), and diffusive and total oxygen uptake (DOU, TOU). All other parameters were excluded from the PCA as they correlated strongly with one of the mentioned parameters (correlation >0.74, Pearson correlation, Supplement Table S2). For comparability, Station SV I was excluded from the PCA. Therefore, the figure reflects relations of different parameters in Fram Strait in water depths of 1000–2500 m.

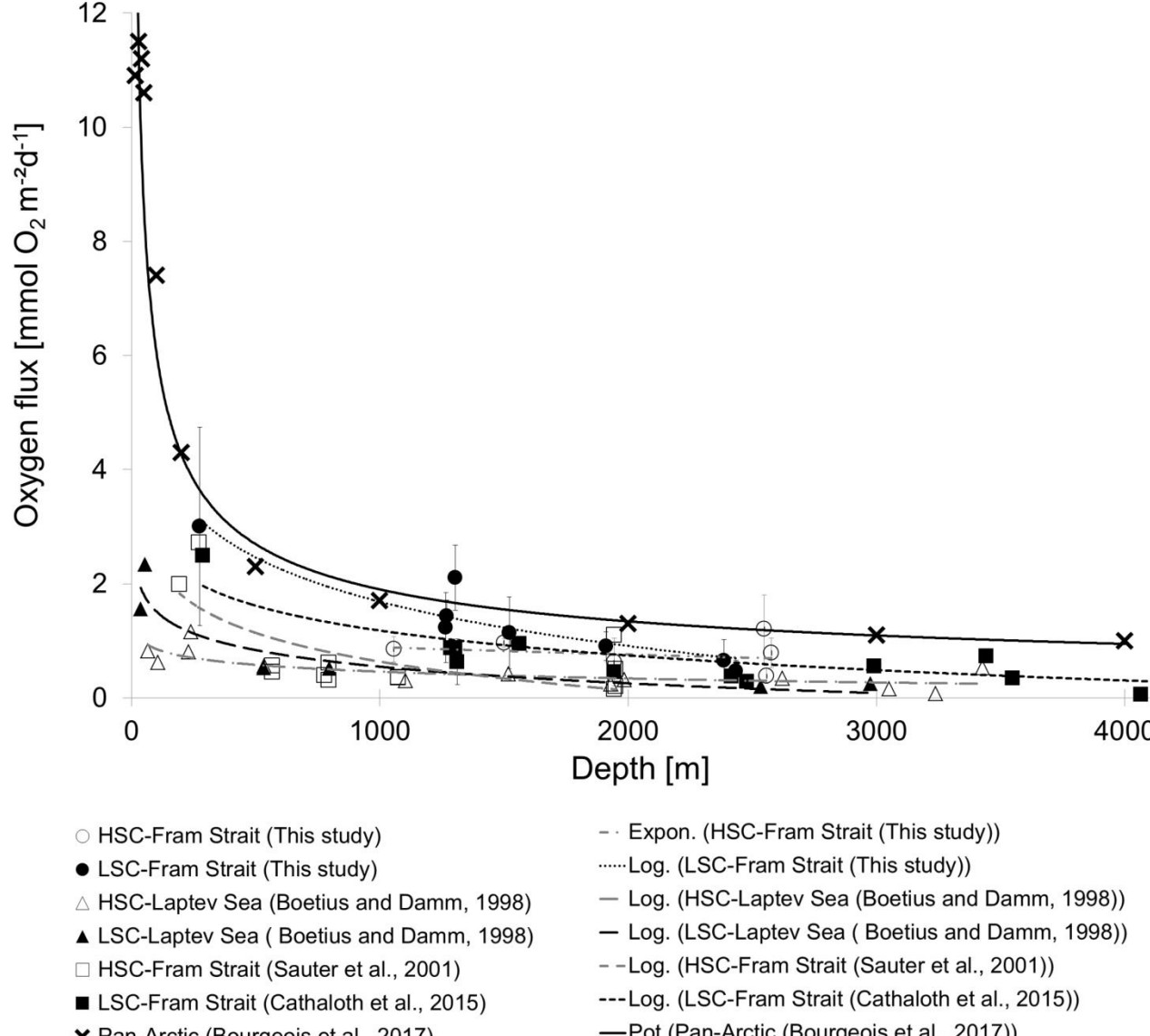

**Figure 6.** Sediment oxygen uptakes in different water depths (15 – 4000 m) in areas with high sea-ice concentration (HSC, ≥30 % sea-ice concentration) and low sea-ice concentration (LSC, <30 % sea-ice concentration) from this study and from literature data for the Laptev Sea, Fram Strait, and Pan-Arctic region and related regressions. HSC regression from this study: y=-0.124ln(x)+1.7388 ($R^2$=0.0255); LSC regression from this study: y=-1.119ln(x)+9.4144 ($R^2$=0.8695); HSC regression from Sauter et al. (2001): y=-0.727ln(x)+5.6587 ($R^2$=0.5026); LSC regression from Cathalot et al. (2015): y=-0.63ln(x)+5.534 ($R^2$=0.7013); HSC regression from Boetius and Damm (1998): y=-0.172ln(x)+1.6496 ($R^2$=0.6074); LSC regression from Boetius and Damm (1998): y=-0.421ln(x)+3.4515 ($R^2$=0.8428); Pan-Arctic regression from Bourgeois et al (2017): y=7.1338e$^{-6E-04x}$ ($R^2$=0.7288). Regression types were chosen based on best fit ($R^2$). The model of Bourgeois et al. (2017) included diffusive oxygen fluxes and total oxygen fluxes, while all other references refer only to diffusive oxygen fluxes.

**Data availability**

Data are available via the PANGAEA data archive. https://www.pangaea.de/

**Authors contribution**

RH, UB, and FW designed this study; RH and UB were responsible for the measurements and calculations of the oxygen fluxes. Additionally, UB was responsible for the measurement of the grain size parameters and bromide concentrations, and pre-processing and identification of the meio- and macrofauna densities and macrofauna biomasses. CH was responsible for

5    the measurements of the water content and the biogenic compounds. RH was responsible for the calculation of the porosity, the solute exchange, and the BPc and performed all statistical analyses within "R". CH was responsible for the statistical analyses in PRIMER. RH prepared the manuscript with contributions from all co-authors.

**Competing interests**

The authors declare that they have no conflict of interest.

**Acknowledgement**

We are very grateful to Paul Renaud and two anonymous reviewers who considerably helped us to improve this manuscript. We further like to acknowledge the help and effort made by the officers and crew members of RV *Polarstern* during our expeditions to the Fram Strait. In addition, we like to thank Volker Asendorf and Axel Nordhausen for preparing the Lander

5   and executing the deployment and recovery during the expeditions, Cäcilia Wiegand and Ines Schröder for the production of the Clark-typ oxygen sensors, Bart Beuselinck for granulometric analyses, Anja Pappert, Svenja Schütte and Meike Spill for biogenic compound and bacterial density analyses, Naomi De Roeck, Thomas Luypaert, Emiel Platjouw, Ellen Pape and Katja Guilini for processing of macro- and meiofauna samples, Thomas Krumpen for providing the additional sea-ice concentrations at station EG V,Pier Buttigieg and Christiane Hassenrück for support regarding statistical analyses in R, and

10   Autun Purser for editing support. This research was funded by the European Union FP7 Project SenseOCEAN Marine Sensors for the 21st century (Grant Agreement Number 614141), by the Flemish Research Fund, by institutional funds of the Alfred -Wegener- Institut Helmholtz-Zentrum für Polar-und Meeresforschung, and by the infrastructure framework of the FRAM project (Frontiers in Arctic Marine Monitoring).

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

# Tables

**Table 1.** General station information regarding water depth, sampling date, location and station ID in the data archive Pangaea. Order of stations for each area follows the water depth gradient.

| Area | Station name | Water depth (m) | Sampling date | Latitude (dd.ddd °N) | Longitude [dd.ddd °E] | Pangaea Station ID |
|------|--------------|-----------------|---------------|----------------------|------------------------|---------------------|
| EG | EG I | 1056 | 17th June 2014 | 78.973 | -05.290 | PS85/0436-1 |
|  | EG II | 1500 | 18th June 2014 | 78.933 | -04.650 | PS85/0441-1 |
|  | EG III | 1944 | 19th June 2014 | 78.803 | -03.875 | PS85/0445-1 |
|  | EG IV | 2592 | 31st July 2015 | 78.862 | -02.710 | PS93/0058-12 |
|  |  | 2519 |  | 78.914 | -02.961 | PS93/0058-17 |
|  | EG V | 2558 | 20th June 2014 | 78.505 | -02.817 | PS85/0454-3 |
| WS | SV I | 275 | 6th August 2015 | 79.028 | 11.087 | PS93/0066-2 |
|  | HG I | 1244 | 24th June 2014 | 79.133 | 06.1065 | PS85/0470-3 |
|  |  | 1288 | 10th August 2015 | 79.138 | 06.0835 | PS93/0080-9 |
|  | HG I Lander | 1258 | 26th June 2014 | 79.142 | 06.124 | PS85/0476-1 |
|  |  | 1282 | 10th August 2015 | 79.134 | 06.092 | PS93/0080-8 |
|  | SV IV | 1304 | 8th August 2015 | 79.029 | 06.999 | PS93/0074-3 |
|  | HG II | 1492 | 24th June 2014 | 79.132 | 04.906 | PS85/0469-2 |
|  |  | 1550 | 9th August 2015 | 79.130 | 04.902 | PS93/0078-2 |
|  | HG III | 1905 | 24th June 2014 | 79.106 | 04.585 | PS85/0468-1 |
|  |  | 1916 | 8th August 2015 | 79.208 | 04.600 | PS93/0077-2 |
|  | HG IV | 2403 | 22nd June 2014 | 79.065 | 04.183 | PS85/0460-4 |
|  |  | 2465 | 27th July 2015 | 79.065 | 04.179 | PS93/0050-19 |
|  | HG IV Lander | 2493 | 24th June 2014 | 79.052 | 04.138 | PS85/0466-1 |
|  |  | 2278 | 27th July 2015 | 79.083 | 04.337 | PS93/0050-18 |
|  | N5 | 2548 | 3rd August 2015 | 79.938 | 03.193 | PS93/0060-10 |

**Table 2.** Sea-ice cover (%) and % of days with sea-ice cover on different time scales across the Fram Strait. The values are given in mean values ± standard deviation and number of samples in brackets. Sea-ice data a year before sampling are mean values for the period 01.07.2013–30.06.2014 for stations only sampled in 2014 and 01.08.2014–31.07.2015 for stations only sampled in 2015. For stations sampled in both years, data of both periods were combined. The date of sampling is given in Table 1.

| Station name | | EG I | EG II | EG III | EG IV | EG V | SV I | HG I | HG I Lander | SV IV | HG II | HG III | HG IV | HG IV Lander | N5 |
|---|---|---|---|---|---|---|---|---|---|---|---|---|---|---|---|
| Water depth (m) | | 1056 | 1500 | 1944 | 2556 | 2558 | 275 | 1266 | 1270 | 1304 | 1521 | 1911 | 2434 | 2386 | 2548 |
| 2001–2015 | Sea-ice coverage (%) | 80 ± 24 (5101) | 79 ± 24 (5101) | 74 ± 26 (5102) | 64 ± 31 (5102) | 47 ± 33 (5102) | 5 ± 16 (5102) | 1 ± 6 (5101) | 1 ± 6 (5101) | 0 ± 4 (5101) | 2 ± 3 (5101) | 4 ± 13 (5102) | 7 ± 18 (5102) | 6 ± 17 (5102) | 32 ± 33 (5102) |
| | Days with sea-ice coverage within a year before sampling (%) | 97 | 97 | 95 | 92 | 84 | 15 | 4 | 4 | 2 | 8 | 11 | 18 | 17 | 66 |
| Year before sampling | Sea-ice coverage [%] | 82 ± 20 (364) | 80 ± 21 (364) | 75 ± 27 (364) | 72 ± 24 (365) | 56 ± 34 (364) | 1 ± 5 (365) | 1 ± 7 (729) | 1 ± 7 (729) | 0.1 ± 2 (365) | 4 ± 12 (729) | 5 ± 14 (729) | 10 ± 21 (729) | 9 ± 19 (729) | 40 ± 31 (365) |
| | Days with sea-ice coverage within a year before sampling (%) | 100 | 100 | 100 | 98 | 93 | 6 | 4 | 4 | 0 | 13 | 16 | 25 | 24 | 82 |
| Since 01.05. till sampling | Sea-ice coverage [%] | 79 ± 15 (48) | 77 ± 16 (49) | 81 ± 13 (50) | 56 ± 24 (90) | 67 ± 22 (51) | 1 ± 4 (96) | 5 ± 9 (77) | 15 ± 9 (78) | 0 (98) | 10 ± 12 (77) | 10 ± 12 (77) | 19 ± 15 (71) | 19 ± 16 (71) | 35 ± 23 (94) |
| | Days with sea-ice coverage within a year before sampling (%) | 100 | 100 | 100 | 99 | 100 | 10 | 18 | 17 | 0 | 32 | 34 | 45 | 45 | 89 |
| Month before sampling | Sea-ice coverage [%] | 76 ± 16 (31) | 73 ± 17 (31) | 84 ± 10 (31) | 57 ± 22 (31) | 74 ± 19 (31) | 1 ± 4 (30) | 9 ± 11 (31) | 8 ± 11 (31) | 0 (30) | 16 ± 12 (31) | 15 ± 12 (31) | 22 ± 17 (31) | 23 ± 18 (31) | 25 ± 21 (31) |
| | Days with sea-ice coverage within a year before sampling (%) | 100 | 100 | 100 | 100 | 100 | 7 | 28 | 24 | 0 | 41 | 41 | 52 | 52 | 74 |

**Table 3.** Mean values ± standard deviation and number of samples in brackets for each measured parameter at each station. Chl *a* and Phaeo abbreviates chlorophyll a and phaeophytin concentrations, respectively. The CPE is the chloroplastic pigment equivalent and the sum of Chl *a* and Phaeo. Chl *a*–CPE ratio indicates the available labile carbon source, while the Chl *a*–Phaeo ratio indicates the relative age of the carbon source. TOC is the percentage of total organic carbon, while FDA is the bacterial enzymatic turnover rate. The BPc value represents the bioturbation potential of the macrofauna community. DOU and TOU are the diffusive and the total oxygen uptake, respectively, while C-DOU and C-TOU are the related diffusive and total carbon fluxes. No value could be calculated for solute exchange across the sea-water-interface at EG II. For station abbreviations the reader is referred to paragraph 2.1.

| Parameter category | Parameter | Station | | | | | | | | | | | | | |
|---|---|---|---|---|---|---|---|---|---|---|---|---|---|---|---|
| | | EG I | EG II | EG III | EG IV | EG V | SV I | HG I | HG I Lander | SV IV | HG II | HG III | HG IV | HG IV Lander | N5 |
| | Water depth (m) | 1056 | 1500 | 1944 | 2556 | 2558 | 275 | 1266 | 1270 | 1304 | 1521 | 1911 | 2434 | 2386 | 2548 |
| Sediment property | Median grain size (µm) | 13.4 ± 1.2 (15) | 15.1 ± 1.7 (15) | 20.3 ±3.9 (15) | 31.6 ± 7.3 (15) | 74.2 ± 29.3 (13) | 12.3 ± 2.7 (15) | 12.7 ± 6.0 (30) | NA | 20.4 ± 6.4 (15) | 12.7 ± 5.8 (30) | 19.3 ± 5.3 (29) | 23.8 ± 5.3 (30) | NA | 10.4 ± 2.9 (15) |
| | Portion of grain size >63 µm (%) | 3.5 ± 1.5 (15) | 8.6 ± 2.9 (15) | 18.6 ± 6.0 (15) | 29.5 ± 6.8 (15) | 52.2 ± 6.7 (15) | 17.7 ± 2.2 (15) | 11.4 ± 5.7 (30) | NA | 24.4 ± 5.6 (15) | 12.6 ± 6.0 (30) | 20.1 ± 4.2 (29) | 24.5 ± 5.3 (30) | NA | 20.7 ± 2.6 (15) |
| | Water content (%) | 51 ± 7 (15) | 48 ± 7 (15) | 46 ± 9 (15) | 48 ± 10 (15) | 42 ± 6 (15) | 51 ± 14 (15) | 66 ± 5 (30) | NA | 55 ± 5 (15) | 62 ± 4 (30) | 55 ± 5 (30) | 51 ± 8 (30) | NA | 60 ± 5 (15) |
| | Porosity | 0.76 ± 0.06 (15) | 0.73 ± 0.06 (15) | 0.71 ± 0.08 (15) | 0.73 ± 0.08 (15) | 0.68 ± 0.06 (15) | 0.75 ± 0.14 (15) | 0.88 ± 0.03 (30) | NA | 0.80 ± 0.04 (15) | 0.85 ± 0.03 (30) | 0.80 ± 0.04 (30) | 0.77 ± 0.06 (30) | NA | 0.84 ± 0.03 (15) |
| Food availability | Chl a (µg ml$^{-1}$ sediment$^{-1}$) | 0.5 ± 0.4 (15) | 0.5 ± 0.3 (15) | 0.4 ± 0.3 (15) | 0.6 ± 0.5 (15) | 0.63 ± 0.4 (14) | 12.7 ± 3 (15) | 3 ± 1 (29) | 2.5 ± 1.5 (10) | 2.2 ± 1.1 (14) | 2.0 ± 1.2 (30) | 2.1 ± 0.8 (30) | 1.3 ± 0.6 (30) | 1.1 ± 0.6 (15) | 1.2 ± 0.4 (14) |
| | Phaeo (µg ml$^{-1}$ sediment$^{-1}$) | 4.2 ± 2.2 (15) | 3.7 ± 2.4 (15) | 3.0 ± 2.2 (15) | 7.2 ± 5.6 (15) | 6.7 ± 3.6 (15) | 67.3 ± 10.8 (15) | 30.9 ± 8.8 (30) | 16.4 ± 8.6 (10) | 24.4 ± 20 (14) | 18.5 ± 8.6 (29) | 20.0 ± 6.2 (30) | 12.4 ± 6.2 (30) | 9.8 ± 5.7 (15) | 14.8 ± 3.8 (13) |
| | CPE (µg ml$^{-1}$ sediment$^{-1}$) | 4.7 ± 2.6 (15) | 4.2 ± 2.7 (15) | 3.4 ± 2.5 (15) | 7.8 ± 6.1 (15) | 7.4 ± 4.2 (15) | 80.0 ± 13.1 (15) | 34.0 ± 9.7 (30) | 18.9 ± 9.9 (10) | 26.7 ± 21.3 (14) | 20.4 ± 9.6 (29) | 22.1 ± 6.7 (30) | 13.6 ± 6.7 (30) | 10.7 ± 6.2 (15) | 16.0 ± 4.2 (13) |
| | Chl *a*–CPE ratio | 0.10 ± 0.02 (15) | 0.11 ± 0.02 (15) | 0.10 ± 0.02 (15) | 0.08 ± 0.01 (14) | 0.09 ± 0.02 (15) | 0.16 ± 0.02 (15) | 0.09 ± 0.02 (30) | 0.13 ± 0.03 (10) | 0.09 ± 0.01 (15) | 0.09 ± 0.02 (30) | 0.10 ± 0.02 (30) | 0.10 ± 0.02 (30) | 0.10 ± 0.02 (15) | 0.07 ± 0.02 (15) |
| | Chl *a*–Phaeo ratio | 0.11 ± 0.03 (15) | 0.13 ± 0.02 (15) | 0.11 ± 0.03 (15) | 0.08 ± 0.02 (14) | 0.10 ± 0.02 (15) | 0.19 ± 0.03 (15) | 0.10 ± 0.03 (30) | 0.16 ± 0.04 (10) | 0.10 ± 0.02 (15) | 0.10 ± 0.02 (30) | 0.11 ± 0.03 (30) | 0.11 ± 0.02 (30) | 0.12 ± 0.03 (15) | 0.08 ± 0.02 (15) |
| Other biogenic compounds | TOC (%) | 0.55 ± 0.05 (14) | 0.44 ± 0.04 (15) | 0.45 ± 0.04 (15) | 0.51 ± 0.11 (15) | 0.53 ± 0.09 (15) | 1.58 ± 0.27 (15) | 1.37 ± 0.08 (28) | NA | 0.98 ± 0.13 (15) | 1.05 ± 0.19 (30) | 0.92 ± 0.11 (30) | 0.69 ± 0.07 (30) | NA | 0.88 ± 0.03 (15) |
| | Organic matter (%) | 7.1 ± 1.0 (15) | 3.5 ± 0.6 (15) | 3.5 ± 0.6 (15) | 6.6 ± 0.7 (15) | 5.0 ± 0.9 (15) | 8.0 ± 2.2 (15) | 9.1 ± 2.9 (30) | NA | 8.0 ± 1.0 (15) | 10.6 ± 1.3 (29) | 11.4 ± 3.8 (28) | 6.5 ± 0.9 (29) | NA | 8.4 ± 0.4 (15) |
| | Proteins (µg ml$^{-1}$ sediment$^{-1}$) | 100 ± 20 (15) | 122 ± 22 (15) | 120 ± 22 (15) | 337 ± 80 (15) | 259 ± 43 (15) | 3253 ± 475 (15) | 998 ± 314 (30) | NA | 686 ± 85 (14) | 1053 ± 95 (30) | 1004 ± 313 (30) | 530 ± 64 (30) | NA | 748 ± 76 (15) |

| | | | | | | | | | | | | | | | |
|---|---|---|---|---|---|---|---|---|---|---|---|---|---|---|---|
| | Lipids (nmol ml$^{-1}$ sediment$^{-1}$) | 2.9 ± 1.1 (15) | 5.2 ± 2.1 (14) | 4.2 ± 2.4 (14) | 5.3 ± 2.2 (14) | 8.4 ± 2.8 (15) | 49.7 ± 21.0 (15) | 10.4 ± 7.1 (30) | NA | 22.3 ± 10.9 (15) | 16.4 ± 8.5 (30) | 13.7 ± 5.5 (29) | 16.6 ± 16.3 (30) | NA | 8.5 ± 3.4 (15) |
| | FDA (nmol ml$^{-1}$ sediment$^{-1}$ h$^{-1}$) | 1.9 ± 0.7 (15) | 1.1 ± 0.8 (15) | 1.3 ± 0.8 (15) | 2.6 ± 2.1 (15) | 2.1 ± 1.1 (15) | 31.3 ± 12.2 (15) | 4.7 ± 1.5 (30) | NA | 1.7 ± 0.6 (15) | 3.3 ± 2.1 (30) | 3.0 ± 1.3 (30) | 2.8 ± 1.7 (30) | NA | 2.2 ± 0.6 (15) |
| Benthic community | Bacteria density (Cells 10$^9$ ml$^{-1}$ sediment) | 1.60 (1) | 1.57 (1) | 1.55 (1) | 1.57 ± 0.09 (4) | 1.56 (1) | NA | 1.79 ± 0.13 (4) | 1.14 ± 0.20 (3) | 1.83 ± 0.43 (3) | 1.81 ± 0.08 (4) | 1.29 ± 0.19 (4) | 1.49 ± 0.07 (4) | 9.28 ± 0.35 (4) | 1.54 ± 0.04 (3) |
| | Meiofauna density (ind. 10cm$^{-2}$) | 229 (1) | 83 (1) | 86 (1) | 192 ± 79 (4) | 245 (1) | 1150 ± 159 (3) | 333 ± 134 (3) | 357 ± 151 (5) | 402 ± 123 (3) | 277 ± 75 (4) | 273 ± 83 (4) | 352 ± 141 (4) | 293 ± 202 (6) | 268 ± 98 (3) |
| | Macrofauna biomass (mg m$^{-2}$) | 3524 (1) | 1971 (1) | 1301 (1) | 433 ± 287 (3) | 450 (1) | 45370 ± 25609 (3) | 12196 ± 13652 (4) | 6929 (1) | 8733 ± 1671 (3) | 1325 ± 479 (4) | 6186 ± 6137 (4) | 2784 ± 1578 (4) | 836 (1) | 8166 ± 7364 (3) |
| | Macrofauna density (ind. m$^{-2}$) | 1414 (1) | 991 (1) | 284 (1) | 1058 ± 722 (3) | 1064 (1) | 4945 ± 6286 (3) | 2860 ± 1206 (4) | 942 (1) | 4143 ± 2817 (3) | 2471 ± 612 (4) | 4343 ± 2818 (4) | 1148 ± 542 (4) | 417 (1) | 2023 ± 409 (3) |
| Community functions | Solute exchange (mmol Br m$^{-2}$d$^{-1}$) | 29.3 (1) | NA | 57.7 (1) | 17.7 (1) | 28.1 (1) | 38.8 ± 1.8 (3) | 51.3 ± 14.1 (4) | NA | 39.9 ± 6.3 (3) | 38.9 ± 13.0 (5) | 53.2 ± 27.3 (3) | 50.8 ± 39.3 (2) | NA | 15.0 ± 3.1 (3) |
| | BPc | 644 (1) | 318 (1) | 93 (1) | 55 ± 25 (3) | 132 (1) | 1586 ± 1042 (3) | 556 ± 266 (4) | 397 (1) | 909 ± 852 (8) | 199 ± 51 (4) | 391 ± 90 (4) | 74 ± 40 (4) | 70 (1) | 106 ± 39 (3) |
| Oxygen flux | DOU (mmol O$_2$ m$^{-2}$d$^{-1}$) | 0.9 ± 0.2 (4) | 1.0 ± 0.1 (2) | 0.6 ± 0.4 (4) | 0.8 ± 0.3 (4) | 0.4 ± 0.1 (10) | 3.0 ± 1.7 (6) | 1.2 ± 0.6 (12) | 1.2 ± 0.3 (15) | 2.1 ± 0.6 (8) | 1.1 ± 0.6 (8) | 0.9 ± 0.3 (7) | 0.5 ± 0.2 (8) | 0.7 ± 0.4 (18) | 1.2 ± 0.6 (8) |
| | TOU (mmol O$_2$ m$^{-2}$d$^{-1}$) | 0.9 ± 0.3 (2) | 1.6 (1) | 1.5 ± 0.1 (2) | 1.1 ± 0.1 (2) | 1.0 ± 0.2 (4) | 5.1 ± 0.2 (3) | 1.9 ± 0.6 (5) | 1.3 ± 0.2 (4) | 1.8 ± 0.2 (3) | 1.1 ± 0.2 (5) | 1.0 ± 0.2 (5) | 1.5 ± 0.5 (5) | 0.5 ± 0.2 (5) | 1.2 ± 0.3 (3) |
| | DOU/TOU | 1.00 | 0.63 | 0.40 | 0.73 | 0.40 | 0.59 | 0.63 | 0.92 | 1.17 | 1.00 | 0.90 | 0.33 | 1.40 | 1.00 |
| Carbon flux equivalent | C-DOU (mmol C m$^{-2}$d$^{-1}$) | 0.7 ± 0.2 (4) | 0.7 ± 0.2 (2) | 0.5 ± 0.3 (4) | 0.6 ± 0.2 (10) | 0.4 ± 0.1 (4) | 2.3 ± 1.3 (6) | 1.0 ± 0.5 (12) | 0.9 ± 0.3 (15) | 1.6 ± 0.4 (8) | 1.1 ± 0.7 (8) | 0.7 ± 0.2 (7) | 0.4 ± 0.1 (8) | 0.5 ± 0.3 (18) | 0.9 ± 0.5 (8) |
| | C-TOU (mmol C m$^{-2}$d$^{-1}$) | 0.7 ± 0.3 (2) | 1.3 (1) | 1.1 ± 0.1 (2) | 0.8 ± 0.1 (4) | 1.0 ± 0.2 (2) | 3.9 ± 0.2 (3) | 1.5 ± 0.5 (5) | 1.0 ± 0.1 (4) | 1.4 ± 0.2 (3) | 0.8 ± 0.2 (5) | 0.8 ± 0.2 (5) | 1.1 ± 0.4 (5) | 0.4 ± 0.2 (5) | 1.0 ± 0.2 (3) |

**Table 4.** ANOSIM and SIMPER results of the meio- and macrofauna community within sea-ice categories. The table shows that there are differences in the macrofauna community between the area of high sea-ice concentration (HSC) and low sea-ice concentration (LSC), while this is not the case for the meiofauna community. The most contributing taxa regarding the in-group similarity within the sea-ice categories and the dissimilarity between the sea-ice categories are given in Supplement Table 8.

| | | Meiofauna density | | Macrofauna density | | Macrofauna biomass | |
|---|---|---|---|---|---|---|---|
| **ANOSIM** | Global R | 0.143 | | 0.266 | | 0.227 | |
| | p-value | 0.036 | | 0.005 | | 0.004 | |
| **SIMPER** | | HSC | LSC | HSC | LSC | HSC | LSC |
| | | In−group similarity | In−group similarity | In−group similarity | In−group similarity | In−group similarity | In−group similarity |
| | | 66.0 % | 72.5 % | 35.4 % | 56.1 % | 27.4 % | 32.0 % |
| | | Dissimilarity between groups | | Dissimilarity between groups | | Dissimilarity between groups | |
| | | 32.1 % | | 55.9 % | | 75.3 % | |

