# Peer review of "Deep-sea benthic communities and oxygen fluxes in the Arctic Fram Strait controlled by sea-ice cover and water depth"

_Biogeosciences, 2017_

## Referee Comment (RC1) · P. Renaud (Referee) · 13 Feb 2018

P. Renaud (Referee)

paul.renaud@akvaplan.niva.no

Review of Hoffmann et al. 'Deep-sea benthic communities and oxygen fluxes'. . . (Paul Renaud, reviewer)

General comments

The authors have collected a vast amount of data from regions that are not easily accessible over two years and across a range of depths in the most important gateway to and from the Arctic Ocean. Investigations of how variability in environmental and especially benthic-process data vary spatially here are carried out with a focus on the

potential impacts of sea-ice cover and water depth. The study is valuable just in terms of the data it provides in this important but understudied region. In addition, the authors provide a largely balanced discussion of many of the critical points here.

I am concerned about several issues that are not brought into the discussion (time scales of response and which variables may have meaningful causative links) or are brought perhaps too forward (sea ice cover, which is in my mind less of a driver than the water mass properties responsible for both the ice cover itself but also the productivity regime and vertical flux). Finally, I urge more caution in development of future scenarios without a better understanding of whether nutrient loads can sustain increased production in the currently ice-covered areas.

Specific comments

1. Time scales of response. Direct (and only linear) correlations between environmental parameters and O2 flux may be misleading, or non-representative, depending on when the samples are taken (and when relative to the bloom/flux phenologies among different stations). Benthic biomass/density/structure likely respond to various factors (especially food-related parameters which are often covariates of depth) in a more seasonally (or up to decadally) integrated fashion, whereas O2 consumption/C remineralization are often more responsive to food inputs on a much shorter time scale (approx. weeks) (e.g. Renaud et al. 2008 DSR II). This must be considered in your interpretation.

2. As noted in the Methods, the difference between the eastern and western Fram Strait and potential consequences for benthic processes go beyond ice cover. Advected POC/PON/dissolved nutrients and warmer temperatures on the eastern side are far greater than on the western side. Of course that is linked to why the ice is there, but in this case ice is more of a covariate and perhaps less likely a causative factors. In addition, different zooplankton and microbial communities can well lead to different 'food' deposition. This must be considered in detail if the two transects are to

be comparable.

3. Methods: it appears that most of the variables measured were only assessed from the top 1 cm of sediment. Can you provide a justification (data-based) for this? For meiofauna it is often the top 2-3 cm that contain the majority of the fauna, and for macrofauna, at least the top 5 cm, even at deep-sea depths.

4. Ice cover in the two 'regions' is essentially 70-80% vs 1-10% (heavy ice/no ice). Except for EG V (and N5 which is often excluded from analysis) there is nothing in between. How might this affect your results/interpretation? Many of the results from N5 are more similar to the LSC than the HSC stations (see comment 2 above).

5. Ice cover as the key? factor. Related to comment 2 above, have you evaluated whether correlations/differences between benthic parameters and ice cover are the strongest relationships among your data? Primary productivity, vertical flux attenuation, and essentially food supply to the sediment surface may or may not be caused by sea ice in any way. Or it could be a feature of Arctic vs Atlantic water supply that causes a 'cascade of processes' and sea ice cover may just be a covariate with limited or even no direct causative effect (hence a logic problem on p 15 l 23-25). Your discussion implies that ice is the overriding factor but I do not see where you tested for this, or if it is even possible to disentangle all these variables to isolate depth as the key factor. If you ran similar analyses but grouped stations based on water mass characteristics instead of ice cover you would find the same result.

6. P 12 l 6: unclear what water column nutrients, presented as a snapshot without context of 'preformed' (winter) concentrations add here. Bloom phenology certainly is responsible for e.g. the lower nitrate in WS vs EG. Consider removing these data. The discussion on p 15-16 and then sec 4.2 is not really based on the data collected, but more of a general pattern documented in the literature. I agree some of this should be included, but wouldn't a more extended and balanced discussion of benthic process rates and the other factors (proximal) responsible for variation in these rates be

appropriate here?

7. Nutrient supply under the ice in EG is extremely low and not expected to increase with further melting of sea ice (e.g. Mauritzen et al. 2011 Prog Oceanogr). This casts serious doubt into any scenario where increased PP due to more light is invoked.

Technical corrections/details 1. P4 l6-9: unclear sentence. Perhaps just unnecessary (same for l 14-17 as it just repeats what you have just written) 2. P4, l 24: controlling the benthic ecosystem? Be more specific, including what you mean by 'labile organic matter' (different from benthic chlorophyll?) 3. P6 l 10: if the algorithm can estimate ice cover at over 100% then couldn't values between 0 and 100 also be mismeasurements? Could there be some (automated) check to assure that adjacent pixels are 'similar', or some other way of testing for mismeasurement in this range? 4. Also, is an annual average (vs some other ice cover parameter) the most relevant measure of ice cover? 5. P 6 l 25: frozen not frosted 6. P 8 l 25ff: Was non-local mixing (i.e. non-linear profiles) observed? How was this accounted for in the O2 flux calculation? 7. P 9 l 1-5: how much of the sediment mass could be attributed to salt from the drying process? 8. P 10 l 5: consecutive not subsequent 9. P 10 l 14: 'x to zero mean and unit variance' is unclear 10. P 10 l 21: you must exclude EG II from the analysis. You cannot make the assumption and assign a value. It was fine to exclude the shallow station, and you should do the same with EG II 11. P 10 l 30: you need to indicate whether there was a different depth relationship between the two regions and then say what you did if this was (or was not) the case. 12. P 12 l 6: unclear what water column nutrients, presented as a snapshot without context of 'preformed' (winter) concentrations add here. Bloom phenology certainly is responsible for e.g. the lower nitrate in WS vs EG. Consider removing these data. The discussion on p 15-16 is not really based on the data collected, but more of a general pattern documented in the literature. 13. P 12 l 10ff (and Fig 3). Please indicate any statistical results such that the figures correspond to what is written in the text regarding comparisons between the two regions. Only statistically significant results should be expressed as 'differences' (e.g. solute exchange

is likely NS but significance is implied. Also, please clarify how many stations (and depth profile) each bar represents. This has some bearing on your comments about variability between the two locations. 14. P 12 l 18: 'pelagic food supply indicating parameter in the sediment' rephrase to clarify that these are sediment values. . ..and careful about how you define food quality. Not all organisms each chlorophyll (in fact maybe few actually do). Bacteria themselves are likely food for many organisms, and phaeopigments and other OC may also be quite high quality food for others. 15. P13 l 21: 'which indicates bacterial activity and bacterial remineralisation as the major oxygen consumer' please indicate why you conclude this. Why would bacterial oxygen consumption not be reflected in DOU data. These are effectively two different techniques to measure the same thing, each with underlying assumptions. The conclusion you make regarding the ratio is not supported. 16. P 14 l 7-13: I would focus on the differences among EG and WS stations as revealed by PCA, and not individual variable correlations (which are NOT real correlations but instead are ordination-based relationships! If you want to look for correlation then run that analysis on the raw data). 17. P 16 l 17: but macrofauna biomass has a similar relationship with depth between the two ice-cover systems 18. P 16 l 18-20. Repetitive 19. P 16 l 27: a CCA or RDA would find significant relationships. . . 20. Sec 4.3: first paragraph unnecessary. 21. P 19 l 4: neither citation is in the references. Are you sure the Kortsch ref is appropriate?

---

## Referee Comment (RC2) · Anonymous Referee #2 · 10 Mar 2018

Hoffmann et al. Review, Feb 2018 The authors present an impressive data set from the western and eastern sides of Fram Strait, the Atlantic gateway with the only deep-water inflow into and outflow out of the Arctic Ocean. Deep-sea biological communities and processes are inherently difficult to sample and measure, respectively. The presented data set - comprising both - is indeed a treasure and one of many valuable results of the long-term HAUSGARTEN observatory. There are actually so many variables measured that the reader is a bit challenged to follow the story at times. The article undoubtedly merits publication, but requires some adjustments. I suggest the following: 1. The intro-duction will benefit from turning the lists of which factors depend on which other factors into a narrative explaining how they influence each other. This change would neces-

sarily make the introduction a bit longer, but improve the logic, flow and justification for the study. Also, the authors would help the reader by providing a bit of background why they estimate remineralization of new production rather than calculating it because they can. 2. The discussion (especially 4.1 and 4.2) repeats the results to a large extent. Instead, it should place the results in the context of the extensive literature from the area and beyond. I recommend the authors summarize their findings more concisely and discuss their results in the context of, for example, the pan-Arctic scale Progress in Oceanography issue from 2015, primary productions model estimates covering the area, the series of three articles from Patrai – Codispoti 2013 etc. 3. Water depth and vertical flux are well-documented highly influential factors structuring benthic communities both in terms of biodiversity and biomass/abundance anywhere in the ocean, in addition to sea ice cover. While these factors are mentioned in the discussion (without much literature support actually), it should also be noted more prominently that eastern Fram Strait receives constant inflow of particle rich Atlantic water, and this advective input adds to the vertical flux (see for example Wassmann et al. 2015 PiO for a summary). It is indeed complex to separate out the effects of water mass properties including particle content, and ice cover – a fact that should be acknowledged. 4. The authors said they struggled to find some relevant information (e.g. on primary production) for the western Fram Strait side, and therefore used values from the central Arctic. They might consider the results of the SFB313 that spent years investigating East Greenland including the slope, including carbon remineralization, primary production, benthic community structure etc., http://www.springer.com/us/book/9783540672319. Was the region never covered in any of the primary production models? Some additional useful information from eastern Fram Strait is also available, e.g. Wlodarska-K. et al. 2004 in DSRII. Small corrections / comments: P4 l6 I would not call primary production and oxygen flux an ecosystem component, they are rate measurements of processes. The benthic community is an ecosystem component. P4 Delete l6-8 (redundant to previous sentence). L4 l9 Rather 'nutrient concentrations' (or which property of nutrients?) P4l10 If this is to be general across the globe, add 'In general, benthic community ...'

P4l14 and elsewhere. I was taught 'therefore' never starts a sentence. P4l20 'Western' Arctic is a rather undefined term, since different nations use it in very different ways, rather give the region. P4l21-22 Unclear how the 'better fit' works when one doesn't know what other factors were included. P4l34 No need to repeat the three references for the same aspect since already given in l28 P5l17ff What time period is considered when talking about stable ice cover here? What time period is considered in the number of 0.6 years per decade? (And somewhere in the discussion the author talk about ice thinning, a bit of a contradiction.) P6l4 Why combine sea ice cover and nutrients under one sub-header? I suggest separating those sections. P5l6 rather 'Study area and field sampling' or 'Study area and sample collection'. None of the sample preparation or processing is described here. P6l9 Although both 'data are' and 'data is' is allowed per some dictionaries, it really should be 'data are' (one datum, several data). P6l15 Provide a reference for the nutrient measurement method. P6l20 Which property of phospholipids and proteins and organic matter was measured – presumably concentrations? P6 section 2.3 The methods description is extremely abbreviated, but it is an editor decision if this is sufficient. P7 2.4 What taxonomic resolution was aimed for? P12l6 It would be appropriate to include the nutrient profiles (at least upper water column) into the MS figures rather than the supplement given that the nutrient inventories provide the basis to the level of primary production possible (although measured after the bloom was done presumably). At the very least some concentration ranges should be mentioned. Define 'surface'. P12l16 Why 'indicates'? Later you test this! P12l12 There are different opinions on this, but given that I would find at least a range of densities etc. presented (as is done in the next section 3.4). At the very least, table 2 should be referenced here so that the reader can find the results. P13l31-32 add 'rather than an actual interannual difference' P14l5 Just above you wrote the different is likely related to the months, while this line states it is a spatial difference. Both may be true, but as written the statements seem contradictory. P14l26 Significant indeed, but the authors should mention that the global R values are rather low, same with the macrofauna results. P15 l10 perhaps add ' marginally not significant' P15l16 As phrased, this

is not a question. P15l23 Grammar. If there were a strong link ... we would expect .... (conditional) P16l14/15 This is not the right place to mention this point, move to figure caption or results text. P16l27 opposite to our expectations or in contrast to our expectations. The following PCA sentence is grammatically incorrect. The PCA only shows .. but does not test ... P19l4 The Kortsch paper is on shallow nearshore hard bottom communities, not quite the right reference here. P19l10 In earlier sections the authors talk about 'stable ice conditions' in Fram Strait, while here they state that ice is thinning. Specifying by which metric the conditions are stable will relieve the contradiction. P19l14 My understanding of the Boetius et al. paper is that these authors discussed the high Melosira biomass to be generated on the shelf and maintained (but not produced) over the basin through constrant resupply of –albeit low – nutrients during ice drift, not as a consequence of increasing algal biomass in the central Arctic. General: Someone should switch German to English comma rules throughout. Table 2. Use same number of decimals within one parameter (e.g. days with sea ice has between zero and two decimals). Table 3. Spell out HSC and LSC. Figure 1. Specify time frame for 'general summer sea ice extent', by month and period. Figure 3. Indicate if any of the differences between EG and WS were statistically significant. This and other figures explain abbreviations or say in caption where they are explained. Figure 6. Typos: Arctic missing 'c'. Sauter et al. and Bourgeois et al. missing periods after al.

---

## Author Comment (AC1) · 16 Apr 2018

**Reply to reviewer Paul Renaud**

We would like to thank Paul Renaud for the constructive comments on our manuscript 'Deep-sea benthic communities and oxygen fluxes in the Arctic Fram Strait controlled by sea-ice cover and water depth'.
We will first address the reviewer's 'specific comments' and secondly reply to 'technical corrections/detail' with stating the planned improvements.

In the following, author responses starts with the term 'Reply' and changes, that will be included in the manuscript, are given in blue.

Specific comments:
1. Time scales of response. Direct (and only linear) correlations between environmental parameters and O2 flux may be misleading, or non-representative, depending on when the samples are taken (and when relative to the bloom/flux phenologies among different stations). Benthic biomass/density/structure likely respond to various factors (especially food-related parameters which are often covariates of depth) in a more seasonally (or up to decadally) integrated fashion, whereas O2 consumption/C remineralization are often more responsive to food inputs on a much shorter time scale (approx. weeks) (e.g. Renaud et al. 2008 DSR II). This must be considered in your interpretation.

Reply: We thank the reviewer for this important aspect. As spring bloom data in this region for the studied period (2014-2015) are not yet citable and the authors are no experts in satellite data acquisition and analyses, we will cite the results of Cherkasheva et al. 2014 (http://dx.doi.org/10.1016/j.jmarsys.2013.11.008) to provide information regarding the date of the spring bloom. Furthermore, we will acknowledge that correlations do not necessarily prove causal relationships and that oxygen flux measurements only represent a temporal snapshot. As we did not perform measurements during or right after the bloom deposition, we might have missed the immediate and short-term reaction of the benthos to the fresh organic matter. Nevertheless, we can expect a lower influence of macrofauna on the measured oxygen fluxes, compared to the findings in Renaud et al. 2008 (DSR II), owing to the high contribution of microbial benthic mineralization to the total benthic mineralization in the deep sea (Donis et al., 2016, Sauter et al. 2001, Wenzhöfer and Glud,2002), which is also expressed by the mean DOU/TOU ratio of 0.79 presented in our results.

2. As noted in the Methods, the difference between the eastern and western Fram Strait and potential consequences for benthic processes go beyond ice cover. Advected POC/PON/dissolved nutrients and warmer temperatures on the eastern side are far greater than on the western side. Of course that is linked to why the ice is there, but in this case, ice is more of a covariate and perhaps less likely a causative factors. In addition, different zooplankton and microbial communities can well lead to different 'food' deposition. This must be considered in detail if the two transects are to be comparable.

Reply: We agree with the reviewer and will discuss the complexity of advective and vertical pelagic food input influencing processes in Fram Strait in more detail. Furthermore, we will point out that we used the parameter 'sea-ice cover' as a proxy for primary production patterns. The sea-ice in the western Fram Strait represents a suppressed light availability and a reduced nutrient supply (owing to the main currents WSC and EGC). Both light availability and nutrient supply are the main drivers of primary production. This suits the findings of Pabi et al. (2008, doi:10.1029/2007JC004578), showing contrasting primary production quantities among the western and eastern Fram Strait.

3. Methods: it appears that most of the variables measured were only assessed from the top 1 cm of sediment. Can you provide a justification (data-based) for this? For meiofauna, it is often the top 2-3 cm that contains the majority of the fauna, and for macrofauna, at least the top 5 cm, even at deep-sea depths.

Reply: Indeed, microbial and meiofauna data were assessed from the top 1cm. Macrofauna data and the biogenic sediment compounds, however, were assessed from the top 5cm (MUC cores) and from even deeper sediments (benthic chamber sampled sediments). We will improve the method section to clarify this. Regarding the Meiofauna, we refer to Gorska et al. 2014 (http://dx.doi.org/10.1016/j.dsr.2014.05.010) and regarding microbial data, we refer to Quéric et al., 2004 (https://doi.org/10.1016/j.mimet.2004.02.005). Both studies show that most of the investigated organisms in the Hausgarten area occur in the top 1 cm.

4. Ice cover in the two 'regions' is essentially 70-80% vs 1-10% (heavy ice/no ice). Except for EG V (and N5 which is often excluded from analysis), there is nothing in between. How might this affect your results/interpretation? Many of the results from N5 are more similar to the LSC than the HSC stations (see comment 2 above).

Reply: We will acknowledge in our discussion, that comparing only two sites (heavy ice/no ice) does not allow us to estimate the actual relationship between ice cover and the response variables. We would like to point out that only station SV I station was often excluded from analyses, owing to its exceptional shallow water depth, compared to all other stations. The introduction of a third category (intermediate ice) would only be based on two stations (EG IV and N5). As these two stations are from the same water depth, they would not include the potential impact of water depth, which was identified as important and therefore would weaken the outcome of this approach.

5. Ice cover as the key factor. Related to comment 2 above, have you evaluated whether correlations/differences between benthic parameters and ice cover are the strongest relationships among your data? Primary productivity, vertical flux attenuation, and essentially food supply to the sediment surface may or may not be caused by sea ice in any way. Or it could be a feature of Arctic vs Atlantic water supply that causes a 'cascade of processes' and sea ice cover may just be a covariate with limited or even no direct causative effect (hence a logic problem on p 15 l 23-25). Your discussion implies that ice is the overriding factor but I do not see where you tested for this, or if it is even possible to disentangle all these variables to isolate depth as the key factor. If you ran similar analyses but grouped stations based on water mass characteristics instead of ice cover you would find the same result.

Reply: We verified whether correlations/differences between benthic parameters and ice cover are the strongest relationships among your data by running the PCA. The eigenvalues indicated that 'TOC', 'Chl a' and 'Macrofauna biomass' were responsible for the gradient along the x-axis and 'water depth', organic matter' and 'sea-ice concentration' for the gradient along the y-axis. However, 'sea-ice' is a proxy for light availability and nutrient supply in Fram Strait and therefore represents primary production, whereas water depth represents pelagic mineralization and therefore the fade of organic matter in the pelagic zone. Both process are responsible for the final 'TOC' and 'Chl a' concentrations at the seafloor. To make this clearer, we will include the proxy characterization in the method section, add the eigenvalues of the PCA as to the manuscript and integrate our argumentation in the discussion.

6. P 12 l 6: unclear what water column nutrients, presented as a snapshot without context of 'preformed' (winter) concentrations add here. Bloom phenology certainly is responsible for e.g. the lower nitrate in WS vs EG. Consider removing these data. The discussion on p 15-16 and then sec 4.2 is not really based

on the data collected, but more of a general pattern documented in the literature. I agree some of this should be included, but wouldn't a more extended and balanced discussion of benthic process rates and the other factors (proximal) responsible for variation in these rates be appropriate here?

Reply: We agree with the reviewer and we will remove the nutrient data from the manuscript. Further, we will discuss benthic mineralization and the other proximal factors responsible for variation in these rates in more detail.

7. Nutrient supply under the ice in EG is extremely low and not expected to increase with further melting of sea ice (e.g. Mauritzen et al. 2011 Prog Oceanogr). This casts serious doubt into any scenario where increased PP due to more light is invoked.

Reply: We will rewrite the potential future scenario, include spatial limitations and will point out that this scenario only holds true for areas, where sea-ice disappears and nutrient supply will increase. It will be changed to 'Our scenario is only suitable if sea-ice disappears and nutrient supply increase, which will result in enhanced primary production The development of future Arctic Ocean primary production patterns and changes is still under debate (Wassmann, 2011, Arrigo et al., 2012; Nicolaus et al., 2012, Boetius et al., 2013). However, it is likely that the described scenario becomes true in the Chukchi Sea and the Beaufort Sea, owing to the predicted strengthening of the nutrient rich Pacific inflow (Harada, 2015). Furthermore, owing to an increased atlantification, an increased nutrient supply is also likely for the continental margin at the Barents Sea (Neukermans et al., 2018). In addition, nutrient inflow by glacial and permafrost soil melt is also predicted to increase (Vonk et al., 2015). However, this riverine load might only enhance primary production at the shelf areas and therefore is not relevant for the deep sea. An enhanced primary production in the western Fram Strait is unlikely even if the light availability will increase, as the required nutrient supply increase is not expected for this region (Mauritzen et al., 2011).'

Technical corrections/details
1. P4 l6-9: unclear sentence. Perhaps just unnecessary (same for l 14-17 as it just repeats what you have just written)

Reply: The sentence will be removed.

2. P4, l 24: controlling the benthic ecosystem? Be more specific, including what you mean by 'labile organic matter' (different from benthic chlorophyll?)

Reply: The sentence will be changed to "However, the principal factor controlling microbial activity was most likely the supply of labile organic matter such as CPE, proteins and dissolved free amino acids (Boetius and Damm 1998)."

3. P6 l 10: if the algorithm can estimate ice cover at over 100% then couldn't values between 0 and 100 also be mismeasurements? Could there be some (automated) check to assure that adjacent pixels are 'similar', or some other way of testing for mismeasurement in this range?

Reply: Whenever there was a mismeasurement, the algorithm output was "128". So it is not the case that a sea-ice concentration of 101% or 105% or 112% and so on, could be measured. Therefore, the algorithm does not estimate ice cover over 100%. The sentence will be rewritten to make this point that clearer. Furthermore, we will add information regarding the quantity of these mismeasurements.

4. Also, is an annual average (vs some other ice cover parameter) the most relevant measure of ice cover?

Reply: We will provide some alternative sea-ice concentration periods (mean of 1-month before sampling, mean since first of May (assumed spring bloom onset) till sampling) for the reader. However, as pointed out by the reviewer, "benthic biomass/density/structure likely responds to various factors (especially food-related parameters which are often covariates of depth) in a more seasonally integrated fashion". To acknowledge this, we used the annual sea-ice cover in the PCA.

5. P 6 l 25: frozen not frosted

Reply: We will follow the suggestion of the reviewer and change "frosted" to "frozen".

6. P 8 l 25ff: Was non-local mixing (i.e. non-linear profiles) observed? How was this accounted for in the O2 flux calculation?

Reply: Non-local mixing was observed in some cases and therefore the reported DOUs for those cases are underestimations. However, only eight out of 81 ex situ obtained oxygen microprofiles at various stations and in one out of 34 in situ obtained oxygen microprofiles showed signs of non-local mixing. This information will be added to the method section.

7. P 9 l 1-5: How much of the sediment mass could be attributed to salt from the drying process?

Reply: It was 4.5% ± 1.9 over all samples. We will add this information to the method section.

8. P 10 l 5: consecutive not subsequent

Reply: We will follow the reviewers' suggestions and change 'subsequent' to 'consecutive'

9. P 10 l 14: 'x to zero mean and unit variance' is unclear

Reply: In most applications of a PCA (e.g. as a factor analysis technique), variables are often measured in different units. For such data, the data must be standardized to zero mean and unit variance, a common standardization procedure. If this is not done, high values (e.g. macrofauna biomass with values of ten thousands of mg m$^{-2}$) will get a greater importance than low values (e.g. DOU with values of max. 2.1 mmol O2 m-$^2$d$^{-1}$). Similar terms used for this procedure are 'data normalization' or 'z-scoring'. Though, as we followed the suggestions provided by Buttigieg and Ramette (2014), we decided to follow their term of 'standardization' (https://mb3is.megx.net/gustame/indirect-gradient-analysis/pca).

10. P 10 l 21: you must exclude EG II from the analysis. You cannot make the assumption and assign a value. It was fine to exclude the shallow station, and you should do the same with EG II

Reply: We agree with the reviewers' comment that the assumption of a solute exchange value for EG II is not a valid approach to deal with data gaps. However, as the other parameters included in the PCA from EG II were actually measured, we rather prefer to perform the PCA without the parameters of 'solute exchange' from all stations. With this suggestion, EG II would still be part of the central analysis of the paper. Furthermore, as 'solute exchange' is well correlated with other parameters such as macrofauna biomass, it will still be represented in the PCA.

11. P 10 l 30: You need to indicate whether there was a different depth relationship between the two regions and then say what you did if this was (or was not) the case.

Reply: We will add the information, that the depth relationships in terms of the bottom slope were similar between the two regions to the method section ('study site'). Furthermore, we will add to the results section, that water content, phaeo, and CPE showed a similar water depth relationships within the HSC and LSC categories (Figure S4) compared to the water depth relationship of DOU (Figure 4). We will discuss that the microbial mineralization is the main driver of benthic deep-sea mineralization (see reply to reviewers' specific comment no°1). We will further add to the discussion that microbial density did not show differences between the HSC and LSC categories and therefore was not the biotic link which connected the food input pattern (Figure S4) with the mineralization pattern (Figure 4). There are no studies regarding benthic microbial biomasses or community structure across Fram Strait. Although we identified parameters well correlated with mineralization processes, our study analyses are not able to fully explain the contrasting mineralization pattern between the western and the eastern Fram Strait.

12. P 12 l 6: unclear what water column nutrients, presented as a snapshot without context of 'preformed' (winter) concentrations add here. Bloom phenology certainly is responsible for e.g. the lower nitrate in WS vs EG. Consider removing these data. The discussion on p 15-16 is not really based on the data collected, but more of a general pattern documented in the literature.

Reply: We will follow the reviewers' suggestion and remove the nutrient data from the manuscript.

13. P 12 l 10ff (and Fig 3). Please indicate any statistical results such that the figures correspond to what is written in the text regarding comparisons between the two regions. Only statistically significant results should be expressed as 'differences' (e.g. solute exchange is likely NS but significance is implied). Also, please clarify how many stations (and depth profile) each bar represents. This has some bearing on your comments about variability between the two locations.

Reply: We will improve the text and use the term 'differences' only in case of significant differences. Further, we will indicate significant differences between the stations in the figure and add the number of observations for each bar

As an example, the results of the sediment compounds will read than 'The mean DOU in the EG area ranged between $0.4 \pm 0.1$ mmol $O_2$ m$^{-2}$d$^{-1}$ (n=10) at EG V and $1.0 \pm 0.1$ mmol $O_2$ m$^{-2}$d$^{-1}$ (n=10) at EG II. In the WS area, DOUs at stations within the same water depth range as the EG stations ranged between $0.5 \pm 0.2$ mmol $O_2$ m$^{-2}$d$^{-1}$ (n=8) at HG IV and $2.1 \pm 0.6$ mmol $O_2$ m$^{-2}$d$^{-1}$ (n=8) at SV IV. At the shallow station SV I the DOU reached $3.0 \pm 1.7$ mmol $O_2$ m$^{-2}$d$^{-1}$ (n=6, Table 3). The mean TOU in the EG area ranged between $0.9 \pm 0.3$ mmol $O_2$ m$^{-2}$d$^{-1}$ (n=2) at EG I and $1.6$ mmol $O_2$ m$^{-2}$d$^{-1}$ (n=1) at EG II. Similar mean TOU values were measured in the WS area, at stations within the same water depth range as the EG stations. TOU values ranged between $0.5 \pm 0.2$ mmol $O_2$ m$^{-2}$d$^{-1}$ (n=5) at HG IV Lander and $1.9 \pm 0.6$ mmol $O_2$ m$^{-2}$d$^{-1}$ (n=5) at HG I. At the shallow SV I station TOU reached $5.1 \pm 0.3$ mmol $O_2$ m$^{-2}$d$^{-1}$ (n=3, Table 3). DOU differed significantly between the WS and EG area, while TOU was similar among the areas (Fig. 3, Supplement Table S4).

14. P 12 l 18: 'pelagic food supply indicating parameter in the sediment' rephrase to clarify that these are sediment values and careful about how you define food quality. Not all organisms eat chlorophyll (in fact

maybe few actually do). Bacteria themselves are likely food for many organisms, and phaeopigments and other OC may also be quite high quality food for others.

Reply: Following the suggestion of the anonymous second reviewer, we add ranges of values regarding benthic food supply representing parameters. See our reply to comment no°13.

15. P13 l 21: 'which indicates bacterial activity and bacterial remineralisation as the major oxygen consumer' please indicate why you conclude this. Why would bacterial oxygen consumption not be reflected in DOU data. These are effectively two different techniques to measure the same thing, each with underlying assumptions. The conclusion you make regarding the ratio is not supported.

Reply: We will rephrase the sentence to 'The mean DOU/TOU ratio, which describes the fraction of the total community mediated oxygen flux (TOU) covered by the microbial mediated oxygen flux (DOU, Glud, 2008,) across the entire Fram Strait was 0.79 ± 0.30, with 0.63 ± 0.22 in the EG area and 0.92 ± 0.30 in the WS area, indicating that the total oxygen uptake is mainly microbially mediated.'

16. P 14 l 7-13: I would focus on the differences among EG and WS stations as revealed by PCA, and not individual variable correlations (which are NOT real correlations but instead are ordination-based relationships! If you want to look for correlation then run that analysis on the raw data).

Reply: We will follow the suggestion of the reviewer and rewrite the paragraph to emphasize the differences among EG and WS stations. It will be changed to 'The stations of the WS and EG area both followed the water depth gradient and shallower stations showed the higher oxygen fluxes. However, stations of the EG area were strongly influenced by the sea-ice cover, contained less organic matter and Chl a, and macrofauna biomass, compared to the WS stations.'

However, the reason to perform the PCA was to reveal the relationships between the multiple parameters. Therefore, we performed a PCA in the scaling II mode, which emphasize the relationships between parameters (Buttigieg and Ramette, 2014, https://doi.org/10.1111/1574-6941.12437). We will add this information to the method section. A performance on raw data as suggested by the reviewer is, however, not recommendable due to the reasons presented in the comment no°9. The correlation of single parameters with each other was already given in Table S2 and Figure S3. In addition, a PCA is a procedure that transforms a number of (possibly) correlated variables into a (smaller) number of uncorrelated variables called principal components. Thus a PCA eliminates redundant information. As it also gives the contribution of the single parameter (=strength of influence) to each principle component, the parameters most likely control the investigated area can be identified (Boetius and Damm, 1998, doi: https://doi.org/10.1016/S0967-0637(97)00052-6).

17. P 16 l 17: but macrofauna biomass has a similar relationship with depth between the two ice-cover systems

Reply: We will rephrase the sentence to 'However, when taking both abiotic factors (sea ice and water depth) into account, the water depth-macrofauna density relationship differed between HSC and LSC (Supplement Fig. S4).'

18. P 16 l 18-20. Repetitive

Reply: We will follow the reviewers' suggestion and remove the repetitive content.

19. P 16 l 27: A CCA or RDA would find significant relationships.

Reply: We tested our data regarding the suitability of the usage of a CCA ('decorana'-command in R package vegan) with the outcome that our data showed linear correlations, so only a RDA would be a suitable approach. Usually, an RDA is used to correlate parameters of two information layers: environmental and biotic data, where the former influences the latter. However, from our point of view, we have three information layers: environmental, biotic, and flux data. It is common knowledge that environmental data influence biotic data and both are influencing the flux data. Therefore, we decided for an indirect ordination by a PCA as the direct approach of an RDA incorporates the measured parameters into the ordination, without considering the possible influence of other, unmeasured parameters.

20. Sec 4.3: first paragraph unnecessary.

Reply: We will follow the reviewers' suggestion and remove the first paragraph.

21. P 19 l 4: neither citation is in the references. Are you sure the Kortsch ref is appropriate?

Reply: The reference of Jones et al., 2014 was added to the references, whereas Kortsch et al., 2012 was removed as suggested by the reviewer and we added the reference of Harada (2015) instead.

---

## Author Comment (AC2) · 16 Apr 2018

**Reply to anonymous reviewer**

We would like to thank the anonymous reviewer for the constructive comments on our manuscript 'Deep-sea benthic communities and oxygen fluxes in the Arctic Fram Strait controlled by sea-ice cover and water depth'. Especially the recommended literature was of great help to improve our manuscript. We will first address the reviewer's 'specific comments' and secondly like to reply to 'Small corrections/comments' with stating the planned improvements.

In the following, author responses starts with the term 'Reply' and changes, that will be included in the manuscript, are given in blue.

Specific comments:
1. The introduction will benefit from turning the lists of which factors depend on which other factors into a narrative explaining how they influence each other. This change would necessarily make the introduction a bit longer, but improve the logic, flow and justification for the study. Also, the authors would help the reader by providing a bit of background why they estimate remineralization of new production rather than calculating it because they can.

Reply: We will follow the reviewers' suggestion and rewrite the introduction into a narrative explaining how factors influence each other by giving the information how they single factors are correlated which each other. Further, we will point out the link between the new production and the remineralization. It will be changed to 'Benthic deep-sea remineralisation depends on primary production and is as such closely linked with primary production patterns, known as pelagic–benthic coupling (Graf, 1989). The relationship, however, includes many and partly inter-dependent factors. Benthic deep-sea remineralisation is positively correlated with surface primary production (Graf et al., 1995; Wenzhöfer and Glud, 2002), which is on its turn controlled by light availability and nutrient supply (Kirk, 2011; Cherkasheva et al., 2014; Fernández-Méndez et al., 2015). Though, on an annual basis, only the new production leaves the euphotic zone (Platt et al., 1989), supplying the benthos with organic carbon. Benthic remineralisation is negatively correlated to water depth (Wenzhöfer and Glud, 2002), as it represents a loss of organic carbon by pelagic remineralisation (Rullkötter, 2006, Belcher et al., 2016) and thereby a loss of benthic food. After organic carbon reached the seafloor, it is ingested and remineralised by the benthic community. Benthic community parameters, e.g. biomass, density, structure, and functions of different fauna size classes, are controlled by food supply (and thus by primary production) and water depth (Piepenburg et al., 1997; Flach et al., 2002; Smith et al., 2008) but also by sediment properties (Wheatcroft, 1992; Vanreusel et al., 1995). Benthic remineralisation rates also depend on benthic community biomass (Glud et al., 1994). Furthermore, benthic remineralisation is enhanced if the benthic community intensifies oxygenation of the seafloor (Glud, 2008) and thus also depends on the benthic community structure. Therefore, the ecosystem processes primary production, pelagic remineralisation and benthic remineralisation, as well as the components benthic community biomass, density, and structure are controlled by abiotic and biotic factors and additionally create a cascade of dependencies from the ocean's surface zone of primary production to and within the deep-sea benthos.'

2. The discussion (especially 4.1 and 4.2) repeats the results to a large extent. Instead, it should place the results in the context of the extensive literature from the area and beyond. I recommend the authors summarize their findings more concisely and discuss their results in the context of, for example, the pan-Arctic scale Progress in Oceanography issue from 2015, primary productions model estimates covering the area, the series of three articles from Patrai – Codispoti 2013 etc.

Reply: Indeed, we repeated the results to a large extent, as we first needed to interpret our results before we could start to place the output in the context. However, we will reduce the repetition where ever possible and thereby follow the reviewer's suggestion. For example, the second paragraph of section 4.1 will be changed to 'The results of Pabi et al. (2008) showed that the annual primary production pattern follows the general sea-ice concentration pattern in the Fram Strait and is up to 10-times larger in the WS area compared to the EG area. Thus, the sea-ice concentration represents the general primary production pattern in the Fram Strait. As the sampling was performed in Mid/End of June 2014 and July/August 2015, it is very likely that the spring bloom, which usually starts in May (Cherkasheva et al., 2014), had finished. This is indicated by lower nutrient concentrations in water depth ≤50 m compared to the nutrient concentrations between >50–300 m water depths (Graeve and Ludwichowski, 2017a, b). The N:P ratio in the upper 50 m during the expeditions was six and seven in the EG and WG area, respectively (Graeve and Ludwichowski, 2017a, b), indicating that primary production was nitrate limited, similar to the permanently sea-ice covered central Arctic Ocean (Tremblay et al., 2012, Fernández-Méndez et al., 2015). Furthermore, the timing of our sampling suggests that the increased carbon supply by the spring bloom had already reached the seafloor and enhanced the benthic remineralisation (Graf, 1989) in both areas. The pattern of contrasts between the EG and WS area continued in the benthic food supply, which was also found by Boetius and Damm (1998) for areas with contrasting sea-ice cover at the continental margin of the Laptev Sea.'

We will further integrate the results from the suggested articles.

3. Water depth and vertical flux are well-documented highly influential factors structuring benthic communities both in terms of biodiversity and biomass/abundance anywhere in the ocean, in addition to sea ice cover. While these factors are mentioned in the discussion (without much literature support actually), it should also be noted more prominently that eastern Fram Strait receives constant inflow of particle rich Atlantic water, and this advective input adds to the vertical flux (see for example Wassmann et al. 2015 PiO for a summary). It is indeed complex to separate out the effects of water mass properties including particle content, and ice cover – a fact that should be acknowledged.

Reply: We will add information regarding the advective Atlantic input and acknowledge the complexity to track back the origin of organic matter resource. Indeed, there is quite some knowledge about the vertical carbon flux available. However, most of the data are from the more southerly and mainly sea-ice free locations in the Greenland Sea ("The Northern North Atlantic", edited by Schäfer, Schlüter and Thile). Owing to the complexity to separate out the effects of water mass properties, we only cited literature from very closed-by locations and thus, ensure a maximum of reliability of our comparison of remineralization data with the vertical carbon flux.

4. The authors said they struggled to find some relevant information (e.g. on primary production) for the western Fram Strait side, and therefore used values from the central Arctic. They might consider the results of the SFB313 that spent years investigating East Greenland including the slope, including carbon remineralization, primary production, benthic community structure etc., http://www.springer.com/us/book/9783540672319. Was the region never covered in any of the primary production models? Some additional useful information from eastern Fram Strait is also available, e.g. Wlodarska-K. et al. 2004 in DSRII.

Reply: We would like to excuse our unsuccessful literature research and thank the reviewer for the suggested literature. In the meantime, we found modeled primary production in the Arctic, which included estimates of primary production across Fram Strait. This source indicates the expected and

contrasting primary productivity between the EG and WS area (Pabi et al, 2008, doi:10.1029/2007JC004578). In addition, the suggested study of Codespoti et al. (2013, http://dx.doi.org/10.1016/j.pocean.2012.11.006) presents net community production values, which reflect new production and thus will also be used to give a more reliable insight into the relationship between primary production and benthic mineralization in the Fram Strait.

Small corrections/comments:
1. P4 l6 I would not call primary production and oxygen flux an ecosystem component, they are rate measurements of processes. The benthic community is an ecosystem component.

Reply: We will follow the reviewers' suggestion in the rewritten introduction (see reply to reviewers 2 'specific comments' no°1).

2. P4 Delete l6-8 (redundant to previous sentence).

Reply: We will delete the redundant sentence in the rewritten introduction (see reviewers 'specific comments' no°1).

3. L4 l9 Rather 'nutrient concentrations' (or which property of nutrients?)

Reply: Following the suggestions of the reviewer 1 Paul Renaud, all data regarding nutrients will be removed from the manuscript. However, we will add information regarding the nutrient state of the Fram Strait in the discussion.

4. P4l10 If this is to be general across the globe, add 'In general, benthic community …'

Reply: We will add the term 'In general,' to the sentence.

5. P4l14 and elsewhere. I was taught 'therefore' never starts a sentence.

Reply: We have to disagree with the reviewer and refer to the following websites:
http://grammarist.com/grammar/therefore/
https://www.iup.edu/writingcenter/writing-resources/grammar/common-problems-with-however,-therefore,-and-similar-words/

6. P4l20 'Western' Arctic is a rather undefined term, since different nations use it in very different ways, rather give the region.

Reply: We will follow the reviewers' suggestion and change 'western Arctic' to 'Chukchi and Beaufort Sea'.

7. P4l21-22 Unclear how the 'better fit' works when one doesn't know what other factors were included.

Reply: We will change the sentence to 'A pan-arctic benthic remineralisation model showed a better fit when water depth and benthic chlorophyll data (representing food supply from primary production) were taken into account, compared to a model using only water depth as controlling factor (Bourgeois et al., 2017). This indicates that surface primary production patterns and water depth are both relevant factors controlling benthic remineralisation in the Arctic Ocean.'.

8. P4l34 No need to repeat the three references for the same aspect since already given in l28 P5l17ff What time period is considered when talking about stable ice cover here? What time period is considered in the number of 0.6 years per decade? (And somewhere in the discussion the author talk about ice thinning, a bit of a contradiction.)

Reply: We will remove the repetitive references. The cited references only mentioned 'stable ice cover' without data support. Therefore, the dataset presented in our manuscript actually describes the sea-ice conditions for the first time in reliable, satellite-based numbers. The time-period for the sea-ice rejuvenation will be added. However, we have to disagree with the reviewer that a sea-ice rejuvenation is contradicting with a sea-ice thinning. Multi-year sea-ice is thicker than perennial, first-year sea-ice. Consequently, when sea-ice becomes younger, it is likely that it becomes thinner as well, which we pointed out in the introduction (P5L19).

9. P6l4 Why combine sea ice cover and nutrients under one sub-header? I suggest separating those sections.

Reply: As mentioned in the reviewers' small corrections / comments no°3, data regarding nutrients will be removed from the manuscript. Thereby, the identified issue will be solved.

10. P5l6 rather 'Study area and field sampling' or 'Study area and sample collection'. None of the sample preparation or processing is described here.

Reply: We will follow the reviewers' suggestion and us the term 'Study area and field sampling'

11. P6l9 Although both 'data are' and 'data is' is allowed per some dictionaries, it really should be 'data are' (one datum, several data).

Reply: We will change the term to 'data are' throughout the entire manuscript.

12. P6l15 Provide a reference for the nutrient measurement method.

Reply: As mentioned in the reviewers' small corrections / comments no°3, data regarding nutrients will be removed from the manuscript. However, we will add information regarding the nutrient state of the Fram Strait in the discussion.

13. P6l20 Which property of phospholipids and proteins and organic matter was measured – presumably concentrations?

Reply: We specify the measured property and change the sentence to 'Various biogenic sediment compounds including grain size, water content, chlorophyll a (Chl a) and phaeopigment concentrations (Phaeo), portion of total organic carbon (TOC), phospholipids concentrations, protein concentrations, portion of organic matter, and the bacterial enzymatic turnover rate (FDA) as bacterial activity proxy were determined from the sediments sampled by the MUC and chambers of the autonomous benthic lander system.'

14. P6 section 2.3 The methods description is extremely abbreviated, but it is an editor decision if this is sufficient.

Reply: We are aware of the intense use of abbreviations. However, all abbreviations are common and introduced before, as recommended by the manuscript guidelines of 'Biogeosciences'.

15. P7 2.4 What taxonomic resolution was aimed for?

Reply: We will add the aimed taxonomic resolution, which was at least class level for macrofauna and order level for meiofauna.

16. P12l6 It would be appropriate to include the nutrient profiles (at least upper water column) into the MS figures rather than the supplement given that the nutrient inventories provide the basis to the level of primary production possible (although measured after the bloom was done presumably). At the very least some concentration ranges should be mentioned. Define 'surface'.

Reply: As mentioned in the reviewers' small corrections / comments no°3, data regarding nutrients will be removed from the manuscript. However, we will add information regarding the nutrient state of the Fram Strait in the discussion.

17. P12l16 Why 'indicates'? Later you test this!

Reply: We will remove the sentence, as indeed we later test this.

18. P12l12 There are different opinions on this, but given that I would find at least a range of densities etc. presented (as is done in the next section 3.4). At the very least, table 2 should be referenced here so that the reader can find the results.

Reply: We will follow the reviewers' suggestion and present ranges for the parameter 'median grain size', 'portion of grain size >63 µm', 'water content' and 'porosity' in section 3.2. However, we will deviate from the pattern used in section 3.4 for the parameter Chl $a$, Phaeo, CPE, Chl a/CPE ratio, Chl a/Phaeo ratio, TOC, organic matter, proteins, lipids, FDA, as it would lead to an absolutely illegible paragraph. Therefore, we will present the minimum and maximum values across the entire Fram Strait only for Chl $a$, TOC and organic matter and will not distinguish between the EG and WS area. For the remaining parameter Phaeo, CPE, Chl a/CPE ratio, Chl a/Phaeo ratio, proteins, lipids, and FDA magnitudes will be given. In addition, we will refer the reader to Table 3 (former Table 2), Figure 3 and Supplement Table S4, which holds more detailed information. The text will be changed to 'The sediment bound Chl $a$ concentration ranged between 0.4 ± 0.3 µg ml$^{-1}$ sediment$^{-1}$ (n=15) at EG III and 12.7 ± 3.1 µg ml$^{-1}$ sediment$^{-1}$ (n=15) at SV I (Table 3) and differed significantly between the EG and WS area (Figure 3, Supplement Table S4). A similar pattern was found for sediment bound Phaeo concentrations and CPE concentration with over 4 –times higher median values in the WS area compared to the EG area (Figure 3). The Chl $a$/CPE and Chl $a$/Phaeo ratios did not differ between the EG and WS area (Supplement Table S4), which indicates that the benthic community in both areas fed on a similar food quality and received the spring bloom food supply at the same time, respectively. Sediment bound TOC ranged between 0.44 ± 0.04 % (n=15) at EG II and 1.58 ± 0.27 % (n=15) at SV I and differed between the EG and WS area, similar to organic matter, which ranged between 3.45 ± 0.6 % (n=15) at EG II and 12.0 ± 4.2 % (n=30) at HG III (Table 3, Figure 3, Supplement Table S4). Proteins, lipids and FDA also differed between the EG and WS area with 5.6 –times, 2.3 –times and 1.8 –times higher median values in the WS area, respectively (Figure 3, Supplement Table S4).'

19. P13l31-32 add 'rather than an actual interannual difference'

Reply: We will remove the entire sentence 'These differences are probably a result of the different sampling periods (June in 2014 and end of July/beginning of August 2015), resulting in different Phaeo and CPE concentrations.'. For justification please look at small corrections / comments no°20.

20. P14l5 Just above you wrote the different is likely related to the months, while this line states it is a spatial difference. Both may be true, but as written the statements seem contradictory.

Reply: By removing the sentence in P13l31-32 (small corrections / comments no°19), the inconsistency identified by the reviewer will be solved.

21. P14l26 Significant indeed, but the authors should mention that the global R values are rather low, same with the macrofauna results.

Reply: We agree with the reviewer that the global R values are low and added this information to the text. For example, it will be changed to 'Regarding macrofauna communities based on density (Global R = 0.257, p = 0.007) and biomass (Global R = 0.238, p = 0.003), the ANOSIM revealed significant but weak differences between the HSC and LSC area.'.

22. P15 l10 perhaps add ' marginally not significant'

Reply: We decided to omit the last part of the sentence. It now reads 'Further, the two-way crossed PERMANOVA revealed that the sea-ice coverage (LSC and HSC) explains a significant (p = 0.008) portion of the macrofauna density variability.' We reported that the result of the interaction effect of water depth and sea ice concentration on macrobenthic community biomass was significant. Therefore, it was pointless to look at the effects of the single factors, simply because the test just showed that their effect depends on the effect of the other factor.

23. P15l16 As phrased, this is not a question.

Reply: We will rewrite the sentence to 'The aim of this study was to link contrasting sea-ice conditions with…'

24. P15l23 Grammar. If there were a strong link ... we would expect .... (conditional)

Reply: We will change the sentence to 'If there were a strong link between sea-ice conditions and deep-sea benthic oxygen fluxes, we would expect contrasting primary production, benthic food supply, benthic community parameters and benthic oxygen fluxes between the EG and the WS area.'

25. P16l14/15 This is not the right place to mention this point, move to figure caption or results text.

Reply: We will remove this sentence, as the information is already implemented in the method description of the PCA.

26. P16l27 opposite to our expectations or in contrast to our expectations. The following PCA sentence is grammatically incorrect. The PCA only shows .. but does not test ...

Reply: We will change the sentences to 'This is in contrast to our expectations and to findings of Boetius and Damm (1998). However, a PCA only shows correlations but does not test for the significances of these relationships.'

27. P19l4 The Kortsch paper is on shallow nearshore hard bottom communities, not quite the right reference here.

Reply: Indeed, Kortsch et al is not an appropriate reference, as it deals with benthic changes in a fjord system. The reference will be removed and instead we added Harada (2015, doi: 10.1016/j.gloplacha.2015.11.005).

28. P19l10 In earlier sections the authors talk about 'stable ice conditions' in Fram Strait, while here they state that ice is thinning. Specifying by which metric the conditions are stable will relieve the contradiction.

Reply: We will specify, that the term 'stable conditions' is used in terms of the general pattern of the sea-ice concentration in the Fram Strait (west: high concentration/east: low concentration).

29. P19l14 My understanding of the Boetius et al. paper is that these authors discussed the high Melosira biomass to be generated on the shelf and maintained (but not produced) over the basin through constant resupply of –albeit low – nutrients during ice drift, not as a consequence of increasing algal biomass in the central Arctic. General: Someone should switch German to English comma rules throughout.

Reply: We are thankful for the additional perspective regarding the interpretation of the publication of Boetius et al. (2013). The aim of the sentence is to point out that the Fram Strait benthos did not receive any algae patches (as far as assessable), which would have had a dramatic impact on the microbial and therefore total remineralization and would indicate that our presented mineralization are underestimations. However, as this is not the case (no algae patches found), our results are reliable. However, in order to stress this, the sentence will be changed to 'However, fast sinking algae patches as reported by Boetius et al (2013) in the central Arctic, which would lead to increased benthic mineralization, were not observed during a video transect at EG IV in 2014 (pers. Comm. J. Taylor).'.

We will apply English comma rules by using the free-ware version of gramma software grammaly.com. and we will let the manuscript be checked by a native speaker.

30. Table 2. Use same number of decimals within one parameter (e.g. days with sea ice has between zero and two decimals).

Reply: We will adjust the number of decimals to be consistent throughout one parameter.

31. Table 3. Spell out HSC and LSC.

Reply: We will spell out HSC and LSC in the table caption. It will be changed to 'The table shows that there are differences in the macrofauna community between the highly sea-ice covered area (HSC) and the low sea-ice covered area (LSC), while this is not the case for the meiofauna community.'

32. Figure 1. Specify time frame for 'general summer sea ice extent', by month and period.

Reply: We will specify the month and period for the 'general summer sea-ice extent', which is September 1981-2010 (http://nsidc.org)

33. Figure 3. Indicate if any of the differences between EG and WS were statistically significant. This and other figures explain abbreviations or say in caption where they are explained.

Reply: We will indicate significant differences between the stations in the figure and add the number of observations for each bar according to the reviewer's suggestion and the suggestions of the reviewer Paul Renaud. Further, will explain used abbreviations in the figure and table captions.

34. Figure 6. Typos: Arctic missing 'c'. Sauter et al. and Bourgeois et al. missing periods after al.

Reply: We will correct the typo's in figure 6.

---

## Author Response (AR1)

Author's response

We like to thank the editor Ph.D. Tina Treude for given us the opportunity to re-submit our manuscript entitled 'Deep-sea benthic communities and oxygen fluxes in the Arctic Fram Strait controlled by sea-ice cover and water depth' und for support during the submission process. We further like to thank the referees Paul Renaud and the second anonymous referee for their helpful comments, which improved a first manuscript version substantially. In the following, first the comments from a referee is given, second the author responses (which starts with the term 'Reply'), and third the author's changes in manuscript (given in blue). Furthermore, a track changes version is included in the reply.

We first address the comments of the referee Paul Renaud and afterwards the comments of the anonymous reviewer.

Author's response to referee Paul Renaud:

Specific comments:

1. Time scales of response. Direct (and only linear) correlations between environmental parameters and O2 flux may be misleading, or non-representative, depending on when the samples are taken (and when relative to the bloom/flux phenologies among different stations). Benthic biomass/density/structure likely respond to various factors (especially food-related parameters which are often covariates of depth) in a more seasonally (or up to decadally) integrated fashion, whereas O2 consumption/C remineralization are often more responsive to food inputs on a much shorter time scale (approx. weeks) (e.g. Renaud et al. 2008 DSR II). This must be considered in your interpretation.

Reply: We thank the reviewer for this important aspect. As spring bloom data in this region for the studied period (2014-2015) are not yet citable and the authors are no experts in satellite data acquisition and analyses, we refer to the results of Cherkasheva et al. 2014 (http://dx.doi.org/10.1016/j.jmarsys.2013.11.008) to provide information regarding the date of the spring bloom. Furthermore, we acknowledged that correlations do not necessarily prove causal relationships and that oxygen flux measurements only represent a temporal snapshot. As we did not perform measurements during or right after the bloom deposition, we might have missed the immediate and short-term reaction of the benthos to the fresh organic matter. Nevertheless, we can expect a lower influence of macrofauna on the measured oxygen fluxes, compared to the findings in Renaud et al. 2008 (DSR II), owing to the high contribution of microbial benthic mineralization to the total benthic mineralization in the deep sea (Donis et al., 2016, Sauter et al. 2001, Wenzhöfer and Glud,2002), which is also expressed by the mean DOU/TOU ratio of 0.79 presented in our results.

P5, L27: The onset of the spring bloom usually starts in Mai (Cherkasheva et al., 2014).

P17, L6: However, a PCA only shows correlations which does not necessarily prove causal relationships and does not test for the significance of these relationships.

2. As noted in the Methods, the difference between the eastern and western Fram Strait and potential consequences for benthic processes go beyond ice cover. Advected POC/PON/dissolved nutrients and

warmer temperatures on the eastern side are far greater than on the western side. Of course that is linked to why the ice is there, but in this case, ice is more of a covariate and perhaps less likely a causative factors. In addition, different zooplankton and microbial communities can well lead to different 'food' deposition. This must be considered in detail if the two transects are to be comparable.

Reply: We agree with the reviewer on the differences in advective and vertical food input between the eastern and western Fram Strait. However, the origin of the food input to the benthos can be >3000 km away from the Fram Strait (Lalande et al., 2016, doi:10.1016/j.dsr.2016.04.014). Therefore, a reliable track back and a clear differentiation between the food origin of the eastern and western Fram Strait would be speculative.

P17, L17: Additionally, the origin of the primary production responsible for the benthic food supply is difficult to assess and can be located >3000 km from Fram Strait (Lalande et al., 2016). In turn, the complexity of advective and vertical pelagic food input influencing processes in the Fram Strait is not considered in the ecosystem snapshot.

Furthermore, we pointed out that we used the parameter 'sea-ice cover' as a proxy for primary production patterns. The sea-ice in the western Fram Strait represents a suppressed light availability and a reduced nutrient supply (owing to the main currents WSC and EGC). Both light availability and nutrient supply are the main drivers of primary production. This suits the findings of Pabi et al. (2008, doi:10.1029/2007JC004578), showing contrasting primary production quantities among the western and eastern Fram Strait.

P17, L21: To summarise, sea-ice cover in the Fram Strait is a proxy for light availability and nutrient supply and therefore represents primary production in Fram Strait. In addition, water depth represents a proceeding degradation state of settling organic material towards the sea floor (Belcher et al., 2016). Both processes are responsible for the food supply to the benthos. Therefore, the independent factors 'sea-ice cover' and 'water depth' were the most important abiotic factors in the Fram Strait as they controlled the benthic food supply.

3. Methods: it appears that most of the variables measured were only assessed from the top 1 cm of sediment. Can you provide a justification (data-based) for this? For meiofauna, it is often the top 2-3 cm that contains the majority of the fauna, and for macrofauna, at least the top 5 cm, even at deep-sea depths.

Reply: Indeed, microbial and meiofauna data were assessed from the top 1cm. Macrofauna data and the biogenic sediment compounds, however, were assessed from the top 5cm (MUC cores) and from even deeper sediments (benthic chamber sampled sediments). We improved the method section to clarify this.

P7, L19: For the determination of the meiofauna density and identification of meiofauna taxa, sediment subsamples were taken with modified syringes (3.14 cm² cross-sectional area) from MUC recovered sediment cores after oxygen flux measurements were performed and from benthic chambers. The first centimetre of each sample, usually holding the highest meiofauna density (Górska et al., 2014), was stored in borax buffered 4 % formaldehyde solution at 4 °C.

P7, L30: For macrofauna analyses, the 0–5 cm layer from MUC sediment cores and the entire remaining sediment from the benthic chambers was used, sieved over a 500 µm mesh and stored in borax buffered 4 % formaldehyde and stained with Rose Bengal (Heip et al., 1985).

Regarding the Meiofauna, we refer to Gorska et al. 2014 (http://dx.doi.org/10.1016/j.dsr.2014.05.010) and regarding microbial data, we refer to Quéric et al., 2004 (https://doi.org/10.1016/j.mimet.2004.02.005). Both studies show that most of the investigated organisms in the Hausgarten area occur in the top 1 cm.

4. Ice cover in the two 'regions' is essentially 70-80% vs 1-10% (heavy ice/no ice). Except for EG V (and N5 which is often excluded from analysis), there is nothing in between. How might this affect your results/interpretation? Many of the results from N5 are more similar to the LSC than the HSC stations (see comment 2 above).

Reply: We acknowledged in our discussion, that comparing only two sites (heavy ice/no ice) does not allow us to estimate the actual relationship between ice cover and the response variables. We would like to point out that only station SV I was often excluded from analyses, owing to its exceptional shallow water depth, compared to all other stations. The introduction of a third category (intermediate ice) would only be based on two stations (EG IV and N5). As these two stations are from the same water depth, they would not include the potential impact of water depth, which was identified as important and therefore would weaken the outcome of this approach.

P17, L21: Furthermore, by comparing only two sites (HSC/LSC) a statistical investigation of the actual relationship between ice cover and the response variables is not possible.

5. Ice cover as the key factor. Related to comment 2 above, have you evaluated whether correlations/differences between benthic parameters and ice cover are the strongest relationships among your data? Primary productivity, vertical flux attenuation, and essentially food supply to the sediment surface may or may not be caused by sea ice in any way. Or it could be a feature of Arctic vs Atlantic water supply that causes a 'cascade of processes' and sea ice cover may just be a covariate with limited or even no direct causative effect (hence a logic problem on p 15 l 23-25). Your discussion implies that ice is the overriding factor but I do not see where you tested for this, or if it is even possible to disentangle all these variables to isolate depth as the key factor. If you ran similar analyses but grouped stations based on water mass characteristics instead of ice cover you would find the same result.

Reply: We verified whether correlations/differences between benthic parameters and ice cover are the strongest relationships among your data by running the PCA. The eigenvalues indicated that 'TOC', 'Chl a' and 'Macrofauna biomass' were responsible for the gradient along the x-axis and 'water depth', organic matter' and 'sea-ice concentration' for the gradient along the y-axis. However, 'sea-ice' is a proxy for light availability and nutrient supply in Fram Strait and therefore represents primary production, whereas water depth represents pelagic mineralization and therefore the loss of organic matter in the pelagic zone. Both process are responsible for the final 'TOC' and 'Chl a' concentrations at the seafloor. To make this clearer, we included the proxy characterization in the method section, added the eigenvalues of the PCA to the manuscript and integrated our argumentation in the discussion.

P14, L26: The PCA on station specific, ex situ obtained mean values (Fig. 5) revealed that water depth was positively correlated with median grain size and negatively correlated with the DOU, the TOU, bacterial density, and the BPc. Sea-ice concentration was negatively correlated with the porosity, Chl a, TOC, organic matter, and solute exchange. Similarly, macrofauna biomass was negatively correlated with, water depth, sea-ice concentration, and the median grain size. The stations of the WS and EG area both followed the water depth gradient and shallower stations showed the higher oxygen fluxes. However, stations of the EG area were strongly influenced by the sea-ice cover, contained less organic matter and Chl a, and macrofauna biomass, compared to the WS stations. The two dimensions of the plot explained 72 % of the total variability of the data (Fig. 5). The eigenvalues indicated that 'Chl a', 'TOC', and 'Macrofauna biomass' (-0.89, -0.88, -0.83, respectively) were responsible for the gradient along the x-axis and 'Bacterial density', 'water depth', organic matter' and 'sea-ice concentration' (0.59, -0.57, -0.54, respectively) for the gradient along the y-axis.

6. P 12 l 6: unclear what water column nutrients, presented as a snapshot without context of 'preformed' (winter) concentrations add here. Bloom phenology certainly is responsible for e.g. the lower nitrate in WS vs EG. Consider removing these data. The discussion on p 15-16 and then sec 4.2 is not really based on the data collected, but more of a general pattern documented in the literature. I agree some of this should be included, but wouldn't a more extended and balanced discussion of benthic process rates and the other factors (proximal) responsible for variation in these rates be appropriate here?

Reply: We agree with the reviewer and removed the nutrient data from the manuscript. Further, we discussed benthic mineralization and the other proximal factors responsible for variation in these rates as well as the relation between primary production and benthic mineralization in more detail. Changes were made throughout the entire sections.

7. Nutrient supply under the ice in EG is extremely low and not expected to increase with further melting of sea ice (e.g. Mauritzen et al. 2011 Prog Oceanogr). This casts serious doubt into any scenario where increased PP due to more light is invoked.

Reply: We revised the potential future scenario, include spatial limitations and point out that this scenario only holds true for areas, where sea-ice disappears and nutrient supply will increase.

P19, L20: Our scenario is only suitable if sea-ice disappears and nutrient supply increase, which will result in enhanced primary production The development of future Arctic Ocean primary production patterns and changes is still under debate (Wassmann, 2011, Arrigo et al., 2012; Nicolaus et al., 2012, Boetius et al., 2013). However, it is likely that the described scenario becomes true in the Chukchi Sea and the Beaufort Sea, owing to the predicted strengthening of the nutrient rich Pacific inflow (Harada, 2015). Furthermore, owing to an increased atlantification, an increased nutrient supply is also likely for the continental margin at the Barents Sea (Neukermans et al., 2018). In addition, nutrient inflow by glacial and permafrost soil melt is also predicted to increase (Vonk et al., 2015). However, this riverine load might only enhance primary production at the shelf areas and therefore is not relevant for the deep sea. An enhanced primary production in the western Fram Strait is unlikely even if the light availability will increase, as the required nutrient supply increase is not expected for this region (Mauritzen et al., 2011).

Technical corrections/details

1. P4 l6-9: unclear sentence. Perhaps just unnecessary (same for l 14-17 as it just repeats what you have just written)

Reply: The sentence was removed.

2. P4, l 24: controlling the benthic ecosystem? Be more specific, including what you mean by 'labile organic matter' (different from benthic chlorophyll?)

Reply: The sentence was changed to:

P4, L28: However, the principal factor controlling microbial activity in their study was most likely the supply of labile organic matter such as chloroplastic pigment equivalents (CPE; Thiel, 1978), proteins and dissolved free amino acids.

3. P6 l 10: if the algorithm can estimate ice cover at over 100% then couldn't values between 0 and 100 also be mismeasurements? Could there be some (automated) check to assure that adjacent pixels are 'similar', or some other way of testing for mismeasurement in this range?

Reply: Whenever there was a mismeasurement, the algorithm output was "128". So it is not the case that a sea-ice concentration of 101% or 105% or 112% and so on, could be measured. Therefore, the algorithm does not estimate ice cover over 100%. The sentence was improved to make this point clearer. Furthermore, we added information regarding the quantity of these mismeasurements.

P6, L14: Satellite mismeasurements, which were <0.5% of the long-term data, resulted in an algorithm output value of "128" and were omitted from the dataset.

4. Also, is an annual average (vs some other ice cover parameter) the most relevant measure of ice cover?

Reply: We provided some alternative sea-ice concentration periods (mean of 1-month before sampling, mean since first of May (assumed spring bloom onset) till sampling) for the reader. However, as pointed out by the reviewer, "benthic biomass/density/structure likely responds to various factors (especially food-related parameters which are often covariates of depth) in a more seasonally integrated fashion". To acknowledge this, we used the annual sea-ice cover in the PCA.

5. P 6 l 25: frozen not frosted

Reply: We followed the suggestion of the reviewer and change "frosted" to "frozen".

P6, L27: All other samples were shock frozen at -80°C and stored at -20°C until they were analysed at the home laboratory.

6. P 8 l 25ff: Was non-local mixing (i.e. non-linear profiles) observed? How was this accounted for in the O2 flux calculation?

Reply: Non-local mixing was observed in some cases and therefore the reported DOUs for those cases are underestimations. However, only eight out of 81 ex situ obtained oxygen microprofiles at various

stations and in one out of 34 in situ obtained oxygen microprofiles showed signs of non-local mixing. This information was added to the method section.

7. P 9 l 1-5: How much of the sediment mass could be attributed to salt from the drying process?

Reply: It was 4.5% ± 1.9 over all samples. We added this information to the method section.

P9, L11: Non-local mixing was observed in some microprofiles and therefore the reported DOUs for those cases are underestimations. However, only at eight out of 81 ex situ obtained oxygen microprofiles at various stations and at one out of 34 in situ obtained oxygen microprofiles signs of non-local mixing were observed.

8. P 10 l 5: consecutive not subsequent

Reply: We followed the reviewers' suggestions and change 'subsequent' to 'consecutive'

P10, L11: The analysed data were obtained during two consecutive years (Table 1).

9. P 10 l 14: 'x to zero mean and unit variance' is unclear

Reply: In most applications of a PCA (e.g. as a factor analysis technique), variables are often measured in different units. For such data, the data must be standardized to zero mean and unit variance, a common standardization procedure. If this is not done, high values (e.g. macrofauna biomass with values of ten thousands of mg m$^{-2}$) will get a greater importance than low values (e.g. DOU with values of max. 2.1 mmol O2 m$^{-2}$d$^{-1}$). Similar terms used for this procedure are 'data normalization' or 'z-scoring'. Though, as we followed the suggestions provided by Buttigieg and Ramette (2014), we decided to follow their term of 'standardization' (https://mb3is.megx.net/gustame/indirect-gradient-analysis/pca).

10. P 10 l 21: you must exclude EG II from the analysis. You cannot make the assumption and assign a value. It was fine to exclude the shallow station, and you should do the same with EG II

Reply: We agree with the reviewers' comment that the assumption of a solute exchange value for EG II is not a valid approach to deal with data gaps. However, as the other parameters included in the PCA from EG II were actually measured, we rather prefer to perform the PCA without the parameters of 'solute exchange' from all stations. With this suggestion, EG II would still be part of the central analysis of the paper. Furthermore, as 'solute exchange' is well correlated with other parameters such as macrofauna biomass, it will still be represented in the PCA.

11. P 10 l 30: You need to indicate whether there was a different depth relationship between the two regions and then say what you did if this was (or was not) the case.

Reply: We added the information, that the depth relationships in terms of the bottom slope were similar between the two regions to the method section ('study site'). Furthermore, we indicate whether there

were different depth relationships of variables between the two regions. We discussed that the microbial mineralization is the main driver of benthic deep-sea mineralization (see reply to reviewers' specific comment no°1). We further added to the discussion that microbial density did not show differences between the HSC and LSC categories and therefore was not the biotic link which connected the food input pattern (Figure S4) with the mineralization pattern (Figure 4).

P5, L32: Thereby the stations in the EG area (namely EG I, EG II, EG III, EG IV and EG V) and in the HG area (namely SV I, HG I, SV IV, HG II, HG III, HG IV, and N5) form a bathymetric transect with a similar bottom slope of ~11°.

For changes made in discussion section, please see the track changes version of the manuscript in attachment.

12. P 12 l 6: unclear what water column nutrients, presented as a snapshot without context of 'preformed' (winter) concentrations add here. Bloom phenology certainly is responsible for e.g. the lower nitrate in WS vs EG. Consider removing these data. The discussion on p 15-16 is not really based on the data collected, but more of a general pattern documented in the literature.

Reply: We followed the reviewers' suggestion and removed the nutrient data from the manuscript.

13. P 12 l 10ff (and Fig 3). Please indicate any statistical results such that the figures correspond to what is written in the text regarding comparisons between the two regions. Only statistically significant results should be expressed as 'differences' (e.g. solute exchange is likely NS but significance is implied). Also, please clarify how many stations (and depth profile) each bar represents. This has some bearing on your comments about variability between the two locations.

Reply: We improved the text and used the term 'differences' only in case of significant differences. Further, we indicated significant differences between the stations in the figure and add the number of observations for each bar.

As an example, the results of the sediment compounds read now 'The mean DOU in the EG area ranged between $0.4 \pm 0.1$ mmol $O_2$ m$^{-2}$d$^{-1}$ (n=10) at EG V and $1.0 \pm 0.1$ mmol $O_2$ m$^{-2}$d$^{-1}$ (n=10) at EG II. In the WS area, DOUs at stations within the same water depth range as the EG stations ranged between $0.5 \pm 0.2$ mmol $O_2$ m$^{-2}$d$^{-1}$ (n=8) at HG IV and $2.1 \pm 0.6$ mmol $O_2$ m$^{-2}$d$^{-1}$ (n=8) at SV IV. At the shallow station SV I the DOU reached $3.0 \pm 1.7$ mmol $O_2$ m$^{-2}$d$^{-1}$ (n=6, Table 3). The mean TOU in the EG area ranged between $0.9 \pm 0.3$ mmol $O_2$ m$^{-2}$d$^{-1}$ (n=2) at EG I and $1.6$ mmol $O_2$ m$^{-2}$d$^{-1}$ (n=1) at EG II. Similar mean TOU values were measured in the WS area, at stations within the same water depth range as the EG stations. TOU values ranged between $0.5 \pm 0.2$ mmol $O_2$ m$^{-2}$d$^{-1}$ (n=5) at HG IV Lander and $1.9 \pm 0.6$ mmol $O_2$ m$^{-2}$d$^{-1}$ (n=5) at HG I. At the shallow SV I station TOU reached $5.1 \pm 0.3$ mmol $O_2$ m$^{-2}$d$^{-1}$ (n=3, Table 3). DOU differed significantly between the WS and EG area, while TOU was similar among the areas (Fig. 3, Supplement Table S4).

14. P 12 l 18: 'pelagic food supply indicating parameter in the sediment' rephrase to clarify that these are sediment values and careful about how you define food quality. Not all organisms eat chlorophyll (in fact maybe few actually do). Bacteria themselves are likely food for many organisms, and phaeopigments and other OC may also be quite high quality food for others.

Reply: We followed the suggestion of the anonymous second reviewer, added ranges of values regarding benthic food supply representing parameters.

P12, L30: The sediment bound Chl a concentration ranged between 0.4 ± 0.3 µg ml-1 sediment-1 (n = 15) at EG III and 12.7 ± 3.1 µg ml-1 sediment-1 (n = 15) at SV I (Table 3) and differed significantly between the EG and WS area (Figure 3, Supplement Table S4). A similar pattern was found for sediment bound Phaeo concentrations and CPE concentration with over 4 –times higher median values in the WS area compared to the EG area (Figure 3). The Chl a/CPE and Chl a/Phaeo ratios did not differ between the EG and WS area (Supplement Table S4), which indicates that the benthic community in both areas fed on a similar food quality and received the spring bloom food supply at the same time, respectively. Sediment bound TOC ranged between 0.44 ± 0.04 % (n = 15) at EG II and 1.58 ± 0.27 % (n = 15) at SV I and differed between the EG and WS area, similar to organic matter, which ranged between 3.45 ± 0.6 % (n = 15) at EG II and 12.0 ± 4.2 % (n = 30) at HG III (Table 3, Figure 3, Supplement Table S4). Proteins, lipids and FDA also differed between the EG and WS area with 5.6 times, 2.3 times, and 1.8 times higher median values in the WS area, respectively (Figure 3, Supplement Table S4).

15. P13 l 21: 'which indicates bacterial activity and bacterial remineralisation as the major oxygen consumer' please indicate why you conclude this. Why would bacterial oxygen consumption not be reflected in DOU data. These are effectively two different techniques to measure the same thing, each with underlying assumptions. The conclusion you make regarding the ratio is not supported.

Reply: We rephrased the sentence to:

The mean DOU/TOU ratio, which describes the fraction of the total community mediated oxygen flux (TOU) covered by the microbial mediated oxygen flux (DOU, Glud, 2008,) across the entire Fram Strait was 0.79 ± 0.30, with 0.63 ± 0.22 in the EG area and 0.92 ± 0.30 in the WS area, indicating that the total oxygen uptake is mainly microbial mediated.

16. P 14 l 7-13: I would focus on the differences among EG and WS stations as revealed by PCA, and not individual variable correlations (which are NOT real correlations but instead are ordination-based relationships! If you want to look for correlation then run that analysis on the raw data).

Reply: We followed the suggestion of the reviewer and changed the paragraph to emphasize the differences among EG and WS stations.

P14, L29: The stations of the WS and EG area both followed the water depth gradient and shallower stations showed the higher oxygen fluxes. However, stations of the EG area were strongly influenced by the sea-ice cover, contained less organic matter and Chl a, and macrofauna biomass, compared to the WS stations.

However, the reason to perform the PCA was to reveal the relationships between the multiple parameters. Therefore, we performed a PCA in the scaling II mode, which emphasize the relationships

between parameters (Buttigieg and Ramette, 2014, https://doi.org/10.1111/1574-6941.12437). We added this information to the method section. A performance on raw data as suggested by the reviewer is, however, not recommendable due to the reasons presented in the comment no°9. The correlation of single parameters with each other was already given in Table S2 and Figure S3. In addition, a PCA is a procedure that transforms a number of (possibly) correlated variables into a (smaller) number of uncorrelated variables called principal components. Thus a PCA eliminates redundant information. As it also gives the contribution of the single parameter (=strength of influence) to each principle component, the parameters most likely control the investigated area can be identified (Boetius and Damm, 1998, doi: https://doi.org/10.1016/S0967-0637(97)00052-6).

17. P 16 l 17: but macrofauna biomass has a similar relationship with depth between the two ice-cover systems

Reply: We rephrased the sentence to Continuing the cascade of dependencies, benthic community parameters should follow the same pattern as the sea ice at the surface and the benthic food supply parameters. Indeed, there were differences between the EG and WS area regarding meiofauna density and macrofauna density but not in the macrofauna biomass.

18. P 16 l 18-20. Repetitive

Reply: We followed the reviewers' suggestion and remove the repetitive content.

19. P 16 l 27: A CCA or RDA would find significant relationships.

Reply: We tested our data regarding the suitability of the usage of a CCA ('decorana'-command in R package vegan) with the outcome that our data showed linear correlations, so only a RDA would be a suitable approach. Usually, an RDA is used to correlate parameters of two information layers: environmental and biotic data, where the former influences the latter. However, from our point of view, we have three information layers: environmental, biotic, and flux data. It is common knowledge that environmental data influence biotic data and both are influencing the flux data. Therefore, we decided for an indirect ordination by a PCA as the direct approach of an RDA incorporates the measured parameters into the ordination, without considering the possible influence of other, unmeasured parameters.

20. Sec 4.3: first paragraph unnecessary.

Reply: We followed the reviewers' suggestion and remove the first paragraph.

21. P 19 l 4: neither citation is in the references. Are you sure the Kortsch ref is appropriate?

Reply: The reference of Jones et al., 2014 was added to the references, whereas Kortsch et al., 2012 was removed as suggested by the reviewer and we added the reference of Harada (2015) instead.

Authors response to second anonymous referee:

Specific comments:

1. The introduction will benefit from turning the lists of which factors depend on which other factors into a narrative explaining how they influence each other. This change would necessarily make the introduction a bit longer, but improve the logic, flow and justification for the study. Also, the authors would help the reader by providing a bit of background why they estimate remineralization of new production rather than calculating it because they can.

Reply: We followed the reviewers' suggestion and rewrote the introduction into a narrative explaining how factors influence each other by giving the information how single factors are correlated which each other. Further, we pointed out the link between the new production and the remineralization.

Benthic deep-sea remineralisation depends on primary production and is as such closely linked with primary production patterns, known as pelagic–benthic coupling (Graf, 1989). The relationship, however, includes many and partly inter-dependent factors. Benthic deep-sea remineralisation is positively correlated with surface primary production (Graf et al., 1995; Wenzhöfer and Glud, 2002; Smith et al., 2016), which is on its turn controlled by light availability and nutrient supply (Kirk, 2011; Cherkasheva et al., 2014; Fernández-Méndez et al., 2015). Though, only the annual new production leaves the euphotic zone (Platt et al., 1989) and can supply the benthos with organic carbon. Benthic remineralisation is negatively correlated to water depth (Jahnke et al, 1990; Jahnke, 1996; Wenzhöfer and Glud, 2002), as it represents a loss of organic carbon by pelagic remineralisation (Rullkötter, 2006; Belcher et al., 2016) and thereby a loss of benthic food. After organic carbon reached the seafloor, it is ingested and remineralised by the benthic community. Benthic community parameters, e.g. biomass, density, structure, and functions of different fauna size classes, are controlled by food supply (and thus by primary production) and water depth (Piepenburg et al., 1997; Flach et al., 2002; Smith et al., 2008) but also by sediment properties (Wheatcroft, 1992; Vanreusel et al., 1995). Benthic remineralisation rates also depend on benthic community biomass (Glud et al., 1994). Furthermore, benthic remineralisation is enhanced if the benthic community intensifies oxygenation of the seafloor (Glud, 2008) and thus also depends on the benthic community structure. Therefore, the ecosystem processes primary production, pelagic remineralisation and benthic remineralisation, as well as the components benthic community biomass, density, and structure are controlled by abiotic and biotic factors and additionally create a cascade of dependencies from the ocean's surface zone of primary production to and within the deep-sea benthos.

2. The discussion (especially 4.1 and 4.2) repeats the results to a large extent. Instead, it should place the results in the context of the extensive literature from the area and beyond. I recommend the authors summarize their findings more concisely and discuss their results in the context of, for example, the pan-Arctic scale Progress in Oceanography issue from 2015, primary productions model estimates covering the area, the series of three articles from Patrai – Codispoti 2013 etc.

Reply: Indeed, we repeated the results to a large extent, as we first needed to interpret our results before we could start to place the output in the context. However, we reduced the repetition where ever possible and thereby followed the reviewer's suggestion.

For example, the second paragraph of section 4.1 was changed to:

The results of Pabi et al. (2008) showed that the annual primary production pattern follows the general sea-ice concentration pattern in the Fram Strait and is up to 10-times larger in the WS area compared to the EG area. Thus, the sea-ice concentration represents the general primary production pattern in the Fram Strait. As the sampling was performed in Mid/End of June 2014 and July/August 2015, it is very likely that the spring bloom, which usually starts in May (Cherkasheva et al., 2014), had finished. This is indicated by lower nutrient concentrations in water depth ≤50 m compared to the nutrient concentrations between >50–300 m water depths (Graeve and Ludwichowski, 2017a, b). The N:P ratio in the upper 50 m during the expeditions was six and seven in the EG and WG area, respectively (Graeve and Ludwichowski, 2017a, b), indicating that primary production was nitrate limited, similar to the permanently sea-ice covered central Arctic Ocean (Tremblay et al., 2012, Fernández-Méndez et al., 2015). Furthermore, the timing of our sampling suggests that the increased carbon supply by the spring bloom had already reached the seafloor and enhanced the benthic remineralisation (Graf, 1989) in both areas. The pattern of contrasts between the EG and WS area continued in the benthic food supply, which was also found by Boetius and Damm (1998) for areas with contrasting sea-ice cover at the continental margin of the Laptev Sea.

We further integrated the results from the suggested articles.

3. Water depth and vertical flux are well-documented highly influential factors structuring benthic communities both in terms of biodiversity and biomass/abundance anywhere in the ocean, in addition to sea ice cover. While these factors are mentioned in the discussion (without much literature support actually), it should also be noted more prominently that eastern Fram Strait receives constant inflow of particle rich Atlantic water, and this advective input adds to the vertical flux (see for example Wassmann et al. 2015 PiO for a summary). It is indeed complex to separate out the effects of water mass properties including particle content, and ice cover – a fact that should be acknowledged.

Reply: We added information regarding the advective Atlantic input and acknowledge the complexity to track back the origin of organic matter resource (see reply no°2 to specific comments of first referee Paul Renaud). Indeed, there is quite some knowledge about the vertical carbon flux available. However, most of the data are from the more southerly and mainly sea-ice free locations in the Greenland Sea ("The Northern North Atlantic", edited by Schäfer, Schlüter and Thile). Owing to the complexity to separate out the effects of water mass properties, we only cited literature from very closed-by locations and thus, ensure a maximum of reliability of our comparison of remineralization data with the vertical carbon flux.

4. The authors said they struggled to find some relevant information (e.g. on primary production) for the western Fram Strait side, and therefore used values from the central Arctic. They might consider the results of the SFB313 that spent years investigating East Greenland including the slope, including carbon remineralization, primary production, benthic community structure etc., http://www.springer.com/us/book/9783540672319. Was the region never covered in any of the primary production models? Some additional useful information from eastern Fram Strait is also available, e.g. Wlodarska-K. et al. 2004 in DSRII.

Reply: We would like to excuse our unsuccessful literature research and thank the reviewer for the suggested literature. In the meantime, we found modeled primary production in the Arctic, which included estimates of primary production across Fram Strait. This source indicated the expected and contrasting primary productivity between the EG and WS area (Pabi et al, 2008, doi:10.1029/2007JC004578). The suggested study of Codespoti et al. (2013, http://dx.doi.org/10.1016/j.pocean.2012.11.006), which included values of new production, was used to give a more reliable insight into the relationship between primary production and benthic mineralization in the Fram Strait.

Small corrections/comments:

1. P4 l6 I would not call primary production and oxygen flux an ecosystem component, they are rate measurements of processes. The benthic community is an ecosystem component.

Reply: We followed the reviewers' suggestion, rewrote the introduction (see reply to anonymous reviewer 'specific comments' no°1) and thereby changed the term 'component' to 'process'.

2. P4 Delete l6-8 (redundant to previous sentence).

Reply: We deleted the redundant sentence in the rewritten introduction (see anonymous reviewers 'specific comments' no°1).

3. L4 l9 Rather 'nutrient concentrations' (or which property of nutrients?)

Reply: Following the suggestions of the reviewer Paul Renaud, all data regarding nutrients were removed from the manuscript. However, we added information regarding the nutrient state of the Fram Strait in the 'Study site' section and added the N:P ratio to the discussion.

P16, L19: This is indicated by lower nutrient concentrations at water depth ≤50 m compared to the nutrient concentrations between >50–300 m water depths (Graeve and Ludwichowski, 2017a, b). The N:P ratio in the upper 50 m during the expeditions was six and seven in the EG and WG area, respectively (Graeve and Ludwichowski, 2017a, b), indicating that primary production was nitrate limited, similar to the permanently sea-ice covered central Arctic Ocean (Tremblay et al., 2012; Fernández-Méndez et al., 2015).

4. P4l10 If this is to be general across the globe, add 'In general, benthic community ...'

Reply: We added the term 'In general,' to the sentence.

5. P4l14 and elsewhere. I was taught 'therefore' never starts a sentence.

Reply: We have to disagree with the reviewer and refer to the following websites:

http://grammarist.com/grammar/therefore/

https://www.iup.edu/writingcenter/writing-resources/grammar/common-problems-with-however,-therefore,-and-similar-words/

6. P4l20 'Western' Arctic is a rather undefined term, since different nations use it in very different ways, rather give the region.

Reply: We followed the reviewers' suggestion and change 'western Arctic' to 'Chukchi and Beaufort Sea'.

7. P4l21-22 Unclear how the 'better fit' works when one doesn't know what other factors were included.

Reply: We changed the sentence to 'A pan-arctic benthic remineralisation model showed a better fit when water depth and benthic chlorophyll data (representing food supply from primary production) were taken into account, compared to a model using only water depth as controlling factor (Bourgeois et al., 2017). This indicates that surface primary production patterns and water depth are both relevant factors controlling benthic remineralisation in the Arctic Ocean.'.

8. P4l34 No need to repeat the three references for the same aspect since already given in l28 P5l17ff What time period is considered when talking about stable ice cover here? What time period is considered in the number of 0.6 years per decade? (And somewhere in the discussion the author talk about ice thinning, a bit of a contradiction.)

Reply: We removed the repetitive references. The cited references only mentioned 'stable ice cover' without data support. Therefore, the dataset presented in our manuscript actually describes the sea-ice conditions for the first time in reliable, satellite-based numbers. The time-period for the sea-ice rejuvenation was added. However, we have to disagree with the reviewer that a sea-ice rejuvenation is contradicting with a sea-ice thinning. Multi-year sea-ice is thicker than perennial, first-year sea-ice. Consequently, when sea-ice becomes younger, it is likely that it becomes thinner as well, which we pointed out in the introduction.

P5, L25: However, the sea-ice age becomes younger by 0.6 years per decade (2001–2012, Krumpen et al., 2015), which goes along with a decrease in the sea-ice thickness (Renner et al., 2014; Krumpen et al., 2015). The onset of the spring bloom usually starts in Mai (Cherkasheva et al., 2014).

9. P6l4 Why combine sea ice cover and nutrients under one sub-header? I suggest separating those sections.

Reply: As mentioned in the reviewers' small corrections / comments no°3, data regarding nutrients were removed from the manuscript. Thereby, the identified issue was solved.

10. P5l6 rather 'Study area and field sampling' or 'Study area and sample collection'. None of the sample preparation or processing is described here.

Reply: We followed the reviewers' suggestion and used the term 'Study area and field sampling'

11. P6l9 Although both 'data are' and 'data is' is allowed per some dictionaries, it really should be 'data are' (one datum, several data).

Reply: We changed the term to 'data are' throughout the entire manuscript.

12. P6l15 Provide a reference for the nutrient measurement method.

Reply: As mentioned in the reviewers' small corrections / comments no°3, data regarding nutrients were removed from the manuscript. However, we added information regarding the nutrient state of the Fram Strait (see reply to anonymous reviewer's small corrections / comments no°3).

13. P6l20 Which property of phospholipids and proteins and organic matter was measured – presumably concentrations?

Reply: We specify the measured property and change the sentence to 'Various biogenic sediment compounds including grain size, water content, chlorophyll a (Chl a) and phaeopigment concentrations (Phaeo), portion of total organic carbon (TOC), phospholipids concentrations, protein concentrations, portion of organic matter, and the bacterial enzymatic turnover rate (FDA) as bacterial activity proxy were determined from the sediments sampled by the MUC and chambers of the autonomous benthic lander system.'

14. P6 section 2.3 The methods description is extremely abbreviated, but it is an editor decision if this is sufficient.

Reply: We are aware of the intense use of abbreviations. However, all abbreviations are common and introduced before, as recommended by the manuscript guidelines of 'Biogeosciences'.

15. P7 2.4 What taxonomic resolution was aimed for?

Reply: We added the aimed taxonomic resolution, which was at least class level for macrofauna and order level for meiofauna.

16. P12l6 It would be appropriate to include the nutrient profiles (at least upper water column) into the MS figures rather than the supplement given that the nutrient inventories provide the basis to the level of primary production possible (although measured after the bloom was done presumably). At the very least some concentration ranges should be mentioned. Define 'surface'.

Reply: As mentioned in the reviewers' small corrections / comments no°3, data regarding nutrients were removed from the manuscript. However, we added information regarding the nutrient state of the Fram Strait (see reply to anonymous reviewer's small corrections / comments no°3).

17. P12l16 Why 'indicates'? Later you test this!

Reply: We removed the sentence, as indeed we later test this.

18. P12l12 There are different opinions on this, but given that I would find at least a range of densities etc. presented (as is done in the next section 3.4). At the very least, table 2 should be referenced here so that the reader can find the results.

Reply: We followed the reviewers' suggestion and present ranges for the parameter 'median grain size', 'portion of grain size >63 μm', 'water content' and 'porosity' in section 3.2. However, we deviated from the pattern used in section 3.4 for the parameter Chl *a*, Phaeo, CPE, Chl a/CPE ratio, Chl a/Phaeo ratio, TOC, organic matter, proteins, lipids, FDA, as it would lead to an absolutely illegible paragraph. Therefore, we presented the minimum and maximum values across the entire Fram Strait only for Chl *a*, TOC and organic matter and not distinguish between the EG and WS area. For the remaining parameter Phaeo, CPE, Chl a/CPE ratio, Chl a/Phaeo ratio, proteins, lipids, and FDA magnitudes were given. In addition, we refered the reader to Table 3 (former Table 2), Figure 3 and Supplement Table S4, which holds more detailed information.

The sediment bound Chl *a* concentration ranged between $0.4 \pm 0.3$ μg ml$^{-1}$ sediment$^{-1}$ (n=15) at EG III and $12.7 \pm 3.1$ μg ml$^{-1}$ sediment$^{-1}$ (n=15) at SV I (Table 3) and differed significantly between the EG and WS area (Figure 3, Supplement Table S4). A similar pattern was found for sediment bound Phaeo concentrations and CPE concentration with over 4 –times higher median values in the WS area compared to the EG area (Figure 3). The Chl *a*/CPE and Chl *a*/Phaeo ratios did not differ between the EG and WS area (Supplement Table S4), which indicates that the benthic community in both areas fed on a similar food quality and received the spring bloom food supply at the same time, respectively. Sediment bound TOC ranged between $0.44 \pm 0.04$ % (n=15) at EG II and $1.58 \pm 0.27$ % (n=15) at SV I and differed between the EG and WS area, similar to organic matter, which ranged between $3.45 \pm 0.6$ % (n=15) at EG II and $12.0 \pm 4.2$ % (n=30) at HG III (Table 3, Figure 3, Supplement Table S4). Proteins, lipids and FDA also differed between the EG and WS area with 5.6 –times, 2.3 –times and 1.8 –times higher median values in the WS area, respectively (Figure 3, Supplement Table S4).

19. P13l31-32 add 'rather than an actual interannual difference'

Reply: We removed the entire sentence 'These differences are probably a result of the different sampling periods (June in 2014 and end of July/beginning of August 2015), resulting in different Phaeo and CPE concentrations.'. For justification please look at small corrections / comments no°20.

20. P14l5 Just above you wrote the different is likely related to the months, while this line states it is a spatial difference. Both may be true, but as written the statements seem contradictory.

Reply: By removing the sentence in P13l31-32 (small corrections / comments no°19), the inconsistency identified by the reviewer was solved.

21. P14l26 Significant indeed, but the authors should mention that the global R values are rather low, same with the macrofauna results.

Reply: We agree with the reviewer that the global R values are low. We added this information to the text.

P15, L15: Regarding macrofauna communities based on density (Global R = 0.257, p = 0.007) and biomass (Global R = 0.238, p = 0.003), the ANOSIM revealed significant but weak differences between the HSC and LSC area. SIMPER routine results indicated dissimilarities of 56 % for the macrofauna density and 76 % for the macrofauna biomass between the HSC and LSC areas. The taxa which contributed most to the average similarity within and to the average dissimilarity between the HSC and LSC area are given in Supplement Table S10. The ANOSIM results for water depth groups showed that bathymetry could at least explain the dissimilarity in meiofauna communities based on density (Global R = 0.219; p = 0.01), even if the difference was weak. The SIMPER analysis, however, showed that the observed differences in meiofauna density regarding water depth were mainly due to the marked difference between the shallowest station (SV I at 275 m) and all other stations deeper than 1000 m (dissimilarity >50 %, Supplement Table S11). ANOSIM results for macrofauna communities based on density (Global R = 0.2, p = 0.008) and biomass (Global R = 0.346, p = 0.0001) revealed significant but also weak differences between water depth categories with >50 % dissimilarity between all water depth categories for macrofauna density (except between 1000 m and 1500 m) and macrofauna biomass (SIMPER, Supplement Table S11).

22. P15 l10 perhaps add ' marginally not significant'

Reply: We decided to omit the last part of the sentence. It now reads 'Further, the two-way crossed PERMANOVA revealed that the sea-ice coverage (LSC and HSC) explains a significant (p = 0.008) portion of the macrofauna density variability.' We reported that the result of the interaction effect of water depth and sea ice concentration on macrobenthic community biomass was significant. Therefore, it was pointless to look at the effects of the single factors, simply because the test just showed that their effect depends on the effect of the other factor.

23. P15l16 As phrased, this is not a question.

Reply: We rephrased the sentence to 'The aim of this study was to link contrasting sea-ice conditions with…'

24. P15l23 Grammar. If there were a strong link ... we would expect .... (conditional)

Reply: We changed the sentence to 'If there were a strong link between sea-ice conditions and deep-sea benthic oxygen fluxes, we would expect contrasting primary production, benthic food supply, benthic community parameters and benthic oxygen fluxes between the EG and the WS area.'

25. P16l14/15 This is not the right place to mention this point, move to figure caption or results text.

Reply: We removed the sentence, as the information is already implemented in the method description of the PCA.

26. P16l27 opposite to our expectations or in contrast to our expectations. The following PCA sentence is grammatically incorrect. The PCA only shows .. but does not test ...

Reply: We changed the sentences to 'This is in contrast to our expectations and to findings of Boetius and Damm (1998). However, a PCA only shows correlations but does not test for the significances of these relationships.'

27. P19l4 The Kortsch paper is on shallow nearshore hard bottom communities, not quite the right reference here.

Reply: Indeed, Kortsch et al is not an appropriate reference, as it deals with benthic changes in a fjord system. The reference was removed and instead we added Harada (2015, doi: 10.1016/j.gloplacha.2015.11.005).

28. P19l10 In earlier sections the authors talk about 'stable ice conditions' in Fram Strait, while here they state that ice is thinning. Specifying by which metric the conditions are stable will relieve the contradiction.

Reply: We specified, that the term 'stable conditions' is used in terms of the general pattern of the sea-ice concentration in the Fram Strait (west: high concentration/east: low concentration).

P12, L14: This pattern also occurred in the other short-term datasets and in the long-term dataset. The latter indicated that the sea-ice cover in terms of sea-ice concentration was stable across the Fram Strait the last 15 years (Fig. 2, Supplement Table S3).

29. P19l14 My understanding of the Boetius et al. paper is that these authors discussed the high Melosira biomass to be generated on the shelf and maintained (but not produced) over the basin through constant resupply of –albeit low – nutrients during ice drift, not as a consequence of increasing algal biomass in the central Arctic. General: Someone should switch German to English comma rules throughout.

Reply: We are thankful for the additional perspective regarding the interpretation of the publication of Boetius et al. (2013). The aim of the sentence is to point out that the Fram Strait benthos did not receive any algae patches (as far as assessable), which would have had a dramatic impact on the microbial and therefore total remineralization and would indicate that our presented mineralization are underestimations. As this is not the case (no algae patches found), our results are reliable. However, in order to stress this, the sentence was changed to 'However, fast sinking algae patches as reported by Boetius et al (2013) in the central Arctic, which would lead to increased benthic mineralization, were not observed during a video transect at EG IV in 2014 (pers. Comm. J. Taylor).'.

We applied English comma rules by using the free-ware version of gramma software grammaly.com. and we the manuscript was checked by a native speaker.

30. Table 2. Use same number of decimals within one parameter (e.g. days with sea ice has between zero and two decimals).

Reply: We adjusted the number of decimals to be consistent throughout one parameter.

31. Table 3. Spell out HSC and LSC.

Reply: We spelled out HSC and LSC in the table caption. It was changed to 'The table shows that there are differences in the macrofauna community between the highly sea-ice covered area (HSC) and the low sea-ice covered area (LSC), while this is not the case for the meiofauna community.'

32. Figure 1. Specify time frame for 'general summer sea ice extent', by month and period.

Reply: We specified the month and period for the 'general summer sea-ice extent', which is September 1981-2010 (http://nsidc.org)

33. Figure 3. Indicate if any of the differences between EG and WS were statistically significant. This and other figures explain abbreviations or say in caption where they are explained.

Reply: We indicated significant differences between the stations in the figure and add the number of observations for each bar according to the reviewer's suggestion and the suggestions of the reviewer Paul Renaud. Further, we explained the used abbreviations in the figure and table captions.

34. Figure 6. Typos: Arctic missing 'c'. Sauter et al. and Bourgeois et al. missing periods after al.

Reply: We corrected the typo's in figure 6.

In addition to the above mentioned changes, further changes were made in the manuscript either to increase readability (e.g. improved Fig. 1) or additional found flaws and typos were corrected. All changes that were made are trackable in the provide track-changes-version of the manuscript placed below.

[revised manuscript text omitted]

---

## Editor Decision (ED1)

Author's response

We like to thank the editor Ph.D. Tina Treude for the opportunity to re-submit our manuscript entitled 'Deep-sea benthic communities and oxygen fluxes in the Arctic Fram Strait controlled by sea-ice cover and water depth' und for great support during the submission process. We further like to thank the anonymous referee for her/his helpful comments on our second manuscript version. In the following, first the comments of the referee are given and second the author responses (which starts with the term 'Reply'). Author's changes in the manuscript are given in red. Furthermore, a track changes version is included in the reply.

Introduction
L5 Reword 'which is in turn controlled by light ...'
L7 Reword 'Only the annual new production, however, leaves...'

Reply: We followed the suggestions of the referee and reworded the terms to '…,which is in turn controlled by light…' and 'Only the annual new production, however, leaves...', respectively.

L8 'as it represents' should perhaps rather read something like 'and is reduced by pelagic remineralization that reduces benthic food supply'?

Reply: The sentence was changed into 'Benthic remineralisation is negatively correlated to water depth (Jahnke et al, 1990; Jahnke, 1996; Wenzhöfer and Glud, 2002), a consequence of pelagic remineralisation (Rullkötter, 2006; Belcher et al., 2016) which lowers the benthic food supply.'

Methods 2.2
l15. Why short term examinations? Presumably the data were analyzed with enough time at hand. Rather 'Three time periods were chosen for analysis of sea ice concentrations'?

Reply: We followed the suggestion of the referee and reworded the terms to 'Three additional and shorter time periods were chosen for analysis of sea ice concentrations:…'

Results 3.2
l19. The reader needs to hop back and forth between table 1 (to find the depths) and table 3 (and 2 for that matter) to understand the text. While adding yet another line into the table, adding in depths into the header underneath the stations names would make it easier to read the table. (And in Table 1, I doubt the depth values make sense to give in decimal meters, round to 1 m).

Reply: We followed the suggestion of the referee and added 'water depth' underneath the stations names of table 2 and table 3. In addition, water depth values are given without decimal place.

Table 3: explain abbreviations in table caption (CPE, FDA etc.) – tables should be understood by themselves.

Reply: We added missing explanations of abbreviation to the table caption.

L26 ff. For all other properties, the lowest value is given first, then the highest, would be nice to be consistent here.

Reply: We re-arranged the order of porosity values, with the lowest values always mentioned first.

P12 l3 I am not sure why the data indicate that the food is received at the same time in the two ice regimes, please explain.

Reply: As the Arctic benthos receives mainly one large food input per year, the spring bloom (Wassmann, 2011), and Chl a is degraded to Phaeo over time (Kirk, 2011), the benthic Chl a/Phaeo ratio will be high, when the spring bloom arrives on the seafloor, and decrease with ongoing degradation. Therefore, the Chl a/Phaeo ratio represents the time since the spring bloom arrives. We added this information the 'Material and methods' section.

Results 3.3
l25 Need adverb: 'able to rework the sediment more strongly than' or 'to a larger degree than'

Reply: We followed the suggestion of the referee and rewrote the sentence to 'This indicates that the benthic macrofauna community in the WS area is potentially able to rework the sediment to a larger degree than the benthic macrofauna community in the EG area.'

Results 3.4
p13 l34 Spell out DOU first time used.

Reply: The abbreviation DOU was spelled out first time in the 'Material and Methods' section, p8 L16.

P14 l8 TOU should be spelled out the first time mentioned, not after it has been mentioned several times.

Reply: We followed the suggestion of the referee and omitted the abbreviation of 'TOU' and 'DOU' from the sentence, as both were spelled out first time in the material and method section. It now reads 'The mean DOU/TOU ratio, which describes the fraction of the total community mediated oxygen flux covered by the microbial-mediated oxygen flux (Glud, 2008,)…'.

Results 3.5
l24. Replace 'in turn' with 'consequently,'

Reply: We replaced the term 'in turn' with the word 'consequently'.

l29 delete comma after correlated with

Reply: The comma after 'correlated with' was deleted.

P15 l14 I'd say 'essentially no difference between the HSC and LSC' (R is very low, but not zero)
P15 l20 I'd say 'could at least partly explain' given the rather moderate R value

Reply: We followed the suggestions of the referee and inserted 'essentially no difference between the HSC and LSC' and 'could at least partly explain', respectively.

P16L31 Sentence starting with however is missing a verb – or tie to previous sentence.

Reply: We merged the sentence with the previous. It now reads 'The macrofauna community structure also differed between areas with high and low sea-ice cover, when sea ice and water depth was taken into account.'

P17l Use plural: 'shows correlations which do not'
P17l18 Why 'lower', lower than what? Just 'low' I suggest.
L21 singular 'remineralization responds'

Reply: We followed the suggestions of the referee and changed the text to 'shows correlations which do not', 'low', and 'remineralization responds', respectively.

Discussion 4.2.
l8 Rather 'The new production ... is estimated at 55 g ... '
L16 Typo 'reaching'

Reply: The words were changed according to the referee's suggestion, respectively.

P18 l30 ff Add to the caution that the flux rates also have some error associated with them.

Reply: We added the information, that flux rates also have some methodological-related uncertainties. The sentence reads now 'Furthermore, measurements of the benthic oxygen flux, crucial to evaluate the pelagic-benthic-coupling, remain only snapshots of remineralisation and also have some uncertainties (TOU: ~5 %, DOU: ~ 1 %) due to methodological error propagation.'

Discussion 4.2:
Would Renaud's measurements of respiration and Link's nutrient / oxygen fluxes from the Beaufort Sea provide some insights here?

Reply: Both author measured in shallower water depth compared to our study. Renaud et al., 2007 measurements from the Beaufort Sea are in maximum of 420 m water depth and the Renaud et al., 2008 measurements around Svalbard are in maximum 503 m water depth. Link et al., 2013 measurements were conducted in maximum 577 m water depth. As our measurements were mainly conducted in water

depth >1000 m, we assess that their results will not provide a substantially insight into the topic of 'Primary production and benthic remineralisation in the Fram Strait'.

Discussion 4.3.
l18 I perceive the Boetius study as presenting vertical flux of quite labile material. The four cited studies may be predicting contradicting trends, perhaps acknowledge here that people don't necessarily agree on the net outcome of production, flux and the amount of labile matter reaching benthic communities in the future.

Reply: To acknowledge that research regarding net outcome of production, flux and the amount of labile matter reaching benthic communities in the future may not agree, we rewrote parts of the paragraph. It now reads 'An additional predicted effect of a progressive sea-ice cover reduction is an increasing annual matter flux towards the seafloor (Wassmann, 2011; this study). Results regarding the vertical flux of labile material are, however, contradictory (Hop et al., 2006; van Oevelen et al., 2011; Boetius et al., 2013). Consequently, the change in sea-ice cover in the Arctic Ocean may alter the quality and quantity of the organic matter flux to the seafloor, where it may influence benthic deep-sea communities, in both biomass and species mix (Jones et al., 2014; Harada, 2015). The comparable DOU of the EG and HG site at water depth >1500 m (Fig. 4) indicates, however, that the remineralisation by the deep-sea benthos will possibly remain stable in the Arctic Ocean.'

P19l23 Singular: nutrient supply increases. Period '.' after primary production.
P20 l13 If 'its' refers to observations it should read 'their'

We followed the suggestions of the referee and changed the text to '…nutrient supply increases,', 'Despite their uncertainties,…' and added '.' after '…primary production'.

P19l13 ff I do not follow the logic. This paper is based on a time series. The last paragraph is essentially redundant with what was stated earlier and not really necessary.

General:
All figure captions: Abbreviations should be explained.

Reply: We followed the suggestions of the referee and added abbreviation explanations to the figure captions.

In addition, the manuscript was proofread by a native English speaker, namely Ph.D. Autun Purser from 
[revised manuscript text omitted]

|  |  | 32.1 % | | 55.9 % | | 75.3 % | |